# An Effective Manifold-based Optimization Method for Distributionally Robust Classification

**Jiawei Huang**[1,2], **Hu Ding**[1*]
1, School of Computer Science and Technology, University of Science and Technology of China
2, Department of Computer Science, City University of Hong Kong
`hjw0330@mail.ustc.edu.cn, huding@ustc.edu.cn`

## ABSTRACT

How to promote the *robustness* of existing deep learning models is a challenging problem for many practical classification tasks. Recently, *Distributionally Robust Optimization (DRO)* methods have shown promising potential to tackle this problem. These methods aim to construct reliable models by minimizing the worst-case risk within a local region (called "uncertainty set") around the empirical data distribution. However, conventional DRO methods tend to be overly pessimistic, leading to certain discrepancy between the real data distribution and the uncertainty set, which can degrade the classification performance. To address this issue, we propose a manifold-based DRO method that takes the geometric structure of training data into account for constructing the uncertainty set. Specifically, our method employs a carefully designed "game" that integrates contrastive learning with Jacobian regularization to capture the manifold structure, enabling us to solve DRO problems constrained by the data manifold. By utilizing a novel idea for approximating geodesic distance on manifolds, we also provide the theoretical guarantees for its robustness. Moreover, our proposed method is easy to implement in practice. We conduct a set of experiments on several popular benchmark datasets, where the results demonstrate our advantages in terms of accuracy and robustness.

## 1 INTRODUCTION

Neural network-based image classification methods have achieved significant success (He et al., 2016), but the data shift issue is still a major obstacle that seriously affects their practical performance (Wiles et al., 2022; Fang et al., 2020; Taori et al., 2020). *Data distribution shift* refers to the situation that the training and testing data distributions have a certain degree of discrepancy, which can lead to a decrease in the generalization of the model (Tang et al., 2021; Zhou et al., 2021; Lu et al., 2020; Zhang et al., 2022). Real-world machine learning tasks often encounter such issues due to various reasons, such as the changes of data sources (Koh et al., 2021; Lin et al., 2021) or the process for data collection (Dragoi et al., 2022; García et al., 2015).

If we have sufficient prior knowledge to assume that, the real data distribution falls within an area surrounding the training data (we denote this area as the "**uncertainty set**"), it could be possible to enhance the performance by considering all possible data distributions within such an uncertainty set. Namely, the goal is to promote the worst-case performance in this uncertainty set. This idea is called "**distributionally robust optimization (DRO)**" (Rahimian & Mehrotra, 2019), which has been used in many machine learning models to address distribution shift (Sagawa et al., 2020; Staib & Jegelka, 2019; Samuel & Chechik, 2021). In particular, the DRO methods can also be implemented on neural networks through the techniques like gradient flow (Wang et al., 2022), iterative optimization (Sinha et al., 2018), and surrogate loss (Samuel & Chechik, 2021).

The uncertainty set of DRO is typically defined as the collection of distributions that lie within a bounded distance from the empirical distribution of the training data. Common choices for mea-

---

*Corresponding author.

suring this distance include $f$-divergences (Duchi & Namkoong, 2021) and the Wasserstein distance (Gao et al., 2024; Li et al., 2019a). The Wasserstein distance is a popular metric for comparing data distributions, which effectively captures pointwise distance information between samples, making it well-suited for the tasks like pattern matching, style transfer, and handling distribution shift (Li et al., 2019b; Ling & Okada, 2007). A formal definition of Wasserstein distance is provided in Section 3. The DRO methods relying on Wasserstein distance are called "**Wasserstein-based DRO (WDRO)**" (Mohajerin Esfahani et al., 2018).

DRO offers significant advantages in terms of generalization and robustness across various domains (Gao et al., 2024; Huang et al., 2022; Wang et al., 2022). However, a major challenge in implementing DRO lies in the design of an appropriate uncertainty set. An ideal uncertainty set should effectively capture the underlying distributional uncertainty while can be easily integrated into the formulation for optimization. Consider WDRO as an example, where we denote the training data by $P_{\text{tr}}$. The conventional WDRO approaches often define the following uncertainty set

$$\mathcal{U}(P_{\text{tr}}, \delta) = \{Q \in \mathcal{P}(\mathbb{R}^d) | \mathcal{W}(Q, P_{\text{tr}}) \leq \delta\}, \tag{1}$$

where $\mathcal{P}(\mathbb{R}^d)$ is the set of all distributions in $\mathbb{R}^d$, $\mathcal{W}(\cdot, \cdot)$ indicates the Wasserstein distance function and $\delta > 0$ is a pre-specified range for uncertainty. This formulation implies that the uncertainty set $\mathcal{U}(P_{\text{tr}}, \delta)$ includes all distributions whose Wasserstein distance from $P_{\text{tr}}$ is at most $\delta$. However, we are often confronted with a dilemma of setting $\delta$. On the one hand, $\delta$ should be set large enough to ensure robustness. On the other hand, a large $\delta$ can also lead to a reduced accuracy on real data, as the uncertainty set $\mathcal{U}(P_{\text{tr}}, \delta)$ could encompass unrealistic distributions that deviate significantly from the true data-generating process.

Our intuition for addressing the above issue comes from the fact that realistic high-dimensional data (*e.g.,* images) typically locate nearby a low-dimensional manifold (Roweis & Saul, 2000; Tenenbaum et al., 2000; Yang et al., 2024). In addition, we can utilize the powerful representational capabilities of neural networks to capture the information from the underlying manifold (Chung et al., 2016; Cohen et al., 2020). Under these observations, it is reasonable to assume that the data is supported on a manifold $\mathcal{M}$, and we consider to construct a new uncertainty set as follows,

$$\mathcal{U}_{gw}(P_{\text{tr}}, \delta) = \{Q \in \mathcal{P}(\mathcal{M}) | \mathcal{GW}(Q, P_{\text{tr}}) \leq \delta)\}, \tag{2}$$

where $\mathcal{GW}(\cdot, \cdot)$ represents the Geodesic Wasserstein distance (formally defined in Section 3), and $\mathcal{P}(\mathcal{M})$ is the set of distributions supported on the manifold $\mathcal{M}$. This uncertainty set $\mathcal{U}_{gw}(P_{\text{tr}}, \delta)$ defined in Eq.(2) enjoys two important benefits compared with $\mathcal{U}(P_{\text{tr}}, \delta)$ defined in Eq.(1). First, it restricts the distribution to locate nearby the data manifold, preventing the DRO algorithm from being bothered by irrelevant points in the ambient space $\mathbb{R}^d$. Please see an illustrative case in Figure (1a). Second, we consider the geodesic distance on the manifold, which offers a more precise measure of the semantic distance between samples for classification (Criminisi et al., 2008; Wang et al., 2017). To illustrate this, we consider a toy example shown in Figure (1b): the two samples (images) "$A$" and "$B$" lying on the manifold $\mathcal{M}$ are not semantically similar, although they have shorter pairwise Euclidean distance (compared with their geodesic distance).

However, solving the WDRO problem with the uncertainty set $\mathcal{U}_{gw}(P_{\text{tr}}, \delta)$ presents significant challenges. The first challenge is due to the lack of neat representation for the manifold in neural networks; second, it is hard to add explicit constraints to the objective function to satisfy the uncertainty set in WDRO. In this paper, we aim to develop an effective and easy-to-implement approach to tackle these challenges, and our contributions are summarized below:

1. **A novel WDRO model with manifold guided constraint.** Our proposed model can effectively identify the directions of semantic variation in the data manifold, where the key idea relies on a carefully designed game between Contrastive Learning (CL) (Chen et al., 2020) and Jacobian regularization in the training process. Intuitively, CL is used to encourage the data to be updated along semantic-variant tangent directions of the manifold, while the Jacobian regularization is for suppressing the updating direction orthogonal to the manifold. Our training process equipped with such a game can help us to efficiently realize the WDRO within the uncertainty set defined in Eq.(2).

2. **An efficient algorithm for solving the WDRO model.** We are aware of several WDRO algorithms for neural networks, such as (Sinha et al., 2018; Bui et al., 2022), but they are unable to be directly extended to the manifold constraints, to our best knowledge. To

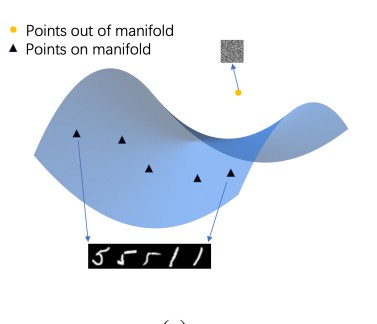
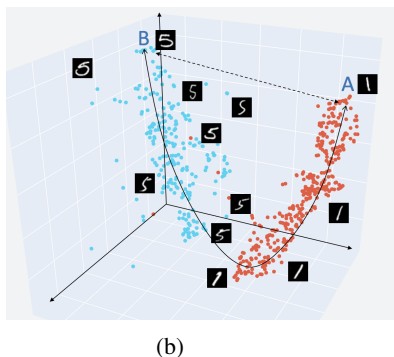

(a)                                                    (b)

Figure 1: (a) Real images typically reside on or nearby a manifold in the high-dimensional space, whereas points far away from the manifold could be meaningless, such as random noise (as represented by the orange point). (b) 3D visualization for a part of the MNIST dataset (Deng, 2012) by t-SNE (Van der Maaten & Hinton, 2008). The black curve and dashed line respectively represent the geodesic and Euclidean distances between two images A (digit "1") and B (digit "5"). Though their geodesic distance is large, their Euclidean distance is relatively short. Obviously, the geodesic distance can reveal their semantic difference more accurately.

efficiently capture the manifold information, we utilize the principal singular vectors for approximating the desired tangent directions of the manifold. Another critical idea in our algorithm is utilizing the accumulated optimization trajectory for estimating the geodesic distance. Thanks to the strong concavity of the DRO's dual form, we can guarantee the robustness with this estimation in theory. Our algorithm is easy to implement, and the experimental results illustrate its effectiveness for classification tasks on several benchmark datasets.

## 1.1 RELATIONS TO OTHER DRO METHODS.

The works of (Staib & Jegelka, 2017; Sinha et al., 2018; Bui et al., 2022) are closely related to ours, as they also use the dual form of WDRO to search for worst-case perturbations. However, these approaches cannot be directly extended to manifold constraints. While they are effective against adversarial attacks, their overly pessimistic nature can decrease performance on clean data. Another approach is called "Group DRO" (Sagawa et al., 2020), which aims to optimize performance across different demographic subgroups, addressing hidden biases and ensuring fairness. The method of $f$-divergence DRO (Namkoong & Duchi, 2016; Duchi & Namkoong, 2021), considers the uncertainty set within a certain $f$-divergence from the nominal distribution. Several studies tried to incorporate geometric information into the uncertainty set. Liu et al. (2022) used the shortest path over a constructed graph as the geodesic distance to guide distribution density transportation. Qiao & Peng (2023) calculated group centrality based on the affinity between data groups, serving as a good nominal distribution for group DRO. A full introduction of related works is placed in Appendix A.2.

## 2 PRELIMINARIES

**Notations.** Let "$P_{\text{tr}}$" represent the input set of $n$ labeled data items $(\mathbf{x}_1, y_1), \cdots, (\mathbf{x}_n, y_n)$. For each $1 \leq i \leq n$, $\mathbf{x}_i$ belongs to the input space $\mathcal{X} \subset \mathbb{R}^d$ and $y_i$ belongs to the label space $\mathcal{Y} = \{1, \cdots, C\}$, with $C$ being the number of classes. In this paper, we interchangeably use $\boldsymbol{p_i}$ and $(\mathbf{x}_i, y_i)$ to denote the $i$-th sample point in $P_{\text{tr}}$. The Euclidean and Frobenius norms are denoted by $|| \cdot ||$ and $|| \cdot ||_F$, respectively. We assume $P_{\text{tr}}$ lies near a low-dimensional manifold $\mathcal{M}$ in $\mathbb{R}^d$; $\mathcal{P}(\mathcal{M})$ denotes the collection of distributions supported on the manifold $\mathcal{M}$. For any point $\mathbf{x} \in \mathcal{M}$, $\text{T}_{\mathcal{M}}(\mathbf{x})$ denotes the *tangent space* of $\mathcal{M}$ at $\mathbf{x}$, which comprises all the tangent vectors at $\mathbf{x}$.

Our neural network is denoted as $\boldsymbol{\Gamma} = \mathbf{h} \circ \mathbf{g}$, where $\mathbf{g}(\cdot) : \mathcal{X} \to \mathcal{E}$ is the encoder, $\mathbf{h}(\cdot) : \mathcal{E} \to \mathcal{Y}$ is the classification head and $\mathcal{E}$ is the feature space. The network $\boldsymbol{\Gamma}$ is parameterized by a vector

$\theta \in \Theta$, where $\Theta$ is the parameter space. We denote the empirical loss as

$$\mathcal{L}(\theta, P_{\text{tr}}) = \sum_{i=1}^{n} \ell(\theta, p_i), \tag{3}$$

where $\ell(\theta, p_i)$ represents the induced loss of the prediction $\mathbf{\Gamma}(\theta, p_i)$. We are aware of several existing works that focus on the learned manifolds in "feature space" (LEE et al., 2022; Ma et al., 2023). But it is worth noting that the manifold studied in our current work is about the "data space". To emphasize this distinction, we refer to the manifolds where $\mathbf{x}$ and $\mathbf{g}(\mathbf{x})$ reside as the **data manifold** and the **representation manifold**, respectively.

Roughly speaking, a manifold is a space where each local area around a point is homeomorphic to a lower dimensional Euclidean space. We refer readers to the recent textbook (Boumal, 2023) for the formal definition of manifold. Given two points on a manifold, their *geodesic* is the curve between them on the manifold that locally minimizes the path length, and the length of the shortest curve is called the *geodesic distance*, which is denoted as $\mathbf{d_g}(\cdot, \cdot)$. We also need the following two assumptions for $\mathcal{M}$, which are both commonly adopted for optimization problems on manifold (Boumal, 2023).

**Assumption 1** *$\mathcal{M}$ is a complete manifold embedded in an Euclidean space.*

Due to the space limit, we leave more explanations on the rationale of Assumption 1 in Section A.1. Before stating the second assumption, we introduce two important definitions.

**Definition 1 (Exponential map (Zhang & Sra, 2016))** *Given a point $p \in \mathcal{M}$ and a vector $\mathbf{v} \in \mathrm{T}_{\mathcal{M}}(p)$, let $\gamma : [0, 1] \to \mathcal{M}$ be the geodesic trajectory starting from $p$, satisfying $\gamma(0) = p$ and $\gamma'(0) = \mathbf{v}$ with the uniform velocity $\|\mathbf{v}\|$. We use $q := \mathrm{Exp}_p(\mathbf{v})$ to denote the endpoint $\gamma(1)$, which is called the exponential map at $(p, \mathbf{v})$.*

The following Definition 2 generalizes the concepts of $\mu$-strongly convexity and $\beta$-smoothness from Euclidean space to manifold.

**Definition 2 (Geodesically $\mu$-strongly convex and $\beta$-smooth)** *We say a function $f : \mathcal{M} \to \mathbb{R}$ is geodesically $\mu$-strongly convex if for any given $\mathbf{x} \in \mathcal{M}, \mathbf{v} \in \mathrm{T}_{\mathcal{M}}(\mathbf{x}), t \in [0, 1]$, the inequality $f\left(\mathrm{Exp}_{\mathbf{x}}(t\mathbf{v})\right) \geq f(\mathbf{x}) + t\langle \nabla^{\mathcal{M}} f(\mathbf{x}), \mathbf{v} \rangle + t^2 \frac{\mu}{2} \|\mathbf{v}\|^2$ holds. Furthermore, we say $f$ is geodesically $\beta$-smooth if for any given $\mathbf{x} \in \mathcal{M}, \mathbf{v} \in \mathrm{T}_{\mathcal{M}}(\mathbf{x}), t \in [0, 1]$, the inequality $f\left(\mathrm{Exp}_{\mathbf{x}}(t\mathbf{v})\right) \leq f(\mathbf{x}) + t\langle \nabla^{\mathcal{M}} f(\mathbf{x}), \mathbf{v} \rangle + t^2 \frac{\beta}{2} \|\mathbf{v}\|^2$ holds.*

In this definition, "$\nabla^{\mathcal{M}}(f(\mathbf{x}))$" represents the *Riemannian gradient* of $f$ at $\mathbf{x}$, a tangent vector in $\mathrm{T}_{\mathcal{M}}(\mathbf{x})$ indicating the steepest ascent direction in function value. Under Assumption 1, it equals the orthogonal projection of the "classical" gradient to the tangent spaces $\mathrm{T}_{\mathcal{M}}(\mathbf{x})$ (Boumal, 2023). *i.e.* $\nabla^{\mathcal{M}} f(\mathbf{x}) = \mathrm{Proj}_{\mathbf{x}}(\nabla f(\mathbf{x}))$, where the $\mathrm{Proj}_{\mathbf{x}}(\cdot)$ is the *orthogonal projection operator* that projects a vector to the tangent space $\mathrm{T}_{\mathcal{M}}(\mathbf{x})$ and $\nabla(f(\mathbf{x}))$ is the gradient in Euclidean space $\mathbb{R}^d$.

**Assumption 2** *For any $\theta \in \Theta$, the loss function $\ell(\theta, \mathbf{x})$ is geodesically $\beta$-smooth w.r.t. any $\mathbf{x} \in \mathcal{M}$.*

## 3 OUR FORMULATION: MANIFOLD-BASED WDRO

We name our model using the uncertainty set (2) as *Manifold-based WDRO (MWDRO)*, which relies on a new concept called "geodesic Wasserstein distance". Given two distributions $P$ and $Q$, the standard Wasserstein distance measures the optimal transportation cost between them (Peyré et al., 2019; Villani, 2009). Following that, we define the geodesic Wasserstein distance, where we use the geodesic distance to measure the pairwise distance between points on the manifold $\mathcal{M}$:

$$\mathcal{GW}(Q, P) = \left( \inf_{\pi \in \Pi(Q, P)} \int_{\mathcal{M} \times \mathcal{M}} \mathbf{d_g}^2(p, q) \mathrm{d}\pi(q, p) \right)^{\frac{1}{2}}, \tag{4}$$

where the $\Pi(Q, P)$ represents the collection of all possible joint distributions combining $Q$ and $P$ as their marginal distributions. $\mathcal{GW}(Q, P)$ indicates the minimum cost required to transport one distribution into another under the geodesic distance $\mathbf{d_g}$.

As mentioned before, our objective is to achieve the distributionally robust optimization over the uncertainty set $\mathcal{U}_{gw}(P_{\mathrm{tr}}, \delta)$, as shown in Eq.(2). To realize this goal, we optimize $\theta \in \Theta$ by minimizing the following MWDRO loss to replace the vanilla empirical loss $\mathcal{L}(\theta, P_{\mathrm{tr}})$:

$$\mathcal{L}_{DR}^{\delta}(\theta, P_{\mathrm{tr}}) = \sup_{Q \in \mathcal{U}_{gw}(P_{\mathrm{tr}}, \delta)} \{\mathbb{E}_{q \in Q}[\ell(\theta, q)]\}. \tag{5}$$

The above formulation is intractable because it needs to consider continuous distributions within the uncertainty set under Wasserstein distance; as discussed in (Gao & Kleywegt, 2023), such a WDRO model involving continuous distributions is an infinite-dimensional optimization problem. Although a number of theoretical works have been proposed for developing tractable formulations for DRO problems (Esfahani & Kuhn, 2015; Shafieezadeh Abadeh et al., 2015; Mohajerin Esfahani et al., 2018; Postek et al., 2016; Lee & Mehrotra, 2015; Hanasusanto & Kuhn, 2018), these methods are not applicable to our case due to the manifold constraint. To solve the problem Eq.(5), we adopt the strongly duality property proposed in (Gao & Kleywegt, 2023, Theorem 1) to obtain the dual form (as Proposition 1 below), and then design our approach with several new ideas relying on this dual form. Note that their original article considers the general metric space rather than manifold. Because the geodesic distance also forms a metric on the manifold, we can derive the strong dual reformulation for Eq.(5) (see Section A.1).

**Proposition 1** *For a given $\theta$ and the data distribution shift threshold $\delta > 0$, the "worst-case" loss in Eq.(5) can be reformulated as*

$$\mathcal{L}_{DR}^{\delta}(\theta, P_{\mathrm{tr}}) = \min_{\nu \geq 0} \{\nu \delta^2 + \mathbb{E}_{P_{\mathrm{tr}}} \ell_s(\theta, p_i, \nu)\}, \tag{6}$$

$$\text{where } \ell_s(\theta, p_i, \nu) := \sup_{q \in \mathcal{M}} \left[\ell(\theta, q) - \nu \mathrm{d}_{\mathbf{g}}^2(q, p_i)\right]. \tag{7}$$

**Remark 1** *Eq.(6) is the dual form of Eq.(5), that is, the optimal value of Eq.(6) is also the optimal value of Eq.(5). A benefit of such reformulation is that we can ignore the complicated uncertainty set $\mathcal{U}_{gw}(P_{\mathrm{tr}}, \delta)$. Instead, we only add a surrogate loss $\mathbb{E}_{P_{\mathrm{tr}}} \ell_s(\theta, p_i, \nu)$ to the Eq.(6), which yields a more succinct formulation for optimizing the problem. However, there are still several significant challenges remaining unsolved,* e.g., *the feasible domain is confined to the manifold $\mathcal{M}$. We will elaborate on the details for solving this new formulation in Section 4.*

## 4 OUR APPROACH FOR SOLVING MWDRO

For ease of understanding, we present an overview of our approach in Section 4.1, showing how to incorporate the (approximate) manifold constraint into the training process. Next, we offer a detailed analysis on the manifold-guided game in Section 4.2, explaining why this game can effectively encodes the manifold's tangent into our model. Finally, we outline our MWDRO algorithm in Section 4.3, together with the theoretical quality guarantees.

### 4.1 THE OVERVIEW OF OUR APPROACH

We adopt the dual formulation from Proposition 1 as the objective function in our model. However, a major challenge for solving the objective function arises from the surrogate loss in Eq.(7), as it is difficult to find the supremum within a manifold (which is a non-convex region in high dimensional space). To solve this optimization problem on the manifold, we require the Riemannian gradient, which is the orthogonal projection of the "classical" gradient onto the tangent spaces (Boumal, 2023), *i.e.,* $\nabla^{\mathcal{M}} f(\mathbf{x}) = \mathrm{Proj}_{\mathbf{x}} (\nabla f(\mathbf{x}))$ as shown in our preliminaries. The central problem, therefore, reduces to recovering the tangent space $T_{\mathcal{M}}(\mathbf{x})$ for any data point $\mathbf{x} \in \mathcal{M}$. However, obtaining the tangent space on the data manifold is not an easy job, as the manifold lacks a closed-form representation. Our idea for addressing this issue comes from the fact that neural networks usually have powerful representation abilities, especially for encoding manifold information (Bronstein et al., 2021; Hu et al., 2023). This motivates us to ask the following question:

*Can neural networks also be used to extract the tangent space of a data manifold?*

**High-level idea.** We should emphasize that the learned representation for manifold from neural networks is not sufficient for extracting tangent space, as demonstrated experimentally in Figure 11

(Appendix G). A key part of our approach is based on a *manifold-guided game* for minimizing a Contrastive Learning (CL) loss (Chen et al., 2020) and a Jacobian regularization term. Roughly speaking, a CL loss is used for amplifying the differences between samples with distinct semantics in the feature space; on the other hand, a Jacobian regularization term tries to suppress the differences in the feature space for preserving the manifold constraint (if we do not have such suppression, the updating direction could seriously deviate from the manifold). The combination of these two components enables the encoder $\mathbf{g}(\cdot)$ to **remain sensitive to variations along the manifold while insensitive to other directions** (a fine-grained analysis is provided in Section 4.2). As a result, the Jacobian matrix $J_{\mathbf{g}}(\mathbf{x})$ aligns its principal singular vectors with (part of) the manifold's tangent space. This alignment implicitly encodes the data manifold's structure and facilitates effective manifold optimization for MWDRO, as detailed in Section 4.3.

Formally, we introduce the loss for our manifold-guided game:

$$\mathcal{L}^*(\theta) = \mathcal{L}_{\text{InfoNCE}} + \lambda_1 \mathbb{E}_{\mathbf{x} \in P_{\text{tr}}} \|J_{\mathbf{g}}(\mathbf{x})\|_F^2, \tag{8}$$

where the term $\mathcal{L}_{\text{InfoNCE}}$ is the objective function of CL (that will be explained latter) and the term $J_{\mathbf{g}}(\mathbf{x}) := \frac{\partial_{\mathbf{x}} \mathbf{g}(\theta, \mathbf{x})}{\partial \mathbf{x}}$ is the Jacobian matrix of the encoder $\mathbf{g}(\cdot)$ w.r.t. the sample $\mathbf{x}$. The $\mathcal{L}_{\text{InfoNCE}}$ and the Jacobian regularization terms are balanced by the coefficient $\lambda_1 > 0$.

For completeness, we provide more details on the popular self-supervised method CL. For each $\boldsymbol{x}_i \in P_{\text{tr}}$, we denote its two randomly generated copies as $\boldsymbol{x}_i', \boldsymbol{x}_i''$ via some random semantic-preserving augmentation operation (such as cropping or rotating the image). Let $P_{\text{tr}}' = \{\boldsymbol{x}_1', \cdots, \boldsymbol{x}_n'\}$ and $P_{\text{tr}}'' = \{\boldsymbol{x}_1'', \cdots, \boldsymbol{x}_n''\}$. Given any $1 \le i \ne j \le n$, we say the couple $(\boldsymbol{x}_i', \boldsymbol{x}_i'')$ is a "positive pair", and the couples $(\boldsymbol{x}_i', \boldsymbol{x}_j')$ and $(\boldsymbol{x}_i', \boldsymbol{x}_j'')$ are "negative pairs". The main idea of CL is to align representations of positive pairs closer, while diversify representations of negative pairs (Tian et al., 2020; Wang & Isola, 2020; Xue et al., 2022). The *Information Noise-Contrastive Estimation (InfoNCE) loss* is a commonly used formulation in CL models (Chen et al., 2020; He et al., 2020):

$$\mathcal{L}_{\text{InfoNCE}} = \mathbb{E}_{i \in [n]} [\ell_{\text{cl}}(\boldsymbol{x}_i)],$$

$$\text{where } \ell_{\text{cl}}(\boldsymbol{x}_i) = -\log \frac{\exp\left(\text{sim}\left(\mathbf{g}(\boldsymbol{x}_i'), \mathbf{g}(\boldsymbol{x}_i'')\right)/\tau\right)}{\sum_{\boldsymbol{x} \in P_{\text{tr}}' \cup P_{\text{tr}}'' \setminus \{\boldsymbol{x}_i', \boldsymbol{x}_i''\}} \exp\left(\text{sim}\left(\mathbf{g}\left(\boldsymbol{x}_i'\right), \mathbf{g}(\boldsymbol{x})\right)/\tau\right)}, \tag{9}$$

$\tau > 0$ is a scalar temperature hyperparameter, and $\text{sim}(\boldsymbol{u}, \boldsymbol{v}) = \boldsymbol{u}^\top \boldsymbol{v} / \|\boldsymbol{u}\| \|\boldsymbol{v}\|$ is the cosine similarity. Intuitively, the InfoNCE loss tends to bring $\mathbf{g}(\boldsymbol{x}_i')$ and $\mathbf{g}(\boldsymbol{x}_i'')$ to be closer, and meanwhile repulse $\mathbf{g}(\boldsymbol{x}_i')$ and $\mathbf{g}(\boldsymbol{x})$.

## 4.2 FINE-GRAINED ANALYSIS ON THE MANIFOLD-GUIDED GAME

In this section, we explain why our proposed method in Section 4.1 can effectively encode the tangent information of the data manifold into our model. More specifically, what is the advantage of combining the CL and Jacobian regularization term in our training method, as proposed in Eq.(8)?

To begin our analysis, we need to define some notations. For each data sample $\mathbf{x} \in \mathcal{M}$, we define the region containing all the augmentations of $\mathbf{x}$ (such as cropping or rotating in CL) on $\mathcal{M}$ as the *Semantic-Invariant region* "$\mathcal{M}_{SI}(\mathbf{x})$". We also denote $\mathcal{M}^\perp := \mathbb{R}^d \setminus \mathcal{M}$ as the *Out-of-Manifold region*. Consider the singular value decomposition (SVD) of $J_{\mathbf{g}}(\mathbf{x}) = \mathbf{U}(\mathbf{x})\boldsymbol{\Sigma}(\mathbf{x})\mathbf{V}^\top(\mathbf{x})$, where the rank of $J_{\mathbf{g}}(\mathbf{x})$ is $r > 0$, the diagonal matrix $\boldsymbol{\Sigma}(\mathbf{x}) = \text{diag}(\sigma_1(\mathbf{x}), \cdots, \sigma_r(\mathbf{x}))$, and the right singular vectors of $\mathbf{V}(\mathbf{x})$ contains $\{\mathbf{v}_1(\mathbf{x}), \cdots, \mathbf{v}_r(\mathbf{x})\}$ (these $r$ vectors form an orthonormal basis). Given a threshold $\tau_0 > 0$, we define the subspace spanned by the principal singular vectors of the Jacobian as:

$$\text{T}_{\text{appr}}(\mathbf{x}, \tau_0) = \left\{ \sum_{i \in I} a_i \mathbf{v}_i(\mathbf{x}) \mid a_i \in \mathbb{R} \right\}, \text{where } I = \{i \mid \sigma_i(\mathbf{x}) \ge \tau_0, 1 \le i \le r\}. \tag{10}$$

In Eq.(10), we truncate the small singular values less than $\tau_0$.

**Why the game helps to extract tangent information?** Due to the loss defined in Eq.(9), the training process of CL encourages the extracted features of $\boldsymbol{x}_i', \boldsymbol{x}_i'' \in \mathcal{M}_{SI}(\mathbf{x})$ to be close, while keeping the features of negative pairs distinct. Simultaneously, it is known that the Jacobian regularization suppresses the feature variations for perturbations of $\mathbf{x}$ across all directions (including both the directions tangent and normal to the manifold) (Ross & Doshi-Velez, 2018). Therefore, we can regard these two objectives as two players in a game: the CL loss preserves feature variations along

the tangents apart from the Semantic-Invariant region $\mathcal{M}_{SI}(\mathbf{x})$, so as to defend against the suppression from the Jacobian regularization. In other words, the manifold-guided game between CL and Jacobian regularization encourages the space spanned by the Jacobian's principal singular vectors (truncated by $\tau_0$), *i.e.*, $\mathrm{T}_{\mathrm{appr}}(\mathbf{x}, \tau_0)$, to approximately align with directions within the tangent space that induce semantic changes. Therefore, $\mathrm{T}_{\mathrm{appr}}(\mathbf{x}, \tau_0)$ serves as an effective approximation of a subset of the tangent $\mathrm{T}_{\mathcal{M}}(\mathbf{x})$, specifically capturing directions of significant semantic variation within $\mathrm{T}_{\mathcal{M}}(\mathbf{x})$. As we will see later in Section 4.3, $\mathrm{T}_{\mathrm{appr}}(\mathbf{x}, \tau_0)$ can be utilized to approximate the computation of the Riemannian gradient.

**Experimental verification.** In the above analysis, we assume that $\mathrm{T}_{\mathrm{appr}}(\mathbf{x}, \tau_0)$ is a subspace of the tangent $\mathrm{T}_{\mathcal{M}}(\mathbf{x})$. We explain the rationale of this assumption here. According to the property of manifold, a tangent plane characterizes the vicinity of a point on the manifold by a local linear approximation. Similarly, the Jacobian matrix $J_{\mathbf{g}}(\mathbf{x})$, which contains the partial derivatives of the features w.r.t. the input data, provides a linear approximation that captures how the feature $\mathbf{g}(\cdot)$ responds to input variations. Under our proposed manifold-guided game, $\mathrm{T}_{\mathrm{appr}}(\mathbf{x}, \tau_0)$ represents a subspace of the tangent $\mathrm{T}_{\mathcal{M}}(\mathbf{x})$ corresponding to semantic changes. Based on the low-dimensional manifold assumption, the dimension of $\mathrm{T}_{\mathrm{appr}}(\mathbf{x}, \tau_0)$ should be significantly smaller than $d$, indicating a low-rank structure of the Jacobian. In Figure 2, we plot the cumulative percentage of the singular values for the Jacobian matrix of the models trained on CIFAR-10/100. We can see that even a small number of top singular vectors can approximate the Jacobian $J_{\mathbf{g}}(\mathbf{x})$ quite well. For instance, in CIFAR-10 the cumulative percentage of the top five singular vectors already take over $95\%$ of the sum of all singular values. This concentration of singular values enables us to compute $\mathrm{T}_{\mathrm{appr}}(\mathbf{x}, \tau_0)$ with greater accuracy and efficiency (see Appendix E).

To see the benefit of using our designed game, we compare it with other models in Figure 2. Our model's cumulative percentage curve shows the steepest increase at the early stage. This suggests that we have effectively suppressed the encoder $\mathbf{g}(\cdot)$'s sensitivity to data variations across most directions. We can reasonably assume that the space spanned by the remaining primary singular vectors aligns with directions of semantic variation within the data manifold's tangent. This alignment facilitates the solving of MWDRO, as detailed in Section 4.3.

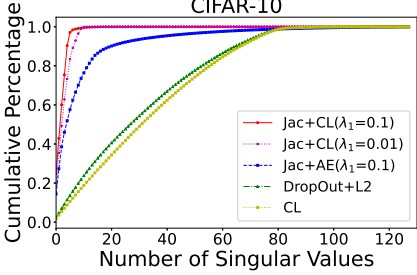 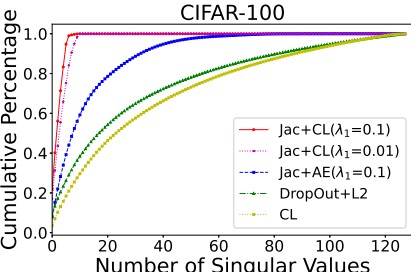

Figure 2: Cumulative percentage of variance of the Jacobian for different models trained on CIFAR-10 and CIFAR-100 using ResNet-18, respectively. The cumulative percentage of variance for the first $k$ singular values is calculated as $\frac{\sum_{i=1}^{k} \sigma_i(\mathbf{x})}{\sum_{i=1}^{r} \sigma_i(\mathbf{x})}$. The red line represents our model as described in Eq.(8). The blue line represents the Auto-Encoder model with Jacobian regularization (Jac+AE), and the green line represents the CL model with Dropout and L2 regularizer (Dropout+L2).

**Remark 2** *We also highlight several additional benefits of our proposed manifold-guided game from a representation learning perspective. By analyzing the characteristics of the encoder $\mathbf{g}(\cdot)$, we show that it is possible to further enhance our approach for capturing a smoother manifold. Due to the space limit, please refer to Appendix B for the detailed discussion.*

## 4.3 THE ALGORITHM VIA THE MANIFOLD-GUIDED GAME

In this section, we provide our algorithm to solve the MWDRO problem by minimizing Eq.(6) based on our analysis in Section 4.2. We minimize the MWDRO loss Eq.(6) with the constraint from the manifold-guided game Eq.(8), to keep the manifold's tangents being encoded into our model

effectively in the training process, *i.e.,*

$$\min_\theta \mathcal{L}_{DR}^\delta(\theta, P_{\text{tr}}), \quad \text{s.t.} \quad \mathcal{L}^*(\theta) \le \epsilon_0, \tag{11}$$

where $\epsilon_0 > 0$ is a pre-specified threshold. We convert the problem (11) into an unconstrained optimization problem by Lagrange duality (Boyd et al., 2004) with $\lambda_2 > 0$ controlling the magnitude of regularization:

$$\min_\theta \mathcal{L}_{DR}^\delta(\theta, P_{\text{tr}}) + \lambda_2 \mathcal{L}^*(\theta). \tag{12}$$

**Sketch of Algorithm 1.** Our MWDRO approach contains two stages. In the first stage, we train the encoder $\mathbf{g}(\cdot)$ by minimizing the manifold-guided loss in Eq.(8). The goal is to ensure $\mathbf{g}(\cdot)$ can yield sufficiently good approximate tangent $T_{\text{appr}}(\mathbf{x}, \tau_0)$ for each data sample $\mathbf{x}$, which is later used for Riemannian optimization to solve the surrogate loss. In the second stage, we optimize Eq.(12) by stochastic gradient descent algorithm. A key part of Algorithm 1 is to compute the "worst-case" distribution in each iteration, which is implemented by Algorithm 2. Roughly speaking, Algorithm 2 employs the Riemannian gradient ascent to approach the worst-case distribution. It is worth noting that the performance of Algorithm 2 heavily relies on the tangent plane provided by the encoder $\mathbf{g}(\cdot)$, and this is also the reason that we add the constraint "$\mathcal{L}^*(\theta) \le \epsilon_0$" in problem (11).

---

**Algorithm 1** Distributional Robust in Data Manifold $\mathcal{M}$

---

**Input:** training set $P_{\text{tr}} = \{(\mathbf{x}_i, y_i)\}_{i=1}^n$, a constant $\nu$, step size sequence $\{\alpha_j > 0\}_{j=0}^N$.
**Output:** the final parameter $\theta^{N+1}$.
 1: Train $\mathbf{g}(\cdot)$ to minimize $\mathcal{L}^*(\theta)$ in Eq.(8). */* The first stage. Training for $N_0$ epochs. */
 2: **for** $j = 1, 2, \cdots, N$ **do**
 3:     Take a sample from $P_{\text{tr}}$ uniformly at random, which is denoted as $p_{i_j}$, and compute a new point $\hat{p}_{i_j} \leftarrow \texttt{Evolve}(\{p_{i_j}\}, \mathbf{g}(\cdot), \nu)$; */* $\{p'_{i_1}, p'_{i_2}, \cdots, p'_{i_N}\}$ returned by the $\texttt{Evolve}$ function (Algorithm 2) is an approximation of the "worst case" distribution. */
 4:     $\theta^{j+1} \leftarrow \theta^j - \alpha_j \nabla_\theta \left( \ell(\theta^j, \hat{p}_{i_j}) - \lambda_2 \left( \ell_{\text{cl}}(p_{i_j}) + \lambda_1 \|J_{\mathbf{g}}(p_{i_j})\|_F^2 \right) \right)$; */* Objective Eq.(12). */
 5: **end for** */* The second stage. */

---

To deal with the objective function $\mathcal{L}_{DR}^\delta(\theta, P_{\text{tr}}) = \min_{\nu \ge 0} \{\nu \delta^2 + \mathbb{E}_{P_{\text{tr}}} \ell_s(\theta, p_i, \nu)\}$ in Eq.(12), we do not explicitly set the radius $\delta$ for the uncertainty set $\mathcal{U}_{gw}(P_{\text{tr}}, \delta)$ in Eq.(2). Instead, we select a $\nu > 0$ to solve the empirical surrogate loss $\mathbb{E}_{P_{\text{tr}}}[\ell_s(\theta, p_i, \nu)]$ rather than prescribing the robust range $\delta$; this approach, initially proposed by Sinha et al. (2018), has been shown to ensure certified distributional robustness. However, we cannot directly apply their method due to the lack of algorithmic tools for dealing with the manifold constraint. Thanks to our constructed approximate tangent $T_{\text{appr}}(\mathbf{x}, \tau_0)$ in Section 4.2, we can approximate the "worst case" distribution by performing gradient ascent on the manifold. This process is detailed in Algorithm 2. In the algorithm, we fix a $\nu > 0$ in Eq.(7), transforming it into a geodesically strongly concave maximization problem. We iteratively approach the optimal "$q$" through Riemannian gradient ascent on $\mathcal{M}$: generating a sequence $\{q^0 = q, q^1, \cdots, q^t, q^{t+1}, \cdots\}$. Let $\hat{u}_{q^t, q}$ denote the unit vector representing the initial direction of the geodesic from point $q^t$ to point $q$ on the manifold. Following optimization theory on manifold (Boumal, 2023), we update with a step size $\alpha > 0$ as follows:

$$\begin{aligned} q^{t+1} &= \text{Exp}_{q^t} \left( \alpha \nabla^{\mathcal{M}} \left( \ell(\theta, q^t) - \nu \mathbf{d}_{\mathbf{g}}^2(q^t, q) \right) \right) \\ &= \text{Exp}_{q^t} \left( \alpha \text{Proj}_{q^t} \left( \nabla \ell(\theta, q^t) \right) - 2\alpha\nu \mathbf{d}_{\mathbf{g}}(q^t, q) \hat{u}_{q^t, q} \right) \end{aligned} \tag{13}$$

**Approximation for the geodesic distance.** A challenge to solve Eq.(13) is how to estimate the geodesic distance $\mathbf{d}_{\mathbf{g}}^2(q^0, q^t)$. We propose an effective approximation technique for the geodesic distance $\mathbf{d}_{\mathbf{g}}(q^0, q^t)$. Specifically, we approximate $\mathbf{d}_{\mathbf{g}}(q^0, q^t)$ using the accumulated optimization trajectory length $\texttt{pt}^t := \sum_{s=0}^{t-1} \|\alpha \cdot \texttt{grad}^t\|$ in the Step 4 of Algorithm 2. Consequently, we can efficiently update the worst-case distribution as the Step 2 and 3 of Algorithm 2. Under mild smoothness assumptions, which typically hold for neural networks, we establish the quality of this approximation in Lemma 1.

**Lemma 1** *Under Assumption 1 and 2, we fix a constant $\nu > \beta$ to set up Algorithm 2 (recall $\beta$ is the smoothness parameter in Assumption 1). Let $q^*$ denote the optimal solution. Then, Algorithm 2 converges linearly; and for any $t > 0$, the following inequality holds:* $\texttt{pt}^t \le \texttt{pt}^\infty \le \sqrt{\kappa}(\sqrt{\kappa} + \sqrt{\kappa - 1})^2 \mathbf{d}_{\mathbf{g}}(q^0, q^*)$, *where* $\kappa = \frac{\nu + \beta}{\nu - \beta}$.

---

**Algorithm 2** `Evolve`$(\Xi, \mathbf{g}(\cdot), \nu)$

---

**Input:** A sample $q$, a constant $\nu$ and the encoder $\mathbf{g}(\cdot)$.
**Output:** $q^{t+1}$.
   **for** t = 1, 2, $\cdots$, until convergence **do**
      1. initialize the "potential" for $q$: $\mathtt{pt}^0 = 0$; */* Using "potential" for approximating the geodesic distance. */*
      2. $\mathtt{grad}^t = \mathrm{Proj}_{q^t}\left(\nabla_q l(\theta, q^t)\right) - 2\nu \mathtt{pt}^t \cdot \hat{u}_{q^t, q}$
      3. $q^{t+1} = \mathrm{Exp}_{q^t}(\alpha \cdot \mathtt{grad}^t)$; */* Conduct the gradient ascent on the manifold. */*
      4. $\mathtt{pt}^{t+1} = \mathtt{pt}^t + \|\alpha \cdot \mathtt{grad}^t\|_2$;
   **end for**

---

Lemma 1 implies that we can utilize the accumulated step size to approximate the geodesic distance. This approximation plays a key role in Theorem 1, which establishes the robustness guarantee under our geodesic distance approximation. The proofs are deferred to Appendix D.

**Theorem 1** *Suppose Assumption 1 and 2 hold. We select an $\nu > \beta$ in the surrogate loss (Eq.(7)), and define $\kappa$ as in Lemma 1. Using the accumulated step size to approximate the geodesic distance, let $\hat{\theta}$ be the optimal solution of the dual formulation under this approximation. Then $\mathbb{E}_{P_{\mathrm{tr}}}\ell_s(\hat{\theta}, p_i, \nu) \leq c'' \times \min_\theta \mathbb{E}_{P_{\mathrm{tr}}}\ell_s(\theta, p_i, \nu)$, where $c'' = \kappa^2(\sqrt{\kappa} + \sqrt{\kappa - 1})^4$.*

**Remark 3** *As an example, if we set $\nu \geq 17\beta$, the result of Theorem 1 indicates that $\mathbb{E}_{P_{\mathrm{tr}}}\ell_s(\hat{\theta}, p_i, \nu) \leq 5.1 \times \min_\theta \mathbb{E}_{P_{\mathrm{tr}}}\ell_s(\theta, p_i, \nu)$. It is also worth noting that a smaller $\beta$ implies a larger range for setting the parameter $\nu$, as we require $\nu > \beta$ in Theorem 1. The dependence of WDRO methods on the smoothness of the loss function has also been discussed in previous works. For instance, Sinha et al. (2018) showed that $\beta$ is typically small for neural networks. In Section D.1, we further explore the value of $\beta$ and reveal an interesting result: Jacobian regularization is also beneficial for smaller $\beta$.*

Algorithm 2 solves the surrogate loss using Riemannian gradient ascent. As introduced in Section 3 and Section 4.1, the Riemannian gradient can be formulated as: $\nabla^{\mathcal{M}}\ell(\theta, \mathbf{x}) = \mathrm{Proj}_\mathbf{x}(\nabla_\mathbf{x}\ell(\theta, \mathbf{x}))$, where $\mathrm{Proj}_\mathbf{x}$ denotes the projection of the Euclidean gradient onto the tangent space $\mathrm{T}_\mathcal{M}(\mathbf{x})$. However, this exact projection is not directly computable due to the lack of analytical formula for the tangent $\mathrm{T}_\mathcal{M}(\mathbf{x})$. To overcome this, as shown in Section 4.2, we approximate the tangent information using $\mathrm{T}_{\mathrm{appr}}(\mathbf{x}, \tau_0)$ by the manifold-guided game. Specifically, We approximate the Riemannian gradient as $\widehat{\nabla}^{\mathcal{M}}\ell(\theta, \mathbf{x}) := \mathrm{Proj}^{\tau_0}_{\mathrm{appr}}(\nabla\ell(\theta, \mathbf{x}))$, where the $\mathrm{Proj}^{\tau_0}_{\mathrm{appr}}(\cdot)$ is the *orthogonal projection operator* that projects the Euclidean gradient $\nabla_\mathbf{x}\ell(\theta, \mathbf{x})$ to the plane $\mathrm{T}_{\mathrm{appr}}(\mathbf{x}, \tau_0)$. In our problem, the loss function $\ell(\theta, \mathbf{x})$ is a composition of functions involving the encoder $\mathbf{g}(\cdot)$. This structure enables us to provide theoretical guarantees regarding the quality of the approximation $\mathrm{Proj}^{\tau_0}_{\mathrm{appr}}(\nabla\ell(\theta, \mathbf{x}))$. Informally, under the assumption that $\mathrm{T}_{\mathrm{appr}}(\mathbf{x}, \tau_0) \subseteq \mathrm{T}_\mathcal{M}(\mathbf{x})$ (we analyzed the rationale behind this assumption in Section 4.2), the error of our approximation $\left\|\nabla^{\mathcal{M}}\ell(\theta, \mathbf{x}) - \widehat{\nabla}^{\mathcal{M}}\ell(\theta, \mathbf{x})\right\|$ is bounded by $O(\tau_0)$ (due to space limit, we defer the formal statement to Theorem 2 in Appendix C).

**Other implementation details for the Algorithm 2.** We replace the exponential map operation $\mathrm{Exp}_{q^t}(\cdot)$ in Step 3 of Algorithm 2 with a retraction operation, *i.e.,* $q^{t+1} = q^t + \alpha \cdot \mathtt{grad}^t$, which is a widely used in manifold-based algorithms (Bécigneul & Ganea, 2019). Further implementation details, including efficient strategies for applying Jacobian regularization and computing the subspace $\mathrm{T}_{\mathrm{appr}}(\mathbf{x}, \tau_0)$, are discussed in Appendix E.

## 5 EXPERIMENTS

We conduct a series of experiments across several scenarios to evaluate our proposed method. Specifically, we test it on three typical distribution shift tasks: dealing with noisy data, attacked data and imbalanced data. All experiments are implemented with PyTorch on a single NVIDIA RTX 6000 Ada. In our experimental setup, we set the hyperparameters $\lambda_1 = 0.01$ and $\lambda_2 = 1$ in Eq.(8) and Eq.(12) respectively to configure the algorithm, unless otherwise specified.

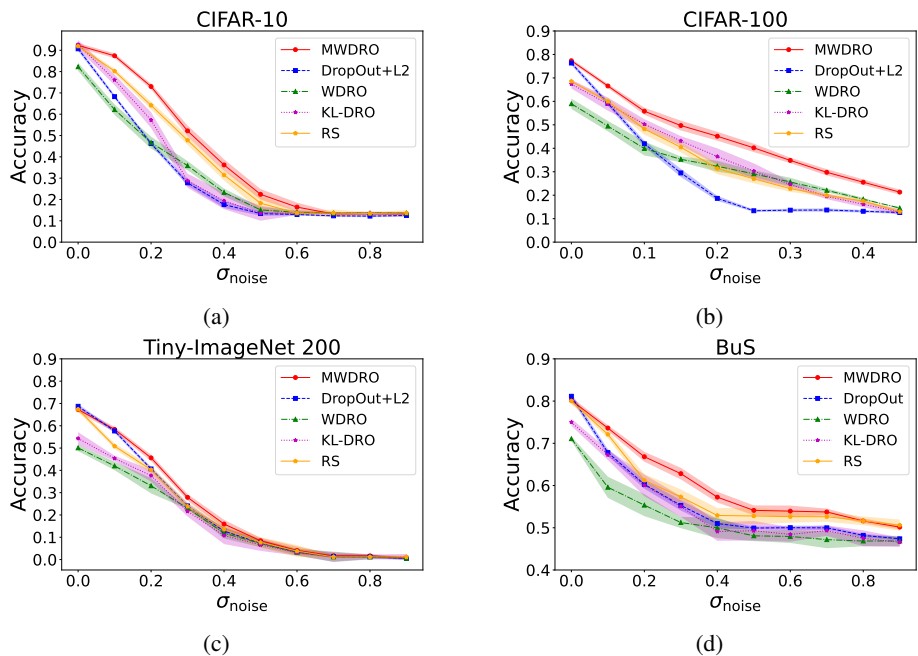

Figure 3: Robustness evaluation under Gaussian white noise perturbations. The x-axis represents the variance of the Gaussian noise, ranging from 0 to 1 (the input data has been normalized). The results indicate that MWDRO enhances robustness across four datasets compared to baselines.

**Basic Evaluation on Noisy Data.** We test our model MWDRO against natural image corruptions by adding Gaussian white noise to images. We use CIFAR-10/100 (Krizhevsky et al., 2012), Tiny-ImageNet-200 (Krizhevsky et al., 2017), and a medical imaging dataset BuS (Mo et al., 2023) for image classification. Our model is compared with DropOut+L2 (He et al., 2016), WDRO (Sinha et al., 2018), KL-DRO (Levy et al., 2020), and Randomized Smoothing (Cohen et al., 2019). Figure 3 demonstrates that our MWDRO method significantly enhances robustness compared to other methods.

**Further Evaluation on Distributional Shift Scenarios.** We conduct additional evaluations on two typical distributional shift scenarios, detailed in Appendix F. Specifically, we conduct assessments of our model's resilience against adversarial attacks using the CIFAR-10, CIFAR-100, and Tiny-ImageNet-200 datasets. Additionally, we evaluate the effectiveness of our proposed method on long-tailed benchmark datasets to test performance under naturally imbalanced data distributions.

**Other Experiments.** In Appendix F.5, we compare the running time with the standard WDRO (which does not consider the data manifold) and other adversarial training algorithms. While our algorithm requires more time, we achieve better performance on accuracy. We conduct other ablation studies in Appendix F.6 to analyze the effectiveness of the combination of the manifold-guided game and the MWDRO method.

## 6    CONCLUSION AND FUTURE WORK

This paper introduces a novel manifold-based WDRO method to enhance the robustness of image classification. We design a game that integrates contrastive learning with Jacobian regularization to capture the manifold structure, allowing us to solve DRO problems constrained by the data manifold. Experiments on several popular benchmark datasets demonstrate the method's advantages in terms of accuracy and robustness. Additionally, our manifold-based model could potentially serve as a framework for other scenarios that require access to the tangent of the data manifold, extending beyond the MWDRO problem. This is an interesting avenue for future research.

ACKNOWLEDGMENTS

The research of this work was supported in part by the National Key Research and Development Program of China (No. 2021YFA1000900), the National Natural Science Foundation of China (No. 62272432, No. 62432016), and the Natural Science Foundation of Anhui Province (No. 2208085MF163). H. Ding is the corresponding author.

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

# A  APPENDIX

## A.1  BACKGROUNDS

In this section, we provide some background on the manifold. For more detailed information, we refer readers to the comprehensive textbook by Boumal (2023).

Assumption 1 states that the manifold is a complete manifold embedded in a Euclidean space. This assumption ensures that the manifold's geometric properties are well-suited for optimization tasks. Roughly speaking, a complete manifold implies that every geodesic can be extended infinitely, preventing the optimization process from encountering boundaries or undefined regions that could disrupt convergence.

Moreover, we assume that the manifold $\mathcal{M}$ is embedded in a Euclidean space $\mathbb{R}^d$, meaning it inherits a well-defined metric from the ambient space $\mathbb{R}^d$. This allows us to define geodesic distances, perform projections, and compute gradients in a manner that is consistent with the underlying geometry

of the ambient Euclidean space. Such an embedded manifold also simplifies many computational tasks, such as the calculation of the Riemannian gradient. Specifically, the *Riemannian gradient* $\nabla^{\mathcal{M}}(f(\mathbf{x}))$ can be computed by the orthogonal projection of the "classical" gradient to the tangent spaces $\mathrm{T}_{\mathcal{M}}(\mathbf{x})$. *i.e.,* $\nabla^{\mathcal{M}} f(\mathbf{x}) = \mathrm{Proj}_{\mathbf{x}}(\nabla f(\mathbf{x}))$, where the $\mathrm{Proj}_{\mathbf{x}}(\cdot)$ is the *orthogonal projection operator* that projects a vector to the tangent space $\mathrm{T}_{\mathcal{M}}(\mathbf{x})$ and $\nabla(f(\mathbf{x}))$ is the gradient in Euclidean space $\mathbb{R}^d$.

For the mathematical rigor, we briefly verify that the prerequisites of Gao & Kleywegt (2023, Theorem 1) are satisfied within our framework, as defined in Eq.(5), by examining the properties of the manifold $\mathcal{M}$. By equipping $\mathcal{M}$ with the geodesic distance, we can verify that it is a Polish space (*i.e.,* a separable and complete metric space). Also, the space of Borel probability measures on $\mathcal{M}$ is well-defined. Additionally, the mild Assumption 4 in Appendix C ensures the continuity of our loss function $\ell(\theta, p)$. These properties allow us to safely apply Gao & Kleywegt (2023)'s strong duality results within our manifold-based WDRO framework, as formalized in Proposition 1.

## A.2 OTHER RELATED WORKS

**More DRO methods.** DRO demonstrate demonstrate significant advantages in terms of generalization and robustness across multiple domains (Gao et al., 2024; Huang et al., 2022; Wang et al., 2022). The works by (Staib & Jegelka, 2017; Sinha et al., 2018; Bui et al., 2022) are closely related to ours, as they also use the dual form of Wasserstein Distributionally Robust Optimization (DRO) to search for worst-case perturbations. However, these approaches cannot extend directly to manifold constraints; while they are effective against adversarial attacks, their overly pessimistic nature can decrease performance on clean data. Another line of works considers the Group DRO (Sagawa et al., 2020), which aims to optimize performance across different demographic subgroups, addressing hidden biases and ensuring fairness. $f$-divergence DRO (Namkoong & Duchi, 2016; Duchi & Namkoong, 2021), on the other hand, considers an uncertainty set within a certain $f$-divergence from the nominal distribution. Unlike Wasserstein distance, $f$-divergence measures the dissimilarity in terms of densities at corresponding points, and becomes infinite when two distributions have different support sets. This intrinsic difference allows conventional WDRO to extend the training support, while $f$-divergence cannot (Liu et al., 2022). Several studies attempt to incorporate geometric information into the uncertainty set. Qiao & Peng (2023) calculate group centrality based on the affinity between data groups, serving as a good nominal distribution for group DRO. Liu et al. (2022) employ a k-nearest neighbor graph structure to capture the geometric structure of the dataset, where shortest paths approximate geodesic distances to guide the transportation of probability mass. Their Geometric Wasserstein DRO framework constrains the distribution support set to observed data points. In contrast, our approach learns the data manifold structure through neural networks. This continuous, differentiable representation offers several advantages: it captures smooth geometric variations in the data manifold rather than relying on discrete graph approximations; furthermore, it naturally accommodates out-of-sample points, which enables more flexible adaptation to unseen data.

**Other related robustness methods.** Importance weighting (IW) has emerged as a powerful family of methods for addressing distribution shift in machine learning (Byrd & Lipton, 2019; Kimura & Hino, 2024). Several advancements have extended IW techniques for tackling various challenges. For instance, Shu et al. (2019) developed a meta-learning framework that learns an explicit weighting function to handle biased training data, such as label noise and class imbalance, by leveraging a small unbiased validation set. Fang et al. (2020) proposed an end-to-end framework that seamlessly integrates weight estimation and weighted classification through an iterative process. More recently, Fang et al. (2023) introduced generalized importance weighting (GIW), which extends traditional IW methods to handle various types of distribution shift, including challenging scenarios where the training and test distributions have different support sets. Jacobian regularization has been used to enhance robustness against small perturbations, such as noisy or adversarial data (Jakubovitz & Giryes, 2018; Hoffman et al., 2019). Jakubovitz & Giryes (2018) empirically demonstrated that it enhances robustness with minimal impact on the network's accuracy on clean data. Hoffman et al. (2019) proposed a computationally efficient implementation of Jacobian regularization. Furthermore, Rifai et al. (2011a;c) applied Jacobian regularization to autoencoders to learn representations that capture the local directions of variation dictated by the data, thereby improving the accuracy of classification tasks.

**Contrastive learning methods.** Contrastive learning is a powerful self-supervised learning paradigm that has significantly advanced representation learning (Wang & Isola, 2020; Xue et al., 2022; Chen et al., 2020; He et al., 2020). Khosla et al. (2020) introduced the supervised contrastive loss, achieving superior performance compared to cross-entropy on several tasks. However, the theoretical understanding of contrastive learning (CL) remains limited. Some works interpret the InfoNCE loss from a mutual information perspective (Bachman et al., 2019; Sordoni et al., 2021), showing it maximizes mutual information between different views of the data. Other studies (Wang & Isola, 2020; Huang et al., 2023) analyze the geometry of the embedding space, revealing that InfoNCE comprises two parts: one ensuring alignment and the other preventing representation collapse. Additionally, CL's feature embedding has been linked to spectral clustering (Balestriero & LeCun, 2022) and manifold learning (Hu et al., 2023).

## B MANIFOLD LEARNING PERSPECTIVE OF CONTRASTIVE LEARNING

In this section, we explore our proposed manifold-guided game through the lens of manifold learning and representation learning. By introducing important properties from contrastive learning, we analyze how our model effectively captures the structure of the data manifold, providing deeper insights into our approach. In Appendix B.1, we introduce several properties commonly discussed in the contrastive learning literature. These properties will aid in analyzing the effects of our manifold-guided approach, which we consider to be a complement to Section 4.2. Furthermore, in Appendix B.2, we show that the Jacobian regularization can enhance contrastive learning by preserving neighbour relationships within the input data, which benefits classification tasks. Building upon this observation, we refine our model to learn smoother manifolds, further improving its ability to capture the structure of the data manifold.

### B.1 PROPERTIES OF THE CONTRASTIVE LEARNING

A number of manifold learning methods, such as t-SNE (Van der Maaten & Hinton, 2008), UMAP (Damrich et al., 2023), and Isomap (Balestriero & LeCun, 2022), are based on the neighbour embedding techniques. These methods aim to preserve essential structural information from high-dimensional data in a low-dimensional embedding space. This is achieved by learning a mapping that encodes data points from high-dimensional space into a low-dimensional space, ensuring that data points that are neighbors in the original space remain closely situated in the embedding space. Specifically, given a set of points $\{\mathbf{x}_1, \cdots, \mathbf{x}_n\}$ and an encoder $\mathbf{g}(\cdot)$, we define the input/embedding affinity matrices $\mathbf{A}, \mathbf{B} \in \mathbb{R}^{n \times n}$, (i) input space affinity: $\mathbf{A}_{i,j}$ denote the affinity between the samples $\mathbf{x}_i$ and $\mathbf{x}_j$ in the original high-dimensional space; (ii) embedding affinity: $\mathbf{B}_{i,j}$ is the affinity computed in the embedding space between $\mathbf{g}(\mathbf{x}_i)$ and $\mathbf{g}(\mathbf{x}_j)$. Then the affinity-preserving entropy is given by $\mathcal{L}_{\text{affinity}}(\mathbf{g}) = -\sum_{i \neq j} \mathbf{A}_{i,j} \log(\mathbf{B}_{i,j})$, which quantifies how well the embedding preserves the pairwise affinities of the original data.

**Definition 3 (Neighbor-preserving property)** *Let $\mathbf{A}$ denote the input space affinity matrix based on a k-nearest neighbor graph $G$ constructed from the input data. Specifically, $\mathbf{A}_{i,j} = \mathbb{I}((i,j) \in G)/|G|$, where $\mathbb{I}$ is the indicator function and $|G|$ is the total number of edges in the graph. The embedding affinity matrix $\mathbf{B}$ is computed using the Cauchy kernel: $\mathbf{B}_{i,j} = \frac{1}{|\mathbf{g}(\mathbf{x}_i)-\mathbf{g}(\mathbf{x}_j)|_2^2+1}$. An encoder $\mathbf{g}(\cdot)$ is called neighbor-preserving if it minimizes the affinity-preserving entropy $\mathcal{L}_{\text{affinity}}(\mathbf{g})$.*

The affinity matrices described in Definition 3 are, in fact, the same as those used in the Stochastic Neighbor Embedding (SNE) (Hu et al., 2023) method. Similarly, contrastive learning can be interpreted as a special case that also minimizes the affinity-preserving entropy, but with different choices for the affinity matrices. We define these affinities as:

$$\mathbf{A}_{i,j} = \begin{cases} 1, & \text{if } \boldsymbol{x}_i \text{ and } \boldsymbol{x}_j \text{ are positive pairs} \\ 0, & \text{otherwise,} \end{cases} \tag{14}$$

$$\mathbf{B}_{i,j} = \frac{\exp\left(\text{sim}\left(f\left(\boldsymbol{x}_i\right), f\left(\boldsymbol{x}_j\right)\right)/\tau\right)}{\sum_{k \neq i} \exp\left(\text{sim}\left(f\left(\boldsymbol{x}_i\right), f\left(\boldsymbol{x}_k\right)\right)/\tau\right)}, \tag{15}$$

where $\mathrm{sim}(\cdot, \cdot)$ denotes a similarity measure (*e.g.,* cosine similarity), and $\tau$ is a temperature parameter. With these choices, the affinity-preserving entropy $\mathcal{L}_{\mathrm{affinity}}(\mathbf{g}) = -\sum_{i \neq j} \mathbf{A}_{i,j} \log(\mathbf{B}_{i,j})$ reduces to the InfoNCE loss in Eq.(9).

By the choice of the input/embedding affinity in Eq.(14)/(15), experimental and theoretical results in prior works (Wang & Isola, 2020; Hu et al., 2023) show that minimizing the InfoNCE loss leads to encoders that approximately achieve the properties of *perfect alignment* and *perfect uniformity* (see Definition 4). Specifically, the global minimizer of the contrastive loss requires that features of positive pairs align ($\mathrm{sim}(\mathbf{g}(\boldsymbol{x}_i'), \mathbf{g}(\boldsymbol{x}_i''))$ tends to 1) and those of negative pairs be as distant as possible.

**Definition 4 (Perfect Alignment/Uniformity (Wang & Isola, 2020))** *(i) Perfect Alignment. An encoder $\mathbf{g}(\cdot)$ achieves perfectly alignment over dataset $P_{\mathrm{tr}}$, if $\mathbf{g}(\boldsymbol{x}_i') = \mathbf{g}(\boldsymbol{x}_i'')$ almost surely for all $\boldsymbol{x}_i', \boldsymbol{x}_i'' \in \mathrm{aug}(\boldsymbol{x}_i), \forall \boldsymbol{x}_i \in P_{\mathrm{tr}}$; (ii) Perfect Uniformity. An encoder $\mathbf{g}(\cdot)$ achieves perfect uniformity over dataset $P_{\mathrm{tr}}$ if the embeddings $\mathbf{g}(\boldsymbol{x}_i), \forall \boldsymbol{x}_i \in P_{\mathrm{tr}}$ are maximally separated as $|P_{\mathrm{tr}}|$ points on the sphere $\mathcal{S}^{d_e-1} \in \mathbb{R}^{d_e}$, where the $d_e$ is the dimension of the embedding space.*

We now re-examine our proposed manifold-guided game from the perspectives of alignment and uniformity. The repulsive force between negative pairs (*i.e.,* pushing apart $\mathbf{g}(\mathbf{x}_i')$ and $\mathbf{g}(\mathbf{x})$ in the InfoNCE loss Eq.(9)) helps preserve maximal information of the data manifold. Simultaneously, the attractive force between the positive pairs (*i.e.,* pulling together $\mathbf{g}(\mathbf{x}_i')$ and $\mathbf{g}(\mathbf{x}_i'')$), combined with Jacobian regularization, minimizes sensitivity orthogonal to the manifold or directed towards the semantic-invariant region $\mathcal{M}_{SI}(\mathbf{x})$.

Based on this analysis, we summarize the desired properties of the encoder $\mathbf{g}(\cdot)$ as follows: **(I) Sensitivity.** The encoder $\mathbf{g}(\boldsymbol{x})$ should change most when the sample $\boldsymbol{x}$ moves along the semantic variant directions within the data manifold; **(II) Insensitivity.** The encoder $\mathbf{g}(\boldsymbol{x})$ should change minimally when moves off the manifold (*i.e.,* $\mathcal{M}^{\perp}$), or towards the Semantic-Invariant region (*i.e.,* $\mathcal{M}_{SI}(\mathbf{x})$).

### B.2 TOWARDS SMOOTHER MANIFOLD AND BETTER REPRESENTATION

In this section, we demonstrate that the Jacobian regularization can enhance contrastive learning by better preserving neighbour relationships within the input data. We have known that the alignment and uniformity properties (see Definition 4) enable the game between CL and Jacobian regularization to shape representations that capture the tangents of the data manifold. However, considering only the contrastive loss may not adequately preserve the neighbor structure. This is because any permutation of the mapping $\boldsymbol{x} \rightarrow \mathbf{g}(\boldsymbol{x})$ for $\boldsymbol{x} \in P_{\mathrm{tr}}$ can achieve perfect alignment and uniformity, thus reaching the global minima of Eq.(9). Figure (4) illustrates this issue with two encoders, $\mathbf{g}_1(\cdot)$ and its permutation $\mathbf{g}_2(\cdot)$. Both achieve perfect alignment and uniformity, but $\mathbf{g}_2(\cdot)$ preserves the neighbor relationship of the input data, clustering similar samples (*e.g.,* birds) in the embedding space. This clustering is beneficial for downstream classification tasks due to the linear separability of the features (Wang & Isola, 2020) (as shown in Figure (4b), we can classify the birds in the embedding space by a linear classifier).

To analyze the quality of neighbor structure preserving, Hu et al. (2023) propose a complexity measure defined as:

$$C(\mathbf{g}) = \mathbb{E}_{\boldsymbol{x}, \boldsymbol{x}'} \left[ \frac{\|\mathbf{g}(\boldsymbol{x}) - \mathbf{g}(\boldsymbol{x}')\|_2}{\|\boldsymbol{x} - \boldsymbol{x}'\|_2} \right], \tag{16}$$

where $\boldsymbol{x}, \boldsymbol{x}'$ are negative pairs as in Eq.(9). This complexity measure captures the degree to which the mapping $\mathbf{g}(\cdot)$ preserves the neighbor structure of the input data in the context of contrastive learning. For example, in Figure (4), $C(\mathbf{g}_2) < C(\mathbf{g}_1)$, which implies $\mathbf{g}_2$ preserves the neighbor structure better than $\mathbf{g}_1$. Building on their findings, the Jacobian regularization can be considered a differential version of this complexity measure, $C(\mathbf{g})$. Our empirical results confirm that Jacobian regularization reduces $C(\mathbf{g})$, thereby enhancing the neighborhood-preserving properties of the model.

However, the Jacobian provides only a local approximation to $C(\mathbf{g})$ as defined in Eq.(16). Intuitively, enforcing smoothness on the Jacobian $J_{\mathbf{g}}(\mathbf{x})$ improves neighborhood preservation by ensuring that the encoder $\mathbf{g}(\cdot)$ changes gradually without abrupt variations. Specifically, this smoothness

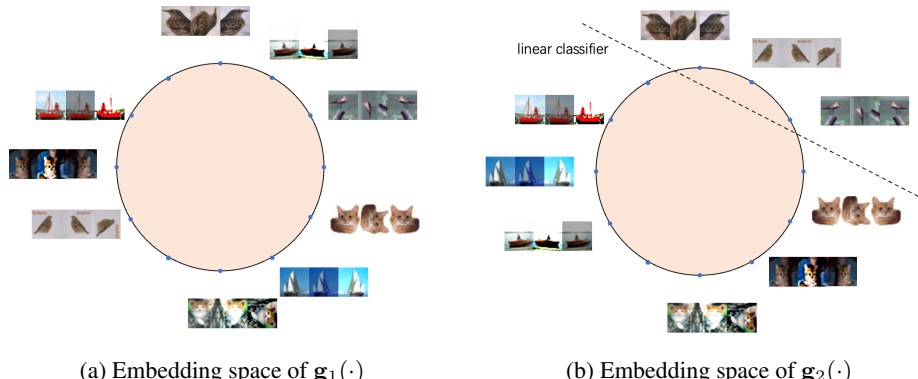

(a) Embedding space of $\mathbf{g}_1(\cdot)$                 (b) Embedding space of $\mathbf{g}_2(\cdot)$

Figure 4: Both encoders $\mathbf{g}_1(\cdot)$ and $\mathbf{g}_2(\cdot)$ achieve perfect alignment and uniformity, reaching the empirical minima of the loss Eq.(9). The encoder $\mathbf{g}_1(\cdot)$ in Figure (4a) is a permutation of the encoder mapping $\mathbf{g}_2(\cdot)$ in Figure (4b). $\mathbf{g}_2(\cdot)$ preserves the affinity of the input dataset $P_{\mathrm{tr}}$, with similar samples (birds, cats and ships) well clustered in the embedding space. We prefer $\mathbf{g}_2(\cdot)$ as the clustered features are beneficial for downstream classification tasks due to their linear separability (Wang & Isola, 2020).

is beneficial for accurately capturing the directions of semantic variation within the data manifold. As is well known, second-order derivatives (*i.e.,* the Hessian), measure the smoothness of the Jacobian (Rifai et al., 2011b; Zhao & Zhang, 2022). However, computing the Hessian matrix is computationally expensive. To address this, we can use the mixup data augmentation technique (Zhang et al., 2018) to enhance our manifold-guided game, as mentioned in Remark 2. Specifically, we create new samples by mixing a sample $\mathbf{x}_i \in P_{tr}$ with one of its neighbor $\mathbf{x}_j$, forming $\lambda\mathbf{x}_i + (1-\lambda)\mathbf{x}_j$, and incorporate these new samples into the training process. This mixup technique helps stabilize the Jacobian when $\mathbf{x}$ moves along the data manifold, favoring a smoother representation.

We construct the mixed dataset using the mixup (Zhang et al., 2018) technique and denote it as $P_{\mathrm{tr}}^{\mathrm{mix}} := \left\{ (\boldsymbol{x}_i', \boldsymbol{x}_j'', \lambda) | \boldsymbol{x}_i' \in P_{\mathrm{tr}}', \boldsymbol{x}_j'' \in P_{\mathrm{tr}}'', \mathbf{A}_{i,j} \neq 0, \lambda \sim \mathrm{Unif}(0,1) \right\}$, where $\mathbf{A}$ is the input affinity matrix in Definition 3 and $\mathrm{Unif}(0,1)$ is the uniform distribution over the interval $(0,1)$.

The modified InfoNCE loss with mixup is defined as:

$$\mathcal{L}_{\mathrm{InfoNCE}}^{\mathrm{mix up}} = \mathcal{L}_{\mathrm{InfoNCE}} - \mathop{\mathbb{E}}_{P_{\mathrm{tr}}^{\mathrm{mix}}} \left[ \log \frac{e^{\mathrm{sim}\left(\lambda\mathbf{g}(\boldsymbol{x}_i') + (1-\lambda)\mathbf{g}(\boldsymbol{x}_j''), \mathbf{g}\left(\lambda\boldsymbol{x}_i' + (1-\lambda)\boldsymbol{x}_j''\right)\right)/\tau}}{\sum_{\boldsymbol{x} \in P_{\mathrm{tr}}' \cup P_{\mathrm{tr}}'' \setminus \left\{\boldsymbol{x}_i', \boldsymbol{x}_i''\right\}} e^{\mathrm{sim}(\mathbf{g}(\boldsymbol{x}_i'), \mathbf{g}(\boldsymbol{x}))/\tau}} \right]. \tag{17}$$

where the notations follow Eq.(9). Then the complete objective function becomes:

$$\mathcal{L}^*(\theta) = \mathcal{L}_{\mathrm{InfoNCE}}^{\mathrm{mix up}} + \lambda_1 \mathop{\mathbb{E}}_{\boldsymbol{x} \in P_{\mathrm{tr}}' \cup P_{\mathrm{tr}}'' \cup P_{\mathrm{tr}}^{\mathrm{mix}}} \|J_{\mathbf{g}}(\mathbf{x})\|_F^2, \tag{18}$$

The mixup-enhanced loss encourages the model to produce consistent embeddings for interpolated inputs, helping to smooth the representation and stabilize the Jacobian without the costly Hessian regularization.

We compute the $C(\mathbf{g})$, as defined in Eq.(16), for well-trained models on the CIFAR-10 and CIFAR-100 datasets. The results, presented in Table 1, show that Jacobian regularization, particularly when combined with the mixup technique, significantly enhances the neighborhoods preservation in feature embeddings. By stabilizing the Jacobian $J_{\mathbf{g}}$ as $\mathbf{x}$ traverses the data manifold, our model encodes the data manifold more smoothly. This yields two key advantages: first, it facilitates better extraction of tangents on the data manifold; second, the clustered embeddings are beneficial for downstream classification tasks, as illustrated in Figure 4.

Table 1: We compute $C(\mathbf{g})$ (as defined in Eq.(16)) for various well-trained models on CIFAR-10 and CIFAR-100 datasets. The methods compared include: Contrastive Learning (CL, Eq.(9)), Contrastive Learning with Jacobian regularization (CL+Jac, Eq.(8)), and Contrastive Learning with both Jacobian regularization and mixup (CL+Jac+mixup, Eq.(17)). We normalize all the results by dividing them over the results of CL. Lower values indicate better neighbour preservation.

| Objectives | CIFAR-10 | CIFAR-100 |
|---|---|---|
| CL | 1.00 | 1.00 |
| CL+Jac | 0.29 | 0.40 |
| CL+Jac+mixup | 0.10 | 0.11 |

## C  APPROXIMATION OF THE RIEMANNIAN GRADIENT

As discussed in Section 4.1, we utilize the strong duality formulation of the MWDRO problem presented in Proposition 1 as our objective function. The main challenge in optimizing this function arises from the surrogate loss defined in Eq.(7). To compute this surrogate loss $\ell_s(\theta, p_i, \nu) := \sup_{q \in \mathcal{M}} [\ell(\theta, q) - \nu \mathbf{d}_{\mathbf{g}}^2(q, p_i)]$, we use the first-order optimization method (Zhang & Sra, 2016). The update rule is shown in Eq.(13). For clarity, we rewrite it here:

$$q^{t+1} = \mathrm{Exp}_{q^t} \left( \alpha \nabla^{\mathcal{M}} \left( \ell(\theta, q^t) - \nu \mathbf{d}_{\mathbf{g}}^2(q^t, q) \right) \right)$$
$$= \mathrm{Exp}_{q^t} \left( \alpha \mathrm{Proj}_{q^t} \left( \nabla \ell(\theta, q^t) \right) - 2\alpha\nu \mathbf{d}_{\mathbf{g}}(q^t, q) \hat{u}_{q^t, q} \right)$$

The above updating steps require the Riemannian gradient, which is the orthogonal projection of the "classical" gradient to the tangent spaces (Boumal, 2023), *i.e.* $\nabla^{\mathcal{M}} f(q^t) = \mathrm{Proj}_{q^t} \left( \nabla f(q^t) \right)$. The central problem, then, is how to recover the tangent space $\mathrm{T}_{\mathcal{M}}(q^t)$ so as to approximate the Riemannian gradient.

In Section 4.2, we design the manifold-guided game for training the feature encoder $\mathbf{g}(\cdot)$. We analyze the singular value decomposition (SVD) of the Jacobian matrix $J_{\mathbf{g}}(\mathbf{x}) = \mathbf{U}(\mathbf{x})\mathbf{\Sigma}(\mathbf{x})\mathbf{V}^\top(\mathbf{x})$, where $\mathbf{U}(\mathbf{x}) \in \mathbb{R}^{d_e \times r}$, $\mathbf{V}^\top(\mathbf{x}) = [\mathbf{v}_1(\mathbf{x}), \cdots, \mathbf{v}_r(\mathbf{x})]^\top \in \mathbb{R}^{r \times d}$, and $\mathbf{\Sigma}(\mathbf{x}) = \mathrm{diag}(\sigma_1(\mathbf{x}), \cdots, \sigma_r(\mathbf{x}))$. We define the space spanned by the principal singular vectors of the Jacobian matrix in Eq.(10), *i.e.*, $\mathrm{T}_{\mathrm{appr}}(\mathbf{x}, \tau_0) := \mathrm{span}\{\mathbf{v}_i(\mathbf{x}) \mid \sigma_i(\mathbf{x}) \geq \tau_0, i \in [r]\}$.

As discussed in our main paper, we aim to approximate the Riemannian gradient $\nabla^{\mathcal{M}} \ell(\theta, q^t)$ by projecting the Euclidean gradient $\nabla \ell(\theta, q^t)$ onto the subspace $\mathrm{T}_{\mathrm{appr}}(q^t, \tau_0)$. We denote this operation as $\mathrm{Proj}_{\mathrm{appr}}^{\tau_0}(\nabla \ell(\theta, q^t))$. In what follows, we analyze the rationale behind this approximation. We begin with assumptions we require.

The key idea is utilizing $\mathrm{T}_{\mathrm{appr}}(\mathbf{x}, \tau_0)$ to approximate a subset of the tangent $\mathrm{T}_{\mathcal{M}}(\mathbf{x})$, capturing directions of significant semantic variation within $\mathrm{T}_{\mathcal{M}}(\mathbf{x})$. We formalize this as Assumption 3.

**Assumption 3** *For some $\hat{\tau} \geq 0$, $\mathrm{T}_{\mathrm{appr}}(\mathbf{x}, \hat{\tau})$ is a subspace of the tangent plane $\mathrm{T}_{\mathcal{M}}(\mathbf{x})$ for any $\mathbf{x} \in \mathcal{M}$.*

Now, let us examine the loss function $\ell(\theta, \mathbf{x})$. We can view it as a composition of functions involving $\mathbf{g}(\mathbf{x})$. The loss $\ell(\cdot)$ is usually continuous w.r.t. $\mathbf{g}(\mathbf{x})$, as detailed in Assumption 4.

**Assumption 4** *The loss function $\ell(\theta, \mathbf{x})$ is differentiable with respect to $\mathbf{g}(\mathbf{x})$. Moreover, $\ell(\theta, \mathbf{x})$ is $L$-Lipschitz with respect to $\mathbf{g}(\mathbf{x})$. This means that for any two inputs $\mathbf{x}_1$ and $\mathbf{x}_2$, the inequality $|\ell(\theta, \mathbf{x}_1) - \ell(\theta, \mathbf{x}_2)| \leq L\|\mathbf{g}(\mathbf{x}_1) - \mathbf{g}(\mathbf{x}_2)\|$ holds for any parameter $\theta$.*

**Remark 4** *In classification tasks, a common choice is $\ell(\theta, \mathbf{x}) = \mathrm{CE}\left(\mathrm{Softmax}\left(\mathbf{g}(\mathbf{x})\right)\right)$, where $\mathrm{CE}(\cdot)$ is the cross-entropy loss. It is well-known that $\nabla_{\mathbf{g}(\mathbf{x})} \ell(\theta, \mathbf{x}) = \mathrm{Softmax}(\mathbf{g}(\mathbf{x})) - y$, where $y$ is the one-hot label corresponding to $\mathbf{x}$. Therefore, $\|\nabla_{\mathbf{g}(\mathbf{x})} \ell(\theta, \mathbf{x})\| \leq 2$, implying that the cross-entropy loss is 2-Lipschitz with respect to $\mathbf{g}(\mathbf{x})$.*

**Theorem 2 (Approximation of Riemannian Gradient)** *Under Assumptions 3 and 4, for any $\hat{\tau} \leq \tau_0 \leq \max_i \sigma_i$, $\mathrm{Proj}_{\mathrm{appr}}^{\tau_0}(\nabla \ell(\theta, q^t))$ approximates the Riemannian gradient $\nabla^{\mathcal{M}} \ell(\theta, q^t)$ with the following error bound: $\left\|\nabla^{\mathcal{M}} \ell(\theta, q^t) - \mathrm{Proj}_{\mathrm{appr}}^{\tau_0}(\nabla \ell(\theta, q^t))\right\|_2 \leq L \cdot \tau_0$.*

*Proof.* By the chain rule, we can express the classical gradient as:

$$\nabla \ell(\theta, q^t) = \frac{\partial \ell}{\partial \mathbf{g}} \mathbf{J_g}(q^t) = \frac{\partial \ell}{\partial \mathbf{g}} \mathbf{U} \mathbf{\Sigma} \mathbf{V}^\top \tag{19}$$

Let $\mathbf{w} = [w_1, \cdots, w_r] = \frac{\partial \ell}{\partial \mathbf{g}} \mathbf{U}$. Then the gradient can be expressed as: $\nabla \ell(\theta, q^t) = \mathbf{w} \mathbf{\Sigma} \mathbf{V}^\top = \sum_{i=1}^r w_i \sigma_i \mathbf{v}_i$. The approximate projection operator with threshold $\tau_0$ is defined as:

$$\mathrm{Proj}_{\mathrm{appr}}^{\tau_0}(\nabla \ell(\theta, q^t)) = \sum_{i:\sigma_i \geq \tau_0} w_i \sigma_i \mathbf{v}_i \tag{20}$$

The Riemannian gradient is obtained by projecting the classical gradient onto the tangent space at $q^t$. This projection can be decomposed as follows:

$$\nabla^{\mathcal{M}} \ell(\theta, q^t) = \mathrm{Proj}_{q^t}\Big( \sum_{i=1}^r w_i \sigma_i \mathbf{v}_i \Big) = \sum_{i=1}^r \mathrm{Proj}_{q^t}\Big( w_i \sigma_i \mathbf{v}_i \Big)$$
$$= \sum_{i:\sigma_i \geq \hat{\tau}} w_i \sigma_i \mathbf{v}_i + \sum_{i:\sigma_i < \hat{\tau}} \mathrm{Proj}_{q^t}\Big( w_i \sigma_i \mathbf{v}_i \Big) \tag{21}$$

Eq.(20) follows directly from the definition of the approximate tangent space $\mathrm{T}_{\mathrm{appr}}(\mathbf{x}, \hat{\tau})$.

By Assumption 3, we have $\mathrm{T}_{\mathrm{appr}}(\mathbf{x}, \hat{\tau}) \subseteq \mathrm{T}_{\mathcal{M}}(\mathbf{x})$, which together with the definition of $\mathrm{T}_{\mathrm{appr}}(\mathbf{x}, \hat{\tau})$ establishes Eq.(21).

The difference is:

$$\nabla^{\mathcal{M}} \ell(\theta, q^t) - \mathrm{Proj}_{\mathrm{appr}}^{\tau_0}(\nabla \ell(\theta, q^t)) = \sum_{i:\hat{\tau} \leq \sigma_i < \tau_0} w_i \sigma_i \mathbf{v}_i + \sum_{i:\sigma_i < \hat{\tau}} \mathrm{Proj}_{q^t}\Big( w_i \sigma_i \mathbf{v}_i \Big) \tag{22}$$

Taking the norm and using the orthogonality of $\mathbf{v}_i$:

$$\|\nabla^{\mathcal{M}} \ell(\theta, q^t) - \mathrm{Proj}_{\mathrm{appr}}^{\tau_0}(\nabla \ell(\theta, q^t))\|_2^2 \leq \sum_{i:\sigma_i < \tau_0} |w_i|^2 \sigma_i^2 \tag{23}$$

By Assumption 4, we have $\|\frac{\partial \ell}{\partial \mathbf{g}}\|_2 \leq L$. Since $\mathbf{U}$ is an orthonormal matrix, $\|\mathbf{U}\|_2 = 1$, thus $\left\|\frac{\partial \ell}{\partial \mathbf{g}} \mathbf{U}\right\|_2 \leq L$. Therefore we have

$$\sum_{i:\sigma_i < \tau_0} |w_i|^2 \sigma_i^2 < \tau_0^2 \sum_{i:\sigma_i < \tau_0} |w_i|^2 < \tau_0^2 \sum_{i=1}^r |w_i|^2 \leq L^2 \tau_0^2 \tag{24}$$

Taking the square root:

$$\|\nabla^{\mathcal{M}} \ell(\theta, q^t) - \mathrm{Proj}_{\mathrm{appr}}^{\tau_0}(\nabla \ell(\theta, q^t))\|_2 \leq L \cdot \tau_0 \tag{25}$$

$\square$

This theorem provides a justification for approximating the Riemannian gradient by projecting onto $\mathrm{T}_{\mathrm{appr}}(\mathbf{x}, \tau_0)$.

## D THEORETICAL ASPECTS OF THE DISTRIBUTIONAL ROBUST

### D.1 JACOBIAN TERM AND THE SMOOTHNESS ASSUMPTION

In this section, we demonstrate that the Jacobian regularization term serves as an upper bound for $\beta$, which characterizes the smoothness of our model as Assumption 2. Since the data manifold $\mathcal{M}$ is embedded in the ambient Euclidean space $\mathbb{R}^d$, without loss of generality, we focus on discussing smoothness within $\mathbb{R}^d$.

We derive the second derivative of $f(\mathbf{g}(\mathbf{x}))$ with respect to the input sample $\mathbf{x}$. The smallest value $\beta$ can take is the largest eigenvalue of the Hessian at any point $\mathbf{x} \in \mathcal{X}$. In our setting, $\beta = \sup_{\mathbf{x} \in \mathcal{X}} \lambda_{\max} \frac{\partial^2 \ell}{\partial \mathbf{x}^2}$.

$$
\begin{aligned}
\frac{\partial^2 \ell}{\partial \mathbf{x}^2} &= \frac{\partial}{\partial \mathbf{x}} \left( \frac{\partial \ell}{\partial \mathbf{x}} \right) = \frac{\partial}{\partial \mathbf{x}} \left( \frac{\partial \ell}{\partial \mathbf{g}} \cdot \mathbf{J_g} \right) = \frac{\partial}{\partial \mathbf{x}} \left( \frac{\partial \ell}{\partial \mathbf{g}} \right) \cdot \mathbf{J_g} + \frac{\partial \ell}{\partial \mathbf{g}} \cdot \frac{\partial \mathbf{J_g}}{\partial \mathbf{x}} \\
&= \mathbf{J_g}^\top \mathbf{H}_\ell \mathbf{J_g} + \sum_{i=1}^m \frac{\partial \ell}{\partial \mathbf{g}_i} \cdot \frac{\partial^2 \mathbf{g}_i}{\partial \mathbf{x}^2} \\
&= \mathbf{J_g}^\top \mathbf{H}_\ell \mathbf{J_g} + \frac{\partial \ell}{\partial \mathbf{g}} \cdot \mathbf{H_g}
\end{aligned}
\tag{26}
$$

The notations used are:

– $\mathbf{J_g}$: Jacobian of $\mathbf{g}$, dimension $d_e \times d$.
– $\mathbf{H}_\ell$: Hessian matrix of the loss $\ell(\cdot)$ with respect to $\mathbf{g}(\cdot)$, dimension $d_e \times d_e$.
– $\mathbf{H_g} = \frac{\partial \mathbf{J_g}}{\partial \mathbf{x}} = \frac{\partial^2 \mathbf{g}}{\partial \mathbf{x}^2}(\mathbf{x})$: Hessian tensor of $\mathbf{g}(\cdot)$, dimension $d_e \times d \times d$.
– $\| \cdot \|_2$: spectral norm of a matrix, *i.e.,* the largest singular value.

Using the sub-multiplicative property of the norm $\| \cdot \|_2$, we have

$$
\lambda_{\max}(\mathbf{J_g}^\top \mathbf{H}_\ell \mathbf{J_g}) \leq \| \mathbf{J_g}^\top \|_2 \cdot \| \mathbf{H}_\ell \|_2 \cdot \| \mathbf{J_g} \|_2
\tag{27}
$$

$$
\lambda_{\max} \left( \frac{\partial \ell}{\partial \mathbf{g}} \cdot \mathbf{H_g} \right) \leq \frac{\partial \ell}{\partial \mathbf{g}} \cdot \left[ \| \mathbf{H_g}(i, :, :) \|_2 \right]_i^\top
\tag{28}
$$

Assuming the second partial derivatives of $\mathbf{g}$ and $\ell$ are continuous, then $\mathbf{H}_\ell$ and $\mathbf{H_g}(i, :, :)$ for all $i \in [d_e]$ are symmetric. Further, it is not hard to verify that $\mathbf{J_g}^\top \mathbf{H}_\ell \mathbf{J_g}$ and $\frac{\partial \ell}{\partial \mathbf{g}} \cdot \mathbf{H_g}$ are also symmetric. Applying Weyl's inequality for eigenvalues, we get:

$$
\begin{aligned}
\lambda_{\max}(\frac{\partial^2 \ell}{\partial \mathbf{x}^2}) &\leq \lambda_{\max}(\mathbf{J_g}^\top \mathbf{H}_\ell \mathbf{J_g}) + \lambda_{\max} \left( \frac{\partial \ell}{\partial \mathbf{g}} \cdot \mathbf{H_g} \right) \\
&= \| \mathbf{H}_\ell \|_2 \cdot \| \mathbf{J_g} \|_2^2 + \frac{\partial \ell}{\partial \mathbf{g}} \cdot \left[ \| \mathbf{H_g}(i, :, :) \|_2 \right]_i^\top
\end{aligned}
\tag{29}
$$

Note that $\| \mathbf{J_g} \|_2 \leq \| \mathbf{J_g} \|_F$. For the Hessian term, following the approximation from (Jakubovitz & Giryes, 2018; Martens et al., 2012), we have $\| \mathbf{H_g}(i, :, :) \|_2 \approx \| \mathbf{J_g}(i, :)^\top \mathbf{J_g}(i, :) \|_2 = \| \mathbf{J_g}(i, :) \|_2^2$. Thus

$$
\frac{\partial \ell}{\partial \mathbf{g}} \cdot \left[ \| \mathbf{H_g}(i, :, :) \|_2 \right]_i^\top \leq \frac{\partial \ell}{\partial \mathbf{g}} \cdot \left[ \| \mathbf{J_g}(i, :) \|_2^2 \right]_i^\top = \sum_{i=1}^{d_e} \frac{\partial \ell}{\partial \mathbf{g}_i} \| \mathbf{J_g}(i, :) \|_2^2 \leq \| \frac{\partial \ell}{\partial \mathbf{g}} \|_\infty \| \mathbf{J_g} \|_F^2
\tag{30}
$$

Finally we have

$$
\lambda_{\max}(\frac{\partial^2 \ell}{\partial \mathbf{x}^2}) \leq (\| \mathbf{H}_\ell \|_2 + \| \frac{\partial \ell}{\partial \mathbf{g}} \|_\infty) \| \mathbf{J_g} \|_F^2
\tag{31}
$$

Consequently, we can conclude that the Jacobian regularization term $\| J_\mathbf{g}(\mathbf{x}) \|_F^2$ provides an upper bound for $\beta$.

## D.2 PROOF OF LEMMA 1

We can verify that the following proposition holds according to Definition 2. Please refer to (Boumal, 2023) for more detailed information.

**Proposition 2** *Under Assumption 2, for a given $p \in \mathcal{M}$ and a constant $\nu > \beta$, the function $\ell(\theta, p^t) - \nu d_{\mathbf{g}}^2(p^t, p)$ is geodesically $(\nu - \beta)$-strongly concave and geodesically $(\nu + \beta)$-smooth.*

*Proof.*[of Lemma 1] Algorithm 2 solves the surrogate loss as defined in Eq.(7). For brevity, we denote $\ell(\theta, q)$ as $\ell(q)$. The surrogate loss is given by:

$$\ell_s(q^0, \nu) := \sup_{q \in \mathcal{M}} \left[ \ell(q) - \nu d_{\mathbf{g}}^2(q, q^0) \right] \tag{32}$$

It can be shown that the function $\ell(q) - \nu d_{\mathbf{g}}^2(q, q^0)$ is geodesically $(\nu - \beta)$-strongly concave and geodesically $(\nu + \beta)$-smooth ($\nu > \beta > 0$). Thus its condition number, defined as the ratio of the smoothness parameter to the strong concavity parameter, satisfies $\kappa = \frac{\nu + \beta}{\nu - \beta}$.

We introduce a sequence of auxiliary functions for the $t$-th iteration of Algorithm 2 as follows,

$$\iota_t(q) = \ell(q) - \nu_t \cdot d_{\mathbf{g}}^2(q^0, q), \tag{33}$$

where $\nu_t$ is the value computed as $\nu \cdot \frac{(\mathtt{p} t^t)^2}{d_{\mathbf{g}}^2(q^0, q^t)}$ (with $\nu_0 = \nu$). It is not hard to note that $\nu_t \geq \nu$ for all $t \geq 0$. Therefore, the function $\iota_t(q)$ is geodesically $(\nu_t - \beta)$-strongly concave and geodesically $(\nu_t + \beta)$-smooth, with $\nu_t > \nu > \beta > 0$. As a result, its condition number satisfies $\kappa_t = \frac{\nu_t + \beta}{\nu_t - \beta} \leq \frac{\nu + \beta}{\nu - \beta} = \kappa$.

The gradient at $q^t$, computed in step 2 of Algorithm 2, is $\nabla^{\mathcal{M}} \iota_t(q^t) = \mathrm{Proj}_{q^t} \left( \nabla_q l(\theta, q^t) \right) - 2\nu \mathtt{p} t^t \cdot \hat{u}_{q^t, q}$, where $\hat{u}_{q^t, q}$ denotes the unit vector representing the initial direction of the geodesic from point $q^t$ to point $q$ on the manifold. Let $q^{t,*}$ denote the optimal solution of maximizing $\iota_t(q)$. According to Algorithm 2, the update step is given by: $q^{t+1} = \mathrm{Exp}_{q^t}(\alpha \cdot \nabla^{\mathcal{M}} \iota_t(q^t))$, where $\alpha \in [0, \frac{2}{\nu + \beta})$ is the step size. We select $\alpha = \frac{1}{\nu + \beta}$. Given that $\iota_t$ is $(\nu_t + \beta)$-smooth, we have:

$$\iota_t(q^{t,*}) - \iota_t(q^{t+1}) \leq \iota_t(q^{t,*}) - \iota_t(q^t) - \frac{1}{2(\nu_t + \beta)} \|\nabla^{\mathcal{M}} \iota_t(q^t)\|^2 \tag{34}$$

Furthermore, by (Boumal, 2023, Lemma 11.28), we obtain: $\|\nabla^{\mathcal{M}} \iota_t(q^t)\|^2 \geq 2(\nu_t - \beta)(\iota_t(q^{t,*}) - \iota_t(q^t))$, substituting this into Eq.(34) yields:

$$\iota_t(q^{t,*}) - \iota_t(q^{t+1}) \leq (1 - \frac{1}{\kappa_t})(\iota_t(q^{t,*}) - \iota_t(q^t)) \leq (1 - \frac{1}{\kappa})(\iota_{t-1}(q^{t-1,*}) - \iota_{t-1}(q^t)) \tag{35}$$

By recursively applying this inequality, we obtain linear convergence:

$$\iota_t(q^{t,*}) - \iota_t(q^{t+1}) \leq (1 - \frac{1}{\kappa})^t(\iota_0(q^{0,*}) - \iota_0(q^1)) \tag{36}$$

Utilizing the geodesically $(\nu - \beta)$-strong concavity, we have:

$$\begin{aligned}
d_{\mathbf{g}}(q^{t+1}, q^{t,*})^2 &\leq \frac{2}{\nu - \beta}(\iota_t(q^{t,*}) - \iota_t(q^{t+1})) \\
&\leq \frac{2}{\nu - \beta}(1 - \frac{1}{\kappa})^t(\iota_0(q^{0,*}) - \iota_0(q^1)) \\
&\leq \frac{2}{\nu - \beta}(1 - \frac{1}{\kappa})^{t+1}(\iota_0(q^{0,*}) - \iota_0(q^0))
\end{aligned} \tag{37}$$

By the $\beta + \nu$ smoothness, we have $\iota_0(q^{0,*}) \leq \iota_0(q^0) + \frac{\beta + \nu}{2} d_{\mathbf{g}}(q^0, q^{0,*})^2$. Then

$$d_{\mathbf{g}}(q^{t+1}, q^{t,*}) \leq \sqrt{\kappa} \sqrt{1 - \frac{1}{\kappa}}^{t+1} d_{\mathbf{g}}(q^0, q^{0,*}). \tag{38}$$

Let $c = \sqrt{1 - \frac{1}{\kappa}} \in (0, 1)$. Then, we can rewrite as: $d_{\mathbf{g}}(q^i, q^*) \leq c^i \sqrt{\kappa} d_{\mathbf{g}}(q^0, q^*)$.

By triangle inequality, for any $i \geq 0$, we have:

$$\begin{aligned}
d_{\mathbf{g}}(q^i, q^{i+1}) &\leq d_{\mathbf{g}}(q^i, q^{i,*}) + d_{\mathbf{g}}(q^{i+1}, q^{i,*}) \\
&\leq c^i \sqrt{\kappa} d_{\mathbf{g}}(q^0, q^*) + c^{i+1} \sqrt{\kappa} d_{\mathbf{g}}(q^0, q^*),
\end{aligned}$$

which simplifies to:

$$\mathbf{d_g}(q^i, q^{i+1}) \le c^i(1+c)\sqrt{\kappa}\mathbf{d_g}(q^0, q^*). \tag{39}$$

Summing over all steps along the trajectory, we obtain:

$$\sum_{i=0}^{t-1}\mathbf{d_g}(q^i, q^{i+1}) \le (1+c)\sqrt{\kappa}\sum_{i=0}^{t-1}c^i\mathbf{d_g}(q^0, q^*) = \frac{(1+c)(1-c^t)}{1-c}\sqrt{\kappa}\mathbf{d_g}(q^0, q^*). \tag{40}$$

Taking $t \to \infty$, we have:

$$\mathtt{pt}^\infty = \sum_{i=0}^{\infty}\mathbf{d_g}(q^i, q^{i+1}) \le \frac{1+c}{1-c}\sqrt{\kappa}\mathbf{d_g}(q^0, q^*).$$

Since $\frac{1+c}{1-c}\sqrt{\kappa} = \sqrt{\kappa}(\sqrt{\kappa} + \sqrt{\kappa-1})^2$, finally we have:

$$\mathtt{pt}^t \le \sqrt{\kappa}(\sqrt{\kappa} + \sqrt{\kappa-1})^2\mathbf{d_g}(q^0, q^*), \forall t \ge 0 \tag{41}$$

$\square$

### D.3 PROOF OF THEOREM 1

As mentioned in our main paper, we do not explicitly set the radius $\delta$ for the uncertainty set $\mathcal{U}_{gw}(P_{\mathrm{tr}}, \delta)$ in Eq.(2). Instead, we select a $\nu > 0$ to solve the empirical surrogate loss $\mathbb{E}_{P_{\mathrm{tr}}}[\ell_s(\theta, p_i, \nu)]$ rather than prescribing the robust range $\delta$; this approach, originally proposed by Sinha et al. (2018), has been shown to ensure certified distributional robustness. Then the objective function $\mathcal{L}_{DR}^\delta(\theta, P_{\mathrm{tr}}) = \min_{\nu \ge 0}\left\{\nu\delta^2 + \mathbb{E}_{P_{\mathrm{tr}}}\ell_s(\theta, p_i, \nu)\right\}$ is simplified to $\mathbb{E}_{P_{\mathrm{tr}}}\ell_s(\theta, p_i, \nu)$. The following proposition provides the relationships between $\delta$ (the radius in Eq.(5)) and $\nu$ (the dual variable in Proposition 1).

**Proposition 3 (Relationships between $\delta$ and $\nu$)** *For a given $\theta$, consider the dual objective function in Eq.(6),*

- *(i) For $\nu_2 \ge \nu_1 \ge 0$, we denote $\delta_1, \delta_2$ such that $\nu_1, \nu_2$ are the minimizer of $\mathcal{L}_{DR}^{\delta_1}(\theta, P_{\mathrm{tr}}), \mathcal{L}_{DR}^{\delta_2}(\theta, P_{\mathrm{tr}})$ respectively. Then we have $\delta_1 \ge \delta_2$.*

- *(ii) With Assumption 1, fix a constant $\nu > \beta$ and let $\delta^2 = -\frac{\partial \sum_{p_i \in P_{\mathrm{tr}}}\ell_s(\theta, p_i, \nu)}{\partial \nu}$. Then $\mathcal{L}_{DR}^\delta(\theta, P_{\mathrm{tr}}) = \nu\delta^2 + \mathbb{E}_{P_{\mathrm{tr}}}\ell_s(\theta, p_i, \nu)$.*

*Proof.*

*Proof of (i)*
We first establish the monotonic relationship between $\nu$ and $\delta$. This result is intuitive, but for completeness, we provide a rigorous proof here. From the strong duality property proposed by Gao & Kleywegt (2023, Theorem 1),

$$\mathcal{L}_{DR}^\delta(\theta, P_{\mathrm{tr}}) = \inf_{\nu \ge 0}\left\{\nu\delta^2 + \sup_Q\left\{\mathbb{E}_Q[\ell(\theta, q)] - \nu\mathcal{GW}(Q, P)\right\}\right\} = \min_{\nu \ge 0}\left\{\nu\delta^2 + \mathbb{E}_{P_{\mathrm{tr}}}\ell_s(\theta, p_i, \nu)\right\} \tag{42}$$

Besides, from the Proposition 1 of (Sinha et al., 2018), we have

$$\sup_Q\left\{\mathbb{E}_Q[\ell(\theta, q)] - \nu\mathcal{GW}(Q, P)\right\} = \mathbb{E}_{P_{\mathrm{tr}}}\ell_s(\theta, p_i, \nu) \tag{43}$$

For any $\nu_2 \ge \nu_1 \ge 0$, Since $\nu_1, \nu_2$ are the minimizer of $\mathcal{L}_{DR}^{\delta_1}(\theta, P_{\mathrm{tr}}), \mathcal{L}_{DR}^{\delta_2}(\theta, P_{\mathrm{tr}})$ respectively. Let $Q_1 \in \arg\sup_Q\left\{\mathbb{E}_Q[\ell(\theta, q)] - \nu_1\mathcal{GW}(Q, P_{\mathrm{tr}})\right\}$, $Q_2 \in \arg\sup_Q\left\{\mathbb{E}_Q[\ell(\theta, q)] - \nu_2\mathcal{GW}(Q, P_{\mathrm{tr}})\right\}$, we have $\delta_1 \in \{\mathcal{GW}(Q, P_{\mathrm{tr}}) \mid Q \in Q_1\}$ and $\delta_2 \in \{\mathcal{GW}(Q, P_{\mathrm{tr}}) \mid Q \in Q_2\}$.

It is not hard to know

$$\sup_{Q} \left\{ \mathbb{E}_Q[\ell(\theta, q)] - \nu_1 \mathcal{GW}(Q, P_{\text{tr}}) \right\} \geq \sup_{Q} \left\{ \mathbb{E}_Q[\ell(\theta, q)] - \nu_2 \mathcal{GW}(Q, P_{\text{tr}}) \right\} \tag{44}$$

Considering that a larger radius $\delta$ results in a larger loss $\mathcal{L}_{DR}^{\delta}(\theta, P_{\text{tr}})$, it follows that $\delta_1 \geq \delta_2$.

*Proof of (ii)*

Now we derive the precise relationship between $\nu$ and $\delta$. From (Gao & Kleywegt, 2023, Lemma 3 (ii)), we know that $\ell_s(\theta, p_i, \nu)$ is convex and non-increasing in $\nu$. Recall that in Eq.(7), we define $\ell_s(\theta, p_i, \nu)$ as $\ell_s(\theta, p_i, \nu) := \sup_{q \in \mathcal{M}} \left[ \ell(\theta, q) - \nu \mathrm{d}_{\mathbf{g}}^2(q, p_i) \right]$. Building on this formulation, we further define the following quantities:

$$\overline{D}(\theta, \nu, p_i) := \limsup_{\delta \downarrow 0} \left\{ \mathrm{d}_{\mathbf{g}}(q_i, p_i) : \ell(\theta, q_i) - \nu \mathrm{d}_{\mathbf{g}}^2(q_i, p_i) \geq \ell_s(\theta, p_i, \nu) - \delta \right\}, \tag{45}$$

$$\underline{D}(\theta, \nu, p_i) := \liminf_{\delta \downarrow 0} \left\{ \mathrm{d}_{\mathbf{g}}(q_i, p_i) : \ell(\theta, q_i) - \nu \mathrm{d}_{\mathbf{g}}^2(q_i, p_i) \geq \ell_s(\theta, p_i, \nu) - \delta \right\}. \tag{46}$$

Furthermore, (Gao & Kleywegt, 2023, lemma 3 (iv)) shows that $\underline{D}^p(\theta, \nu, p_i)$ is a subderivative of $\ell_s(\theta, p_i, \nu)$ with respect to $\nu$. Specifically:

$$-\overline{D}^2(\theta, \nu, p_i) \leq \frac{\partial \ell_s(\theta, p_i, \nu)}{\partial \nu} \leq -\underline{D}^2(\theta, \nu, p_i) \tag{47}$$

Now, define $r_{\min}^2(\theta, p_i, \nu) := \min_{q_i \in \mathcal{M}} \left\{ \mathrm{d}_{\mathbf{g}}(q_i, p_i) \mid \ell(\theta, q_i) - \nu \mathrm{d}_{\mathbf{g}}^p(p_i, q_i) = \ell_s(\theta, p_i, \nu) \right\}$ as the minimum distance between $p_i$ and all the $q_i$'s that attain the supremum of $\ell(\theta, q_i) - \nu \mathrm{d}_{\mathbf{g}}^p(p_i, q_i)$ in $\mathcal{M}$. Similarly, define $r_{\max}^2(\theta, p_i, \nu) := \max_{q_i \in \mathcal{M}} \left\{ \mathrm{d}_{\mathbf{g}}(q_i, p_i) \mid \ell(\theta, q_i) - \nu \mathrm{d}_{\mathbf{g}}^p(p_i, q_i) = \ell_s(\theta, p_i, \nu) \right\}$ as the maximum distance between $p_i$ and all the $q_i$'s that attain the infimum of $\ell(\theta, q_i) - \nu \mathrm{d}_{\mathbf{g}}^p(p_i, q_i)$ in $\mathcal{M}$. Thus, we have the following inequality:

$$-r_{\max}^2(\theta, p_i, \nu) \leq -\overline{D}^2(\theta, \nu, p_i) \leq \frac{\partial \ell_s(\theta, p_i, \nu)}{\partial \nu} \leq -\underline{D}^2(\theta, \nu, p_i) \leq -r_{\min}^2(\theta, p_i, \nu) \tag{48}$$

Note that $\ell(\theta, p_i)$ is geodesically $\beta$-smooth with respect to $p_i$, and $\ell(\theta, q_i) - \nu \mathrm{d}_{\mathbf{g}}^2(q_i, p_i)$ is at least geodesically $(\nu - \beta)$-strongly concave and $(\nu + \beta)$-smooth. Therefore, $r_{\min}^2(\theta, p_i, \nu) = r_{\max}^2(\theta, p_i, \nu)$, which we denotes as $r^2(\theta, p_i, \nu)$ in the following. Finally, we obtain

$$\delta^2 = -\mathbb{E}_{P_{\text{tr}}} \frac{\partial \ell_s(\theta, p_i, \nu)}{\partial \nu} = \mathbb{E}_{P_{\text{tr}}} r^2(\theta, p_i, \nu) \tag{49}$$

$\square$

**Lemma 2** *Under Assumption 1, let the expected surrogate loss be denoted by $\mathcal{L}_s(\theta, P_{\text{tr}}, \nu) = \mathbb{E}_{P_{\text{tr}}} \ell_s(\theta, p_i, \nu)$. The corresponding estimated value, computed with the approximate geodesic distance in Algorithm 2, is denoted as $\widehat{\mathcal{L}}_s(\theta, P_{\text{tr}}, \nu)$. By fixing $\nu > \beta$ and setting $\nu' = c_1 \nu$, where $c_1 = \kappa(\sqrt{\kappa} + \sqrt{\kappa - 1})^4$, we have the following relationship:*

$$\mathcal{L}_s(\theta, P_{\text{tr}}, \nu') \leq \widehat{\mathcal{L}}_s(\theta, P_{\text{tr}}, \nu) \leq \mathcal{L}_s(\theta, P_{\text{tr}}, \nu). \tag{50}$$

*Proof.*

Consider the surrogate loss $\ell_s(\theta, p_i, \nu) := \sup_{q \in \mathcal{M}} \left[ \ell(\theta, q) - \nu \mathrm{d}_{\mathbf{g}}^2(q, p_i) \right]$ in Eq.(7). It is clear that $\ell_s(\theta, p_i, \nu) \geq \ell_s(\theta, p_i, \nu')$ for $\nu \leq \nu'$ as shown in Proposition 3. Now, consider the approximated surrogate loss defined as $\hat{\ell}_s(\theta, p_i, \nu) = \sup_{q \in \mathcal{M}} \left[ \ell(\theta, q) - \nu \hat{\mathrm{d}}_{\mathbf{g}}^2(q, p_i) \right]$, where $\hat{\mathrm{d}}_{\mathbf{g}}^2(q, p_i)$ is an approximated distance as stated by Lemma 1. Therefore, we have

$$\ell_s(\theta, p_i, \nu') \leq \hat{\ell}_s(\theta, p_i, \nu) \leq \ell_s(\theta, p_i, \nu).$$

By taking the expectation over the dataset $P_{\text{tr}}$, we complete the proof.

$\square$

*Proof.*[proof of Theorem 1] Let $c_1 = \kappa(\sqrt{\kappa} + \sqrt{\kappa - 1})^4$ and denote $\nu' = c_1\nu$. Recall that $\hat{\theta}$ is the solution of $\min_\theta \mathbb{E}_{P_{\mathrm{tr}}} \ell_s(\theta, p_i, \nu)$ using the approximate geodesic distance, *i.e.*, $\min_\theta \widehat{\mathcal{L}}_s(\theta, P_{\mathrm{tr}}, \nu) = \widehat{\mathcal{L}}_s(\hat{\theta}, P_{\mathrm{tr}}, \nu)$. By Lemma 2, it follows that

$$\min_\theta \mathcal{L}_s(\theta, P_{\mathrm{tr}}, \nu') \leq \mathcal{L}_s(\hat{\theta}, P_{\mathrm{tr}}, \nu') \leq \min_\theta \widehat{\mathcal{L}}_s(\theta, P_{\mathrm{tr}}, \nu) \leq \min_\theta \mathcal{L}_s(\theta, P_{\mathrm{tr}}, \nu) \tag{51}$$

From Proposition 3(ii), we know that $\frac{\partial \ell_s(\theta, p_i, \nu)}{\partial \nu} = -r^2(\theta, p_i, \nu)$. Recall that $r^2(\theta, p_i, \nu) := \min_{q_i \in \mathcal{M}} \{ \mathrm{d}_{\mathbf{g}}(q_i, p_i) \mid \ell(\theta, q_i) - \nu \mathrm{d}_{\mathbf{g}}^p(p_i, q_i) = \ell_s(\theta, p_i, \nu) \}$ represents the minimum distance between $p_i$ and all the $q_i$'s that attain the supremum of $\ell(\theta, q_i) - \nu \mathrm{d}_{\mathbf{g}}^p(p_i, q_i)$ in $\mathcal{M}$. Furthermore, since the function $\ell(\theta, q_i) - \nu \mathrm{d}_{\mathbf{g}}^2(q_i, p_i)$ is at least geodesically $(\nu - \beta)$-strongly concave and $(\nu + \beta)$-smooth for a given $p_i$, there exists an $S_i := |\nabla \ell(\theta, p_i)| \geq 0$ satisfies $\frac{S_i}{\nu + \beta} \leq r(\theta, p_i, \nu) \leq \frac{S_i}{\nu - \beta}$. Thus, we can express $\ell_s(\theta, p_i, \nu')$ as:

$$\ell_s(\theta, p_i, \nu') = \ell_s(\theta, p_i, \nu \to \infty) + \int_\infty^{\nu'} -r^2(\theta, p_i, t)\mathrm{d}t = \ell(\theta, p_i) + \int_\infty^{\nu'} -r^2(\theta, p_i, t)\mathrm{d}t. \tag{52}$$

$$\int_\infty^{\nu'} -r^2(\theta, p_i, t)\mathrm{d}t \geq \int_\infty^{\nu'} -\left(\frac{S_i}{t + \beta}\right)^2 \mathrm{d}t = \left.\frac{S_i^2}{t + \beta}\right|_\infty^{\nu'} = \frac{S_i^2}{\nu' + \beta}$$

Similarly, for $\ell_s(\theta, p_i, \nu)$, we have:

$$\ell_s(\theta, p_i, \nu) = \ell(\theta, p_i) + \int_\infty^\nu -r^2(\theta, p_i, t)\mathrm{d}t.$$

$$\int_\infty^\nu -r^2(\theta, p_i, t)\mathrm{d}t \leq \int_\infty^\nu -\left(\frac{S_i}{t - \beta}\right)^2 \mathrm{d}t = \left.\frac{S_i^2}{t - \beta}\right|_\infty^\nu = \frac{S_i^2}{\nu - \beta}$$

We have $\ell_s(\theta, p_i, \nu) \leq \ell(\theta, p_i) + \frac{S_i^2}{\nu - \beta}$ and $\ell_s(\theta, p_i, \nu') \geq \ell(\theta, p_i) + \frac{S_i^2}{\nu' + \beta}$. As $\ell(\theta, p_i) \geq 0$, we obtain the following bound:

$$\mathbb{E}_{P_{\mathrm{tr}}} \ell_s(\theta, p_i, \nu) \leq \frac{\nu' + \beta}{\nu - \beta} \mathbb{E}_{P_{\mathrm{tr}}} \ell_s(\theta, p_i, \nu') \tag{53}$$

Combining this with Eq. (51), we have

$$\mathbb{E}_{P_{\mathrm{tr}}} \ell_s(\hat{\theta}, p_i, \nu) \leq \frac{\nu' + \beta}{\nu - \beta} \mathbb{E}_{P_{\mathrm{tr}}} \ell_s(\hat{\theta}, p_i, \nu')$$

$$\leq \frac{c_1\nu + \beta}{\nu - \beta} \min_\theta \widehat{\mathcal{L}}_s(\theta, P_{\mathrm{tr}}, \nu),$$

$$\leq c_1\kappa \min_\theta \mathbb{E}_{P_{\mathrm{tr}}} \ell_s(\theta, p_i, \nu) \tag{54}$$

which completes the proof. $\square$

## E  DETAILS FOR IMPLEMENTATION

In Algorithm 1, for the sake of simplicity, we omit two potentially time-consuming aspects: Jacobian regularization and the SVD for computing the subspace $\mathrm{T}_{\mathrm{appr}}(\mathbf{x}, \tau_0)$. For the former issue, Jacobian regularization can be efficiently implemented using random projection (Hoffman et al., 2019), thereby circumventing the necessity of computing the full Jacobian and only slightly increasing training time. Regarding the latter issue, we utilize an efficient randomized singular value decomposition (SVD) method (Halko et al., 2011), which is particularly well suited to our setting. This is because, under our approach, the top singular vectors of the Jacobian matrix are highly concentrated as shown in Figure 2 and 8, which will benefit both the efficiency and effectiveness of the algorithm. Further details are provided below.

**Randomized SVD method** The randomized SVD algorithm approximates the range of $\mathbf{J_g} \in \mathbb{R}^{d_e*d}$ by projecting it onto a lower-dimensional subspace using a random Gaussian matrix $\mathbf{\Omega} \in \mathbb{R}^{d*d_s}$, where the $d_s$ is the target dimension. Let $\mathbf{Y} = \mathbf{J_g}\mathbf{\Omega} \in \mathbb{R}^{d_e \times d_s}$ denote this projection. We compute the SVD of the smaller matrix $\mathbf{Y}$, yielding $\mathbf{Y} = \widetilde{\mathbf{U}}\widetilde{\mathbf{\Sigma}}\widetilde{\mathbf{V}}$. The approximate right singular vectors of $\mathbf{J_g}$ are given by $\widehat{\mathbf{V}} = \mathbf{\Omega}\widetilde{\mathbf{V}}$. Please refer to (Halko et al., 2011) for more details.

**Proposition 4 ((Halko et al., 2011))** *Let $\mathbf{J_g}$ be a $d_e \times d$ $(d \gg d_e)$ matrix with singular values $\sigma_1 \geq \sigma_2 \geq \cdots \geq \sigma_{d_e}$. We compute the approximate SVD by the randomized method described above, and setting the target dimension to $d_s = 2d_{\tau_0} + 1$. Then the expected approximation error in the Frobenius norm, $\|\mathbf{J_g} - \mathbf{J_g}\widehat{\mathbf{V}}\widehat{\mathbf{V}}^\top\|_F$, is bounded by:*

$$\mathbb{E}\|\mathbf{J_g} - \mathbf{J_g}\widehat{\mathbf{V}}\widehat{\mathbf{V}}^\top\|_F \leq \frac{\sqrt{6}}{2}\left(\sum_{j=d_s+1}^{d_e} \sigma_j^2\right)^{1/2}, \tag{55}$$

The term $\left(\sum_{j=d_s+1}^{d_e} \sigma_j^2\right)^{1/2}$ represents the square root of the sum of the squared singular values beyond the $d_s$-th component. The Jacobian matrix $\mathbf{J_g}$, derived from $\mathbf{g}$ trained using our manifold-guided game approach, shows a rapid decay in its singular values, as demonstrated in Figures 2 and 8. As a result, the majority of the singular values are concentrated in the top few ones. This concentration allows for an efficient and accurate computation of the subspace $\mathrm{T}_{\mathrm{appr}}(\mathbf{x}, \tau_0)$ defined in Eq.(10). To see this more concretely, consider an image task with an input dimension of $d = 224 * 224 * 3$ and an output dimension $d_e = 128$ (typically used in contrastive learning). In this case, performing the singular value decomposition (SVD) on $\mathbf{J_g}$ requires approximately $0.7$ GFLOPs (giga floating-point operations) . For comparison, the cost of a forward propagation in Resnet-18 is $1.8$ GFLOPs.

Moreover, we leverage advanced automatic differentiation tools available in the deep learning framework PyTorch (Paszke et al., 2019). Specifically, we avoid the explicit computation of $\mathbf{J_g}$ by using Jacobian-vector and vector-Jacobian product operations (`jvp` and `vjp`). These operations eliminate the need to store the full Jacobian matrix and further improve computational efficiency. Additionally, we update the approximated tangent information every few iterations in Algorithm 2. In Appendix F.5, we compare the running time with standard WDRO (which does not consider the data manifold) and other adversarial training algorithms. While our algorithm requires more time (around $1.5$ times), it achieves better performance on accuracy.

# F EXPERIMENTS

We conduct a series of experiments across several scenarios to evaluate our proposed method. Specifically, we test it in three typical distribution shift tasks: dealing with noisy data, attacked data and imbalanced data. All experiments ware implemented with PyTorch on a single NVIDIA RTX 6000 Ada, and each instance is repeated by 5 times. In Section F.1, we present the dynamic update mechnism for the hyperparameter $\nu$. In Section F.2 and F.3, we show that our method outperforms current WDRO methods and certain SOTA defense techniques for clean and contaminated data. In Section F.4, we conduct data augmentation based on MWDRO to address the over-fitting issue and improve generalization on limited data. In our experimental setup, we set the hyperparameters $\lambda_1 = 0.01$ and $\lambda_2 = 1$ in Eq.(8) and Eq.(12) respectively to configure the algorithm, unless otherwise specified.

## F.1 SELECTING STRATEGY FOR THE LAGRANGIAN PARAMETER IN EQ.(6)

In this section, we briefly introduce the dynamic adjustment approach for the Lagrangian parameter $\nu$ in Eq.(6) as proposed in **Uni-DR** (Bui et al., 2022). We have modified some notations for our scenarios, and the readers may check the paper for more details on this method.

In Algorithm 3, let $\nu_j$ denote the value of $\nu$ at iteration $j$. The parameter $\eta_\nu$ represents the learning rate for updating $\nu$, determining how quickly $\nu$ adjusts in response to the perturbation discrepancy; $\delta$ is the predefined maximum allowable perturbation radius in Eq.(2); and $\mathbf{d_g}(q_i, \hat{q}_i)$ is the geodesic

distance between the adversarial example $\hat{q}_i$ and the original input $q_i$, which can be approximated by the accumulated steps as shown in Lemma 1. Finally, $\frac{1}{n}\sum_{i=1}^{n} \mathtt{d_g}(q_i, \hat{q}_i)$ represents the average (population) perturbation cost over a batch of $n$ samples.

When computing perturbed samples, $\nu$ is dynamically adjusted, with the adjustment magnitude controlled by the parameter $\eta_\nu$. Intuitively, if the average geodesic distance between adversarial and benign samples is smaller than a specified threshold, $\nu$ decreases in the next iteration to allow adversarial samples $\hat{q}_i$ to deviate further from the benign samples $q_i$, and vice versa.

---

**Algorithm 3** Algorithm 1 using dynamic $\nu$

---

**Input:** training set $P_{\mathrm{tr}} = \{(\mathbf{x}_i, y_i)\}_{i=1}^{n}$, a constant $\nu$, step size sequence $\{\alpha_j > 0\}_{j=0}^{N}$ .
**Output:** the final parameter $\theta^{N+1}$.
 1: Train $\mathbf{g}(\cdot)$ to minimize $\mathcal{L}^*(\theta)$ in Eq.(8).
 2: **for** j = 1, 2, $\cdots$, N **do**
 3:     For each sample $p_i \in P_{\mathrm{tr}}$, compute the corresponding adversarial point $\hat{p}_i \leftarrow$ $\mathtt{Evolve}(\{p_i\}, \mathbf{g}(\cdot), \nu_j)$;
 4:     Updating $\nu_{j+1}$ according to Eq.(56).
 5:     $\theta^{j+1} \leftarrow \theta^j - \alpha_j \frac{1}{n}\sum_{i=1}^{n} \nabla_\theta \left(\ell(\theta^j, \hat{p}_i) - \lambda_2 \left(\ell_{\mathrm{cl}}(p_i) + \lambda_1 \|J_{\mathbf{g}}(p_i)\|_F^2\right)\right)$;
 6: **end for**

---

The dynamic adjustment of $\nu$ is governed by the following update rule:

$$\nu_{j+1} = \nu_j - \eta_\nu \left(\delta - \frac{1}{n}\sum_{i=1}^{n} \mathtt{d_g}(q_i, \hat{q}_i)\right). \tag{56}$$

We still need to select an initial $\nu_0$ to setup this dynamic adjustment strategy. As illustrated in (Bui et al., 2022), the performance of WDOR is relatively insensitive to the choice of $\nu_0$ when the dynamic strategy is employed. Following the recommendation in (Bui et al., 2022), we set $\nu_0 = 0.5$ across all experiments.

## F.2   EVALUATION ON NOISY DATA

To assess the robustness of our model against natural corruptions, we introduce perturbations to each original image using Gaussian white noise. The resulting corrupted image is denoted as $\mathbf{x}' = \mathbf{x} + \epsilon, \epsilon \sim \mathcal{N}\left(0, \sigma_{\mathrm{noise}}^2 \mathbf{I}\right)$. We adjust the noise levels with different values for $\sigma_{\mathrm{noise}}$.

**Datasets.** **CIFAR-**10 and **CIFAR-**100 (Krizhevsky et al., 2012) are two popular datasets for image classification, each of which consists of 60000 color images of $32 \times 32$ pixels, the former one are divided into 10 classes, with 6000 images per class and the latter one contains 100 classes, with 600 images per class. **Tiny-ImageNet-**200 (Krizhevsky et al., 2017) is a subset of the ImageNet dataset, comprising 100000 images across 200 classes, each with 500 images. The images are resized to $64 \times 64$ pixels and include both RGB and grayscale channels. **BuS** (Mo et al., 2023) is a medical image dataset that contains 1519 malignant and 886 benign benign breast ultrasound images. We employ this medical image dataset as noise in medical images is a critical issue.

**Compared methods.** We compare our proposed MWDRO with four baselines: **DropOut+L2** (He et al., 2016) as the standard method for image classification, a practical implementation of **WDRO** proposed by (Sinha et al., 2018), and the **KL-DRO** by (Levy et al., 2020). The randomized smoothing method **RS** proposed by (Cohen et al., 2019).

Figure 5 demonstrates that our MWDRO method significantly enhances robustness compared to other methods, particularly as the noise level increases.

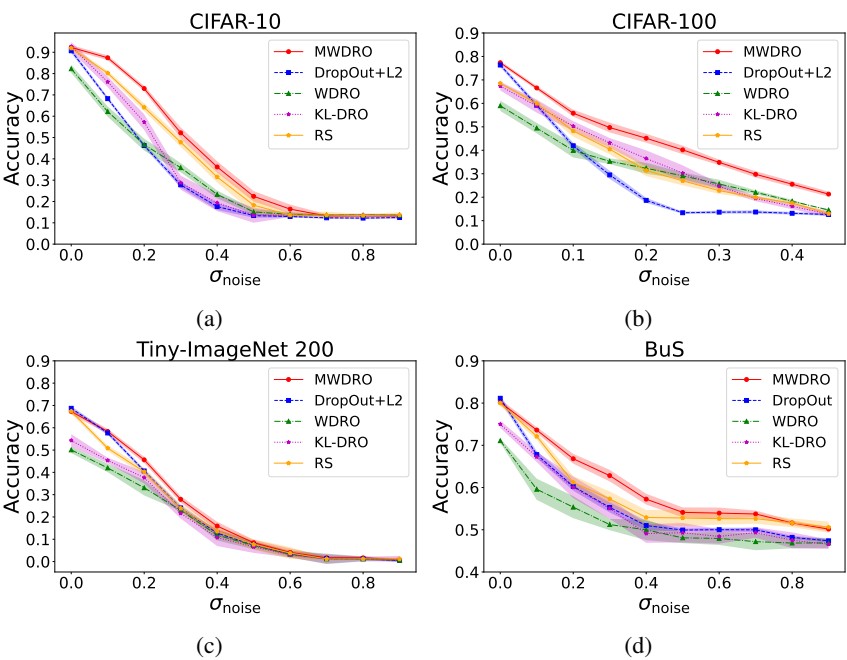

Figure 5: Evaluation of robustness against Gaussian white noise perturbations. The x-axis represents the variance of the additive Gaussian noise, ranging from 0 to 1 (the input data has been normalized). The results demonstrate that our MWDRO method significantly improves robustness across datasets.

### F.3 EVALUATION ON ADVERSARIAL ATTACK

In this section, we evaluate the robustness of our proposed method, MWDRO, against adversarial attacks. We conduct experiments on two widely-used datasets: CIFAR-10, CIFAR-100 and Tiny-ImageNet-200. These datasets are standard benchmarks for assessing the effectiveness of adversarial defenses.

We utilize a variety of adversarial attack methods to evaluate the performance of our model. The attacks include Fast Gradient Sign Method (FGSM) (Madry et al., 2018), Projected Gradient Descent (PGD) (Kurakin et al., 2018), AutoAttack (AA) (Croce & Hein, 2020), the Jacobian-based Saliency Map Attack (JSMA) (Brendel et al., 2019) and C&W (L2) by (Carlini & Wagner, 2017). Complete results are presented in Table 2, Table3, and Table 4.

We compare our proposed MWDRO with four baselines: a practical implementation of **WDRO** proposed by (Sinha et al., 2018), a unified WDRO framework **Uni-DR** (Bui et al., 2022) and the popular adversarial training methods **PGD-AT** (Madry et al., 2018) and **TRADES** (Zhang et al., 2019). For both PGD-AT and TRADES algorithms, we set the defense perturbation bounds $\epsilon$ to match their corresponding attack scenarios as shown in Table 2, 3 and 4. And their adversarial attacks were executed with 40 iterations and a step size of 0.005.

Table 2: Comparisons of clean and adversarial accuracies against different attacks for the Tiny-ImageNet-200. Best scores are highlighted in boldface.

| **Tiny-ImageNet-200** | WDRO | TRADES | PGD-AT | MWDRO ($\delta = 0.01$) | MWDRO ($\delta = 0.005$) |
|---|---|---|---|---|---|
| Clean accuracy | 28.47 | 39.48 | 40.23 | 43.82 | **55.42** |
| PGD ($\varepsilon = 0.01$) | 8.37 | 14.83 | 15.26 | **17.62** | 15.18 |
| AA ($\varepsilon = 0.01$) | 6.13 | 11.23 | 12.38 | **13.48** | 10.73 |
| C&W ($\lambda = 0.01$) | 9.81 | 12.46 | 16.32 | **17.26** | 13.34 |

Table 3: Comparisons of clean data accuracy and adversarial accuracies against different attacks for the CIFAR-10 dataset. The table displays the performance of four different baselines and our proposed method MWDRO. Each method is evaluated on clean data and under various adversarial attacks (FGSM, PGD, JSMA, AA, and C&W) with different perturbation strengths. Best scores are highlighted in boldface.

| **CIFAR-10** | WDRO | TRADES | Uni-DRO ($\delta = 0.03$) | PGD-AT | MWDRO ($\delta = 0.03$) | MWDRO ($\delta = 0.06$) |
|---|---|---|---|---|---|---|
| Clean accuracy | 83.10 | 87.49 | 85.48 | 85.41 | **88.45** | 85.71 |
| FGSM ($\varepsilon = 0.03$) | 57.64 | 73.18 | 73.35 | 72.44 | **77.51** | 76.92 |
| FGSM ($\varepsilon = 0.06$) | 48.38 | 53.33 | 52.65 | 51.58 | **68.28** | 68.11 |
| PGD ($\varepsilon = 0.03$) | 39.37 | 45.40 | 47.65 | 44.81 | **48.28** | 49.32 |
| PGD ($\varepsilon = 0.06$) | 12.38 | 19.59 | 20.02 | 14.23 | 23.58 | **27.65** |
| JSMA ($\varepsilon = 0.03$) | 54.29 | 74.51 | 74.39 | 73.85 | **79.56** | 74.53 |
| JSMA ($\varepsilon = 0.06$) | 40.01 | 66.27 | 67.41 | 64.25 | **74.47** | 73.90 |
| AA ($\varepsilon = 0.03$) | 41.38 | 45.74 | 45.85 | 43.05 | **47.73** | 45.28 |
| AA ($\varepsilon = 0.06$) | 11.11 | 14.78 | 14.02 | 13.69 | 14.34 | **15.89** |
| C&W ($\lambda = 0.01$) | 46.38 | 54.74 | 55.85 | 52.05 | **57.73** | 55.91 |

Table 4: Comparisons of clean data accuracy and adversarial accuracies against different attacks for the CIFAR-100 dataset. The table displays the performance of four baselines and our proposed method MWDRO. Each method is evaluated on clean data and under various adversarial attacks (FGSM, PGD, JSMA, AA, and C&W) with different perturbation strengths. Best scores are highlighted in boldface.

| **CIFAR-100** | WDRO | TRADES | Uni-DRO ($\delta = 0.01$) | PGD-AT | MWDRO ($\delta = 0.01$) | MWDRO ($\delta = 0.02$) |
|---|---|---|---|---|---|---|
| Clean accuracy | 66.28 | 67.94 | 67.67 | 67.64 | **69.58** | 67.25 |
| FGSM ($\varepsilon = 0.01$) | 38.71 | 48.05 | 49.54 | 45.97 | **49.96** | 48.41 |
| FGSM ($\varepsilon = 0.02$) | 20.26 | 35.82 | 36.47 | 29.33 | 35.97 | **37.01** |
| PGD ($\varepsilon = 0.01$) | 41.36 | 44.55 | 45.59 | 42.64 | **46.42** | 46.21 |
| PGD ($\varepsilon = 0.02$) | 17.95 | 31.49 | 32.22 | 22.24 | **33.71** | 32.10 |
| JSMA ($\varepsilon = 0.01$) | 34.15 | 46.51 | 47.71 | 45.37 | **49.24** | 38.91 |
| JSMA ($\varepsilon = 0.02$) | 28.27 | 36.84 | 39.53 | 33.30 | **45.72** | 33.64 |
| AA ($\varepsilon = 0.01$) | 20.38 | 44.74 | 44.85 | 40.32 | **47.52** | 41.74 |
| AA ($\varepsilon = 0.02$) | 9.69 | 14.76 | 15.01 | 14.69 | 15.14 | **15.87** |
| C&W ($\lambda = 0.01$) | 20.38 | 28.74 | **29.85** | 28.05 | 28.93 | 26.44 |

## F.4 EVALUATION ON LONG-TAILED DATA

In this section, we evaluate the effectiveness of our proposed method, MWDRO, in handling long-tailed data distributions. Long-tailed datasets present a challenge due to the significant imbalance between the frequency of common and rare classes. We conduct our experiments on two long-tailed benchmark datasets: CIFAR-10-LT and CIFAR-100-LT. The imbalance in these datasets is quantified by the imbalance factor (IF), which represents the ratio of the number of samples in the most frequent class to those in the least frequent class. We test our method on three versions of these datasets with imbalance factors of 10, 50, and 100. e compare MWDRO with several baseline methods designed to address long-tailed distributions: Decouple (Hong et al., 2021): Disentangle the label distribution from the model prediction; Reweight (Cui et al., 2019): Reweight the loss; Resample (Cui et al., 2019): Over-sampling minor classes; Focal Loss (Lin et al., 2017): Weighting difficult samples; SSL (Khosla et al., 2020): a representation learning method; DRO-LT (Samuel

& Chechik, 2021): A DRO method designed for long-tail learning. We chose three long-tailed versions with imbalance factors (IF) of 10, 50, and 100 for training and the results shows that in highly imbalanced dataset, our MWDRO method combing with basic re-sampling/re-weighting strategy achieve best accuracy.

Table 5: Accuracies of ResNet32 on long-tailed CIFAR-10 and CIFAR-100 datasets. The table shows the performance of various methods across datasets with different imbalance factors (10%, 50%, and 100%). Baseline methods include Decouple, Reweight, Resample, Focal Loss, DRO-LT, and SSL. Our proposed method, MWDRO, and its combination with re-sampling/re-weighting strategies are highlighted. The best accuracy for each dataset and imbalance factor combination is highlighted in boldface.

| Dataset | CIFAR-10-LT | | | CIFAR-100-LT | | |
|---|---|---|---|---|---|---|
| | 100% | 50% | 10% | 100% | 50% | 10% |
| **Decouple (Hong et al., 2021)** | 70.4 | 76.2 | 86.4 | 41.2 | 46.8 | 57.9 |
| **Focal Loss (Lin et al., 2017)** | 70.3 | 76.7 | 86.6 | 38.4 | 44.3 | 55.7 |
| **DRO-LT (Samuel & Chechik, 2021)** | 73.7 | 77.2 | **86.9** | 45.4 | 55.3 | **61.2** |
| **SSL (Khosla et al., 2020)** | 67.3 | 75.4 | 86.5 | 37.5 | 44.0 | 56.7 |
| **Reweight (Cui et al., 2019)** | 70.5 | 74.8 | 86.4 | 34.0 | 43.9 | 57.1 |
| **+MWDRO** | **74.5** | 76.9 | 86.7 | **46.7** | **56.2** | 60.5 |
| **Resample (Cui et al., 2019)** | 66.5 | 74.8 | 86.4 | 33.4 | 43.9 | 55.1 |
| **+MWDRO** | 72.8 | **77.5** | 86.2 | 43.5 | 52.6 | 58.2 |

## F.5 TIME COMPARISON

In Algorithm 1 and Algorithm 2, for the sake of simplicity, we omit two potentially time-consuming aspects: Jacobian regularization and the SVD for computing the subspace $T_{appr}(\mathbf{x}, \tau_0)$. Details on efficient implementation are provided in Appendix E. Table 6 presents the experimental results on running times, normalized relative to the running time of WDRO. The experimental environments are consistent with those in Appendix F.3.

Table 6: Comparison of normalized running time for different methods across datasets. We train each method with 200 epochs. All running times are normalized by dividing them by the running time of WDRO.

| **Dataset** | WDRO | TRADES | Uni-DRO | PGD-AT | MWDRO |
|---|---|---|---|---|---|
| CIFAR-10 | 1.0 | 0.826 | 1.481 | 1.264 | 1.471 |
| CIFAR-100 | 1.0 | 0.947 | 1.489 | 1.172 | 1.434 |
| Tiny-ImageNet-200 | 1.0 | 1.191 | 1.148 | 1.460 | 1.521 |
| BuS | 1.0 | 1.071 | 1.364 | 1.169 | 1.521 |

In the above Table 6, we compare computational running times across various methods using 200 epochs, following standard practices in prior work such as (Bui et al., 2022). As expected, our method requires longer computational time than WDRO because it maintains a similar workflow while incorporating additional data manifold information. As shown in Table 6, our method requires approximately 50% more computation time per epoch compared to classical WDRO.

However, our method exhibits faster convergence — a natural consequence of working with a more focused uncertainty set (a subset of WDRO's uncertainty set). Leveraging this characteristic, we implemented early stopping, where training halts when the robust performance on the validation set shows no improvement for 5 consecutive epochs.

Tables 7 and 8 present the results across various datasets. With early stopping enabled, our method achieves comparable total training times while delivering superior accuracy and robustness, demonstrating an effective balance between computational efficiency and model performance.

Table 7: Comparison of clean accuracy and adversarial robustness across different methods on various datasets. "Clean" represents accuracy on the original test set, while "PGD" shows accuracy under adversarial attack. $\epsilon$ denotes the perturbation bound for PGD attacks.

| Dataset | Evaluation | WDRO | TRADES | Uni-DRO | PGD-AT | MWDRO |
|---|---|---|---|---|---|---|
| CIFAR-10 | Clean | 80.42 | 86.79 | 85.51 | 82.18 | **88.21** |
| | PGD ($\epsilon = 0.03$) | 41.58 | 47.35 | 49.64 | 45.52 | **52.27** |
| CIFAR-100 | Clean | 64.38 | 67.32 | 66.92 | 41.47 | **68.17** |
| | PGD ($\epsilon = 0.01$) | 41.79 | 45.72 | 47.20 | 43.91 | **49.54** |
| Tiny-ImageNet-200 | Clean | 30.09 | 39.11 | 39.37 | 36.87 | **41.36** |
| | PGD ($\epsilon = 0.01$) | 7.41 | 14.67 | **14.92** | 14.46 | 14.80 |
| BuS | Clean | 74.71 | 77.92 | 78.29 | 75.92 | **81.74** |
| | PGD ($\epsilon = 0.01$) | 58.28 | 66.73 | 68.53 | 63.99 | **69.48** |

Table 8: Comparison of normalized running time for different methods across datasets. Early stopping criteria were applied consistently across all methods. All running times are normalized by dividing them by the running time of WDRO.

| Dataset | WDRO | TRADES | Uni-DRO | PGD-AT | MWDRO |
|---|---|---|---|---|---|
| CIFAR-10 | 1.0 | 0.994 | 1.070 | 1.126 | 1.115 |
| CIFAR-100 | 1.0 | 1.15 | 1.139 | 1.142 | 1.108 |
| Tiny-ImageNet-200 | 1.0 | 1.098 | 1.121 | 1.174 | 1.123 |
| BuS | 1.0 | 1.071 | 1.094 | 1.217 | 1.118 |

## F.6 OTHER ABLATION STUDIES FOR THE MWDRO.

We conduct the ablation studies to analyze the effectiveness of the manifold-guided game and the MWDRO method. As shown in Table 9, models trained using the manifold-guided game without DRO (i.e., regularized only by Eq.(8)) demonstrate significant vulnerability to adversarial attacks.

Table 9: We report the Top-1 accuracy on various datasets for both the MWDRO model and models trained without the DRO method, which are regularized solely by the manifold-guided game in Eq.(8). The evaluation includes three types of attacks: PGD ($\epsilon = 0.01$), AA ($\epsilon = 0.01$), and C&W ($\lambda = 0.01$) except for CIFAR-10, except for CIFAR-10, where $\epsilon = 0.03$ is used for both PGD and AA attacks.

| | Cifar10 | | | Cifar100 | | | Tiny-ImageNet-200 | | |
|---|---|---|---|---|---|---|---|---|---|
| | PGD | AA | C&W | PGD | AA | C&W | PGD | AA | C&W |
| Only with Eq.(8) | 43.29 | 42.82 | 48.16 | 24.71 | 28.69 | 20.17 | 7.07 | 7.83 | 10.14 |
| MWDRO | **48.28** | **47.73** | **57.73** | **46.42** | **47.52** | **28.93** | **15.47** | **11.64** | **14.73** |

## F.7 OTHER EXPERIMENTAL RESULTS

In the first stage of Algorithm 1, the encoder $g(\cdot)$ is trained for $N_0$ epochs following the objective of the manifold-guided game, as defined in Eq.(8). This training phase is crucial for obtaining sufficiently accurate tangent information for each data sample $\mathbf{x}$. This stage ensures $\mathbf{g}(\cdot)$ can yield sufficiently good approximate tangent $\mathrm{T}_{\mathrm{appr}}(\mathbf{x}, \tau_0)$ for each data sample $\mathbf{x}$, which is later used

for the gradient ascent to solve the surrogate loss in Eq.(7). Figure 7 illustrates the impact of the hyperparameter $N_0$ on the accuracy of our model across various datasets.

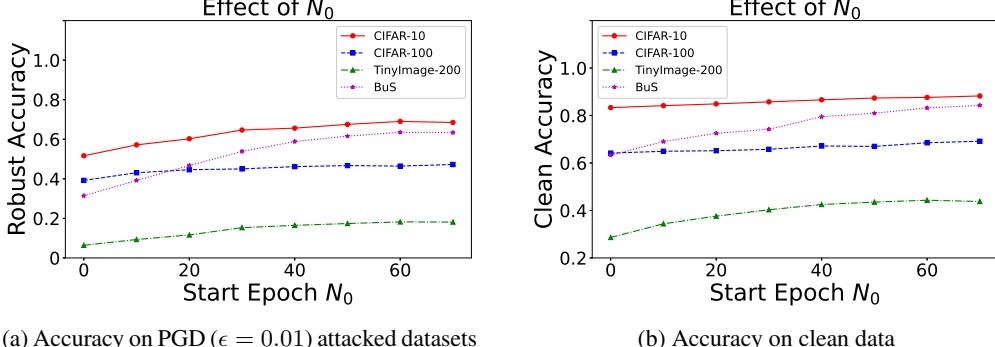

(a) Accuracy on PGD ($\epsilon = 0.01$) attacked datasets      (b) Accuracy on clean data

Figure 6: The effects of the hyperparameter $N_0$ on model accuracy across different datasets. We evaluate the model's performance on CIFAR-10, CIFAR-100, TinyImage-200, and BuS datasets, varying the starting epochs $N_0$ for the second stage of Algorithm 1. Figure 6a and 6b are the achieved accuracy on both attacked (PGD, $\epsilon = 0.01$) and clean datasets, respectively.

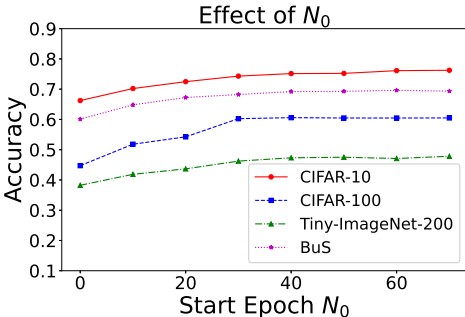

Figure 7: The effects of the hyperparameter $N_0$ on accuracy across different datasets. We evaluate the model's performance on CIFAR-10, CIFAR-100, TinyImage-200, and BuS datasets with various starting epochs $N_0$ for the second stage of Algorithm 1. Gaussian noise with a variance of $\sigma^2 = 0.2^2$ is added to all datasets to test the robustness.

Figure 8 shows the cumulative variance percentage of the Jacobian for various models trained on Tiny-ImageNet-200 and BuS datasets. Our model exhibits the most rapid initial increase in cumulative variance percentage, indicating that fewer singular vectors are needed to approximate the tangent space effectively.

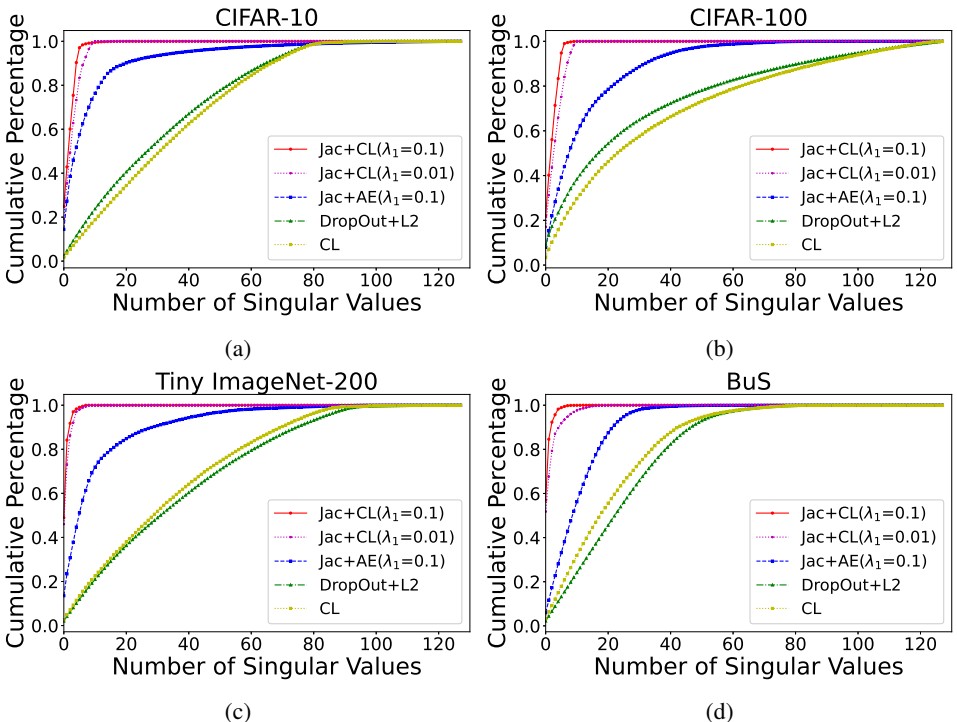

Figure 8: Cumulative percentage of variance of the Jacobian for different models trained on Tiny-ImageNet-200 and BuS datasets using ResNet-18, respectively. The cumulative percentage of variance for first $k$ singular values is calculated as $\frac{\sum_{i=1}^{k} s_i(\mathbf{x})}{\sum_{i=1}^{r} s_i(\mathbf{x})}$. The red line represents our model as described in Eq.(8). The blue line represents the Auto-Encoder model with Jacobian regularization (Jac+AE), the green line represents the CL model with Dropout and L2 regularizer (DropOut+L2), the yellow line represents the CL model without Jacobian regularization (CL).

## G VISUALIZATIONS

This section presents comprehensive visualizations demonstrating our manifold-guided approach's mechanisms and effectiveness. We showcase the method's ability to learn meaningful geometric structures through various experiments, including analysis of manifold dynamics on toy datasets, comparison of learned tangent spaces across different models, and visualization of perturbation trajectories on the MNIST dataset. Our results illustrate how the proposed approach successfully constrains perturbations to follow the intrinsic data manifold structure.

Figure 9 provides a schematic illustration of our methodology. The encoder $\mathbf{g}(\boldsymbol{x})$ should change most when the sample $\boldsymbol{x}$ moves along the semantic variant directions within the data manifold; while changing minimally when moves off the manifold (*i.e.,* $\mathcal{M}^\perp$), or towards the Semantic-Invariant region (*i.e.,* $\mathcal{M}_{SI}(\mathbf{x})$). This results in a distinctive "spiked" distribution of the Jacobian matrix's singular values (as shown in Figure 2), where the top singular vectors naturally correspond to the semantic variant directions in the tangent.

Figure 10 illustrates the procedure for obtaining the perturbed samples described in Algorithm 2 for a 2D toy manifold dataset. The blue dots are generated based on a quadratic function $\mathbf{x}_2 = -\mathbf{x}_1^2$. Specifically, we sample points along $\mathbf{x}_2 = -\mathbf{x}_1^2$ and add Gaussian noise with variance $\delta^2 = 0.01$. To train the encoder $\mathbf{g}$, we employ our manifold-guided game framework, where the augmentation operation is simplified to adding small noise. The tangent at any given point $\mathbf{x}_1$ is approximated using the largest singular vector of the Jacobian $\mathbf{J_g}((\mathbf{x}_1, \mathbf{x}_2))$. For further simplification, we assume each point has a gradient direction of $(0, 1)$ when optimizing the surrogate loss in Eq. equation 7, implying that perturbation directions for all points are the unit vector $(0, 1)$ as depicted in Figure 10(b).

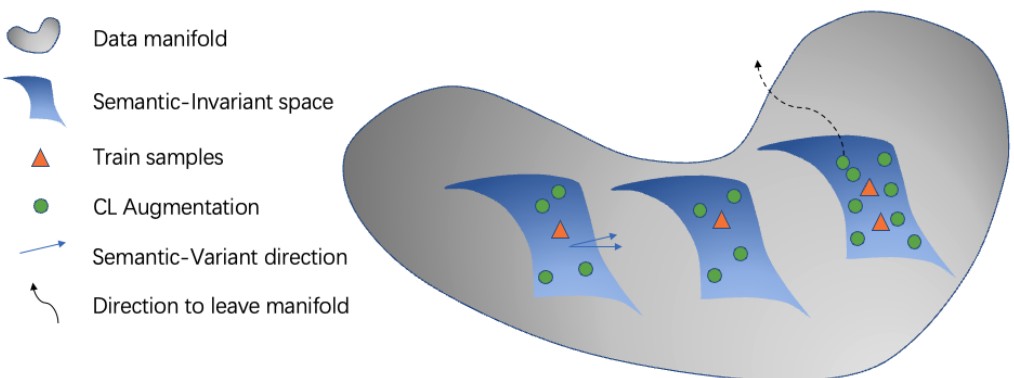

Figure 9: This picture depicts the mechanism of our proposed manifold-guided game.

We select 7 ($S_1$ to $S_7$ marked in Figure 10) points from $\mathbf{x}_2 = -\mathbf{x}_1^2$ as the starting points and observe the trajectory with or without the tangent constraint. Figure 10(a) illustrates that our manifold-guided game extracts local tangent information, effectively constraining points to move approximately along the manifold rather than other directions. Figure 10(b) shows that the samples move freely along the given gradient direction.

Figure 11 provides a qualitative comparison of tangent spaces learned by different models: (i) displays original image from the CIFAR-10 dataset. The subsequent rows show extracted tangent vectors from three models: (ii) the standard ResNet-18; (iii) a model trained with a smoothness penalty term $\lambda \mathbb{E}_{\epsilon \sim \mathcal{N}(0,\sigma^2)} \| f(\mathbf{x} + \epsilon) - f(\mathbf{x}) \|_2^2$ ($\lambda$ takes 0.01), and (iv) a model trained with our proposed manifold guided game. This comparison highlights that our method more effectively captures meaningful directions of variation within the data manifold, consistent with the analysis in Section 4.2 and Appendix B.

Figure 10 illustrates the perturbation trajectories on the MNIST dataset. (a) depicts an original sample from the MNIST dataset. (b) and (c) depict perturbed samples after 50 iterations, generated using the LeNet model trained with standard training and our manifold-guided game, respectively. (d) provides a t-SNE visualization (perplexity = 20) of MNIST digits, along with the perturbed samples from each iteration. Different colors represent distinct digit classes (0–9), while the green and black lines illustrate the perturbation trajectories. While the t-SNE does not perfectly represent the underlying geometric structure of the dataset, it offers valuable insights. Specifically, perturbations without manifold constraints can push data points to meaningless regions. In contrast, our manifold-guided game effectively restricts the exploration space, resulting in a smaller and more meaningful uncertainty set for Wasserstein Distributionally Robust Optimization (WDRO). This demonstrates that our approach enhances the robustness of the model by aligning perturbations with the intrinsic data geometry.

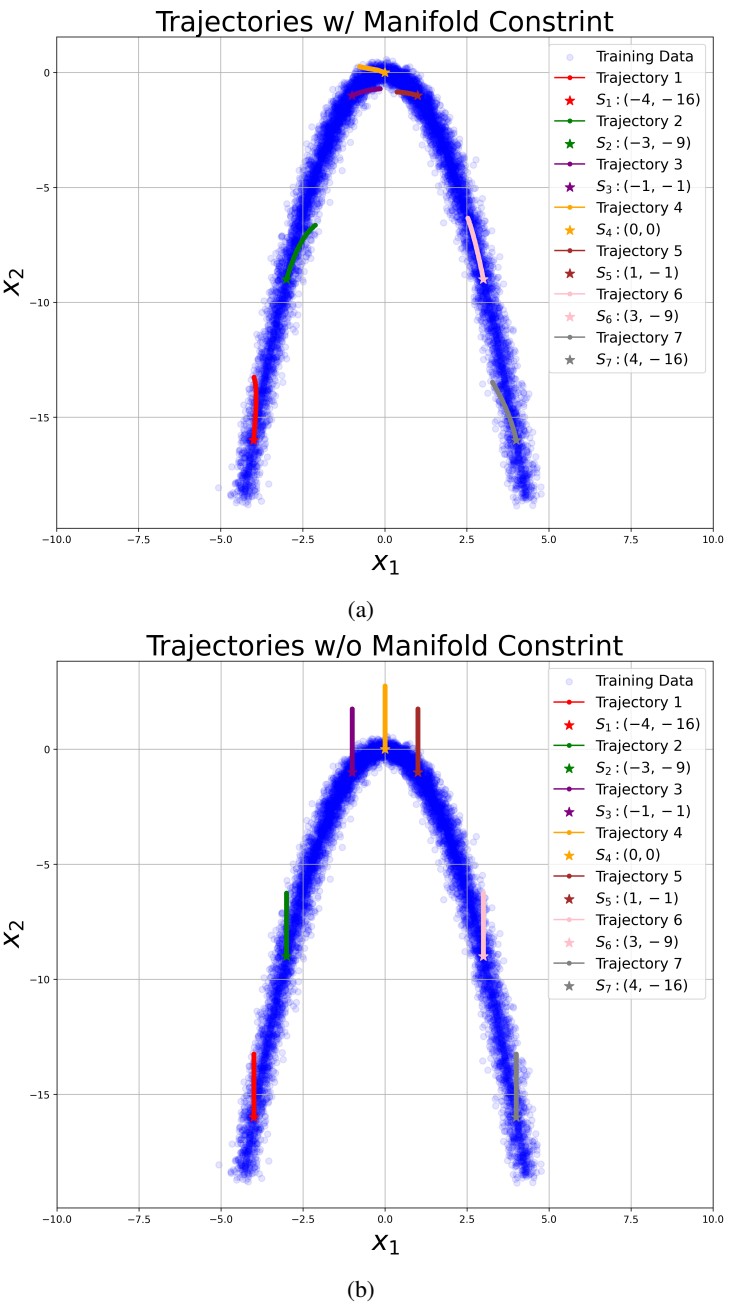

Figure 10: Comparison of trajectory optimization with and without manifold constraints. We apply our manifold-guided game to learn the underlying manifold structure, then simulate the Algorithm 2 for solving the perturbed samples with or without the manifold constraint. (a) shows trajectories constrained to follow the learned manifold structure. (b) shows that perturbation directions for all points are simplified as the unit vector $(0, 1)$. Here $S_i$ are the starting points.

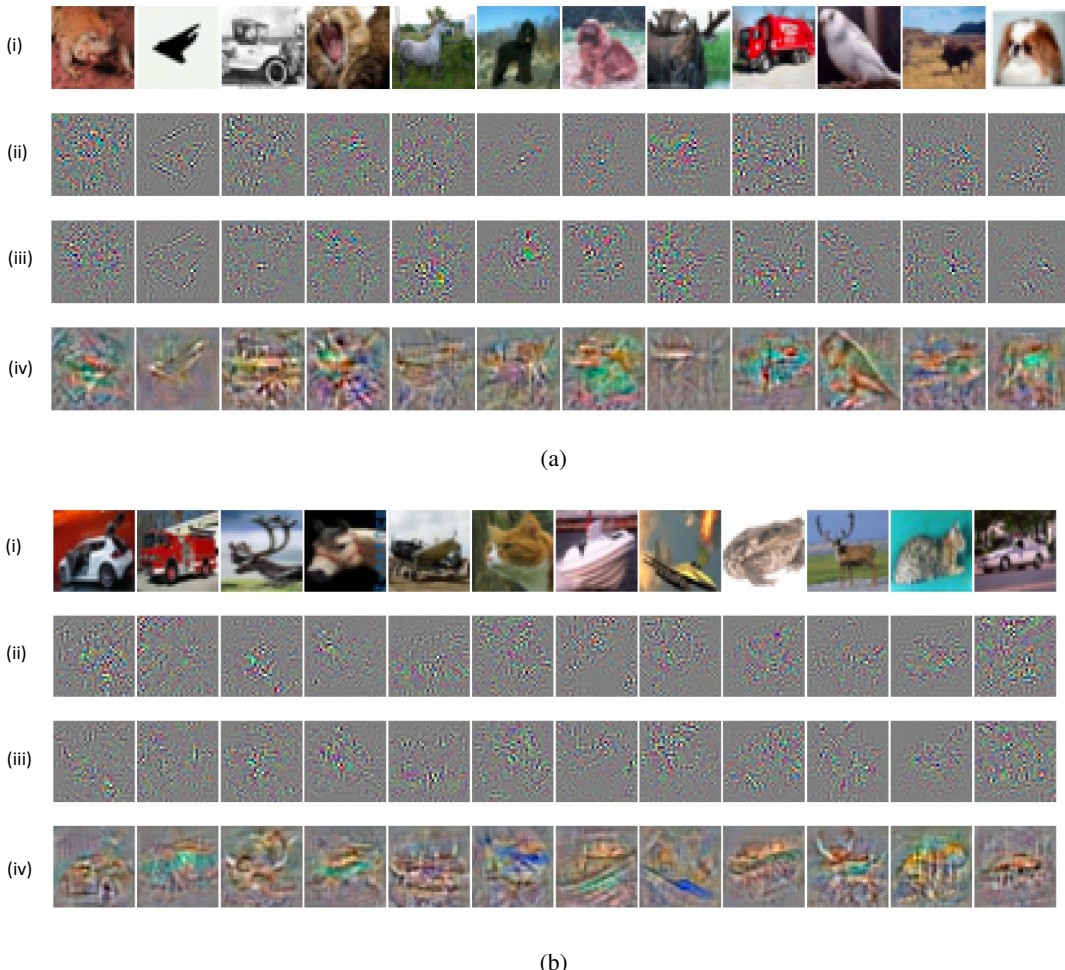

Figure 11: Visualization of the tangents learned by different ResNet-18 models. (i) displays original images from the CIFAR-10 dataset. The subsequent rows show extracted tangent vectors from three models: (ii) the standard ResNet-18; (iii) a model trained with a smoothness penalty term $\lambda\mathbb{E}_{\epsilon\sim\mathcal{N}(0,\sigma^2)}\|f(\mathbf{x}+\epsilon)-f(\mathbf{x})\|_2^2$ ($\lambda$ takes 0.01); and (iv) a model trained with our proposed manifold guided game. These visualizations reveal the models' learned local geometric structure.

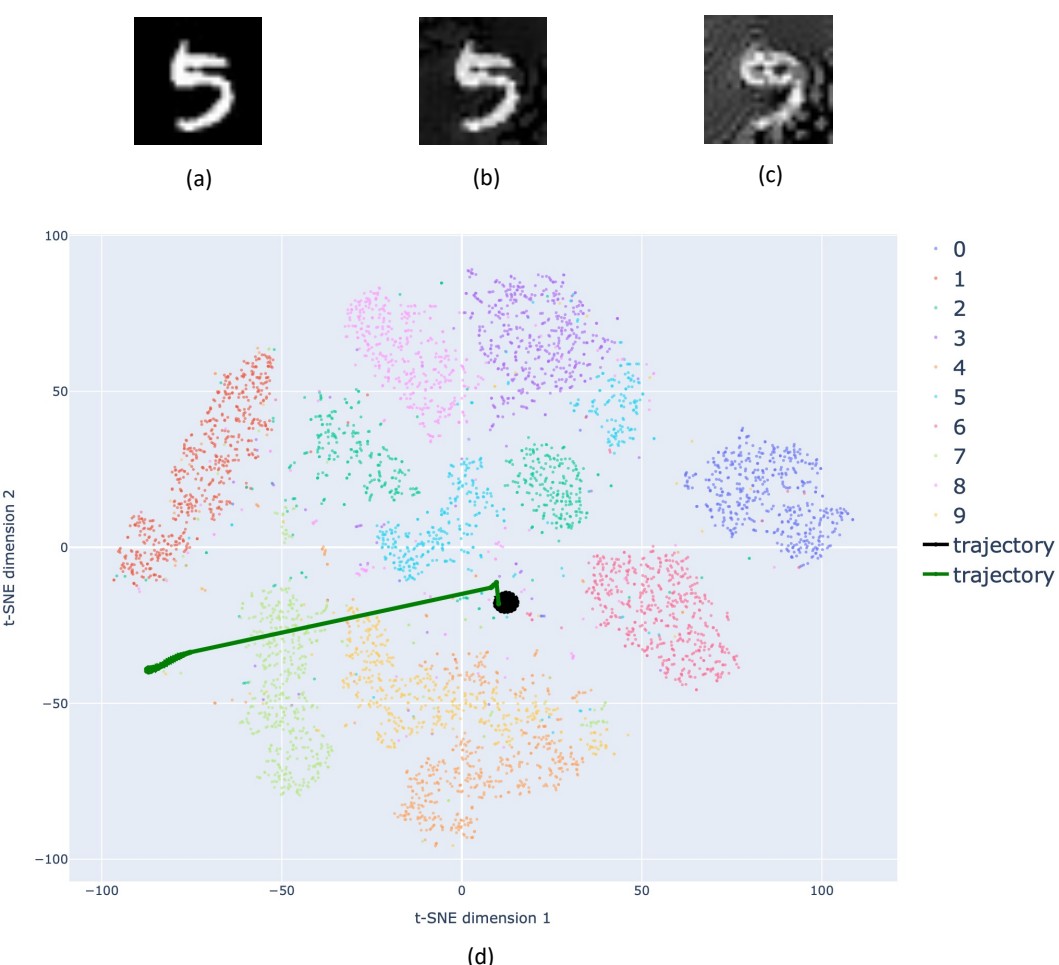

Figure 12: Visualization of perturbation trajectories on the MNIST dataset. (a) An original sample from the MNIST dataset. (b) and (c) show the corresponding perturbed samples after 50 iterations, generated using LeNet model trained with standard methods and our manifold-guided game, respectively. (d) A t-SNE visualization (perplexity = 20) of MNIST digits, along with the perturbed samples mentioned above at each iteration. Different colors represent distinct digit classes (0–9), while the green and black lines illustrate the perturbation trajectories.

