# OpenReview forum: "An Effective Manifold-based Optimization Method for Distributionally Robust Classification"
_ICLR.cc/2025/Conference — ICLR 2025 Poster_

### Official Review · Reviewer_m5cH · 2024-10-29

**Soundness:** 3
**Presentation:** 3
**Contribution:** 2
**Rating:** 5
**Confidence:** 4

**Summary:**

This paper presents a novel manifold-based Distributionally Robust Optimization (DRO) method aimed at enhancing the robustness of deep learning models against distributional shifts. By integrating contrastive learning with Jacobian regularization, the proposed approach captures the manifold structure of the training data to construct a more accurate uncertainty set, thereby improving classification performance under various distributional changes.

**Strengths:**

1.	Writing: This work is presented with good writing style, where the summarized problems with detailed explanations make it easy for readers to understand the problem addressed in this article. However, there exist minor spelling and grammatical errors. The example in Fig.1 helps to understand the connections between geometric representation and semantic tasks.

2.	Novelty: It incorporates geometric structure into the construction of the uncertainty set, potentially leading to more realistic distributional assumptions. Contrastive learning with Jacobian regularization to encode manifold information is novel and seems to consume more geometric information.

3.	Theory: Theorems 1&2 based on geometric scheme provides guarantees on approximation and gradient estimation, offering strong dual reformulations and approximation techniques for geodesic distances, which are crucial for establishing the method's reliability.

The proof seems solid but I have not carefully checked the whole Appendix.

4.	Experiments: Several scenarios and recent baselines are considered, implying improvements in accuracy and robustness under various distributional shifts.

**Weaknesses:**

1.	Approximation. There exists a gap between optimal solutions from original and dual problems. I’m not sure if Theorem 1 states the distances between the original and dual objects?

2.	Selection on v. The bound derived in Theorem 1 heavily relies on v. I would appreciate it if the authors could give more detailed illustrations on the “dynamic mechanism” on setting v empirically?

3.	Broader baselines and empirical settings.
For example, the settings for “Noisy Data” are kind of simple. What’s the variance of the added Gaussian white noise? It is suggested to follow the empirical settings in [1] to widen the difference between the training and testing sets, see Table 1 in [1].

More DRO approaches [1-3] for learning from noisy data are suggested to be included.
Moreover, it is kindly suggested to add introductions to these baselines in the supplementary file for better readability.

[1] https://arxiv.org/abs/2305.14690
[2] https://arxiv.org/abs/1902.07379
[3] https://arxiv.org/abs/2006.04662

4.	Illustrations on manifolds. The paper could benefit from a more in-depth analysis of the algorithm's scalability, especially regarding its performance with large datasets or specific manifold structures.
Synthetic experiments on toy examples with already known geometric structure, e.g., the Swiss or Torus, could help to visualize and estimate the investigated manifolds.


5.	Computational Cost. While the theoretical underpinnings are well-developed, the paper may not provide a comprehensive assessment of the computational efficiency and practicality of the proposed method in real-world applications. Like the computational complexity analysis or empirical time/memory cost.

**Questions:**

Please see the comments in Weakness.

I would be happy to raise my score if these issues are well addressed.

---

> ### Author Response · Authors · 2024-11-23
> **To Reviewer m5cH 1**
>
> We thank the reviewer for the thoughtful comments, and address your concerns below.
>
> **W1**, The gap between optimal solutions from original and dual problems.
>
> Due to the strong duality property, the primal MWDRO problem in Eq.(5) and its dual formulation in Eq.(6) have identical optimal values.
>
> However, computing exact geodesic distances on general manifolds is often computationally intractable.
> To address this challenge, Theorem 1 analyzes the approximation ratio of the surrogate loss when using the estimated geodesic distance in our problem.
> We observe that the optimization problem in Eq.(7) is geodesically strongly concave, which guarantees geometric convergence [4]. This geometric convergence property implies a practical approximation strategy. Lemma 1 shows that by accumulating the optimization steps, we can approximate the geodesic distance up to a constant factor of the true value. Based on this distance approximation, Theorem 1 provides theoretical bounds on the surrogate loss. Specifically, it shows that our computationally feasible approach maintains provable approximation guarantees while avoiding the intractability of exact geodesic distance calculations.
> This theoretical framework connects mathematical optimality and practical implementation, providing a principled foundation for our manifold-based robust optimization method.
>
> **W2**, Selection on $\nu$.
>
> The 'dynamic mechanism' for setting $\nu$ proposed by [3] works as follows:
>
>    $ \nu_{j+1} = \nu_j - \eta_{\nu} \left( \delta - \frac{1}{n} \sum_{i=1}^{n} d_g(q_i, \hat{q_i}) \right). $
>
> Here the $\nu_j$ represents the value of $\nu$ at iteration $j$ in Algorithm 1; $\eta_{\nu}$ denotes the learning rate for updating $\nu$, governing how quickly $\nu$ adjusts in response to the perturbation discrepancy; $\delta$ is the maximum allowable perturbation radius specified in Eq.(7); and $d_g(q_i, \hat{q_i}) $ is the  geodesic distance between the adversarial example $\hat{q_i} $ and the original input $q_i$, approximated through accumulated steps as established in Lemma 1.
> Finally, $\frac{1}{n} \sum_{i=1}^{n} d_g(q_i, \hat{q_i}) $ represents the batch-averaged (population) perturbation cost across $n$ samples.
>
> The mechanism operates by comparing the average geodesic perturbation against the threshold $\delta$. When the population perturbation fall below this threshold, $\nu$ decreases to permit larger deviations between adversarial and original samples. Conversely, when perturbations exceed $\delta$, $\nu$ increases to constrain the adversarial examples closer to their original counterparts. We include the "dynamic mechanism" in **Appendix F.1**.
>
> **W3**,  The variance of the added Gaussian white noise.  More discussion on related works.
>
> **On the variance of the Gaussian noise.** The x-axis represents the variance of the additive Gaussian noise, ranging from 0 to 1. (The input data has been normalized.)
>
> **Comparison with Related Work.**
> Thank you for your references to the related works [1,2,3], which employ dynamic importance reweighting techniques to handle the distribution shift problems. However, there are fundamental differences between their methods and ours:
>
> **1, Different uncertainty set:** From the perspective of Distributionally Robust Optimization (DRO), their "uncertainty set" is primarily focused on distributions over the original input data, potentially incorporating validation datasets. The key difference is that our uncertainty set explicitly accounts for perturbed data, which is crucial for scenarios involving data perturbations, such as adversarial attacks. Since these methods do not involve training on perturbed samples, they may show suboptimal performance when facing adversarial examples.
>
>  **2, Different application scenarios and evaluation metrics:**
> These methods are designed for natural distribution shifts (e.g., covariate shift, label shift) and evaluate performance on datasets  exhibiting such shifts. Our method specifically targets robustness against adversarial perturbations and evaluates performance under various attack scenarios (similar to other WDRO works [5]). While both approaches aim to improve model generalization, they optimize for different types of distribution changes.
>
> **3, Empirical Performance:** Empirically, we observed that methods like GIW [1] are less effective against adversarial examples, despite their strong performance on natural distribution shifts. Experimentally, their model achieves less than 30\% accuracy on PGD-attacked ($\epsilon = 0.03$) examples from the CIFAR-10 dataset (same setting with Table 3). Our model demonstrates improved performance against unseen perturbations.
>
>
> We have expanded the discussion on these points in our related works; please see lines 898–908 for further details.

---

> ### Author Response · Authors · 2024-11-23
> **To Reviewer m5cH 2**
>
> **W4:** Illustrations on manifolds.
>
>
>
> In **Appendix G**, we provide four visual illustrations that demonstrate key aspects of our research. We showcase the method's ability to learn meaningful geometric structures through various perspectives:
>
> 1, The schematic illustration of our methodology (Figure 9);
>
> 2, analysis of manifold dynamics on toy datasets (Figure 10);
>
> 3, comparison of learned tangent spaces across different models (Figure 11);
>
> 4, visualization of perturbation trajectories on the MNIST dataset (Figure 12).
>
>
>
> **W5**. Computational Cost.
>
>
> In **Appendix F**, we compare computational running times across various methods using the same number of epochs (200 epochs). As anticipated, our method requires more time than WDRO because it incorporates additional data manifold information while maintaining a similar workflow. The primary additional computation comes from extracting tangent information through Singular Value Decomposition (SVD) of the Jacobian matrix. To address this, we implemented several optimization techniques detailed in **Appendix E**, including the randomized SVD method. This method proves particularly well-suited for our approach because, as demonstrated in Figure 2, our model's Jacobian exhibits only a few dominant singular values. This characteristic enables the randomized SVD method to achieve both high efficiency and accuracy in approximating the decomposition.
>
> The experimental results show that our method is 50\% slower than the classical WDRO per epoch. However, our method exhibits faster convergence — a natural consequence of working with a more focused uncertainty set (a subset of WDRO's uncertainty set). This faster convergence allows us to implement early stopping and achieve comparable performance with fewer epochs. Specifically, we monitor the robust performance on the validation set during training and stop training when performance stops improving more than 5 epochs.
>
> Finally we achieve similar performance(our model outperforms others) while using similar training time (less than 1.2x). We include this new results in **Appendix F.5** of our paper.
>
> ## Reference
>
>
> [1] Fang et al., Generalizing Importance Weighting to A Universal Solver for Distribution Shift Problems. NeurIPS'23
>
> [2] Shu et al., Meta-Weight-Net: Learning an Explicit Mapping For Sample Weighting. NeurIPS'19
>
> [3] Fang et al., Rethinking Importance Weighting for Deep Learning under Distribution Shift. NeurIPS'20
>
> [4] Boumal, An introduction to optimization on smooth manifolds, 2023.
>
> [5] Bui et al., A unified Wasserstein distributional robustness framework for adversarial training. NeurIPS'22

---

> > ### Author Response · Authors · 2024-11-28
> > **Looking forward to further feedback**
> >
> > We sincerely appreciate your insightful comments, which have greatly helped us enhance the clarity and quality of our manuscript. As noted in our previous response, we have revised the paper in accordance with your suggestions. We hope our revisions have satisfactorily addressed your concerns. If you have any further comments, we will gladly make every effort to address them.

---

> > > ### Author Response · Authors · 2024-12-02
> > >
> > > Dear Reviewers,
> > >
> > > As the deadline of discussion is approaching (less than 24 hours), we would like to know whether our response satisfactorily addressed your concerns. If you have any further comments, we will gladly make every effort to address them.
> > >
> > > Sincerely,
> > >
> > > The authors of #8724

---

### Official Review · Reviewer_P9ds · 2024-10-30

**Soundness:** 3
**Presentation:** 2
**Contribution:** 2
**Rating:** 6
**Confidence:** 3

**Summary:**

This paper proposed a manifold-based distributionally robust optimization method to promote the robustness of existing deep learning models. Specifically, it designed a game that trades off between CL and Jacobian regularization to solve the DRO problem constrained by the data manifold. Both theoretical and empirical results show the robustness of the proposed method.

**Strengths:**

1. The proposed method is simple and effective.
2. This paper provides a comprehensive analysis of their proposed method.

**Weaknesses:**

1. This paper should be further polished and re-organized. The introduction and methodology section should be more concise, and some contents should be moved to a more relevant part. I provide the details as follows: 1) the introduction of WDRO from line 49 to 71 can be more brief. 2) I can understand that the authors discussed the most relevant literature in Section 1.1, but the structure is a little strange. I suggest considering this part as a discussion part such as "Relation to xxx".  3) In the methodology, I also suggest the authors provide the literature or experiment support for the sentence that "we should emphasize that the learned representation for manifold from neural networks is not sufficient for extracting tangent space". 4) Eq (2) as a significant part of the methodology is mentioned in Section 1, which makes the methodology separate. I suggest the comparison between WDRO and MWDRO in the introduction being high-level, while its details can be put into Section 3 as the motivation.
2. Some significant ablation studies are missing. For example, the effect of $\lambda$ and model architectures has not been discussed. I suggest the authors test a range of values of $\lambda$ to test its effect. As for the model architectures, the authors only provided ResNet18, which is insufficient. I suggest the authors try some state-of-the-art models such as ConvNext and Swin.
3.  The authors should clarify the advantages of the proposed method compared with [1] in Related Works and Experiments.

[1] Liu J, Wu J, Li B, et al. Distributionally robust optimization with data geometry[J]. Advances in neural information processing systems, 2022, 35: 33689-33701.

**Questions:**

1. Could you please discuss the effect of $\lambda$ and how to select the optimal value?
2. Could you verify the effectiveness of your proposed method on different model architectures such as ConvNext or Swin?

---

> ### Author Response · Authors · 2024-11-23
> **To Reviewer P9ds**
>
> We thank the reviewer for the thoughtful comments, and address your concerns below.
>
> **W1**, Re-organized of the paper; More support for the backgrounds.
>
> Thank you for your kind suggestions. In our just uploaded temporary revised version, we have revised some description and added additional visualizations and experimental supports. We will carefully modify our paper’s structure after collecting all the final comments from the reviewers and ACs after this rebuttal phase.
>
> Figure 11 demonstrates that our model successfully captures meaningful geometric structure within the data, learning tangent vectors that correspond to interpretable transformations of objects. These transformations appear to align with sensible variations in parts of the objects. In contrast, other methods fail to extract such structured geometric information from the data.
>
> **W3**, The advantages of the proposed method compared with [1].
>
> Thanks for this question. In general, our work and [1] have the following two key differences:
>
> **1. On manifold learning approach:** The paper [1]  employs a k-nearest neighbor graph to capture the geometric structure of the dataset, where the shortest paths approximate geodesic distances to guide the transportation of probability mass. Their Geometric Wasserstein DRO framework constrains the distribution support set to the observed data points. On the other hand, our approach learns the data manifold structure through neural network. This continuous, differentiable representation offers several advantages: it captures smooth geometric variations in the data manifold rather than discrete graph approximations, furthermore, it naturally accommodates out-of-sample points through the learned feature extractor.
>
> **2. On uncertainty set construction:** Their DRO formulation restricts the uncertainty set to distributions over the input data, essentially performing a re-weighting of existing samples without the capacity to incorporate novel data points. Their uncertainty set effectively alleviate the over-pessimism of WDRO, but not well-suited for scenarios involving data perturbations, such as adversarial attack. In contrast, our work specifically addresses the uncertainty introduced by data perturbations. Experimentally, their model achieves less than 30\% accuracy on PGD-attacked ($\epsilon = 0.03$) examples from the CIFAR-10 dataset (same setting with Table 3). Our model demonstrates improved performance against unseen perturbations.
>
> **Q1**, Discuss the effect of $\lambda$ and how to select.
>
> Let us explain the parameters $\lambda_1$ and $\lambda_2$ separately:
>
> In our paper, $\lambda_1$ serve as the coefficient of the Jacobian term in the manifold-guided game $\mathcal{L}^*(\theta)$. As shown in Figures 2 and 8, our method successfully extracts manifold information for $\lambda_1$ values of both $0.1$ and $0.01$. Empirical results show that the performance is not sensitive to the specific choice of the hyperparameter $\lambda_1$ within this range. In our experiments, we set the regularization coefficient of the Jacobian to be $0.01$ unless otherwise noted.
>
> The parameter $\lambda_2$ controls the magnitude of the manifold-guided game $\mathcal{L}^*(\theta)$ when conduct distributional robust optimization. Inspired by the  advances linking contrastive learning and the classification tasks [2,3], we incorporate $\mathcal{L}^*(\theta)$ as a regularization term in the MWDRO framework (6). Similar to well-known regularization techniques like L1/L2-norm regularization and  dropout, this term, while limiting the neural network's capacity to some degree, ultimately enhances both the robustness and accuracy of the classification.
>
> **W2** \& **Q2**, Verify the effectiveness of your proposed method on different model architectures
>
> We have tried some toy neural networks (for synthetic toy datasets, in revised version), LeNet (for MNIST dataset), Resnet-18/32/50 (across different tasks as shown in our experiments). The implementation of our methods does not depend on specific model architecture. It is interesting to observe how our methods perform under more advanced architectures. Given the time limit of rebuttal, we plan to conduct more experimental studies on ConvNext and SWIN as our future work.
>
> ## Reference
>
>
> [1] J. Liu et al., Distributionally robust optimization with data geometry[J]. NeurIPS'23
>
> [2]  W. Huang et al., Towards the generalization of contrastive self-supervised learning. ICLR’23
>
> [3] J. HaoChen et al., Provable Guarantees for Self-Supervised Deep Learning with Spectral Contrastive Loss.  NeurIPS'21

---

> > ### Author Response · Authors · 2024-11-28
> > **Looking forward to further feedback**
> >
> > We sincerely appreciate your insightful comments, which have greatly helped us enhance the clarity and quality of our manuscript. As noted in our previous response, we have revised the paper in accordance with your suggestions. We hope our revisions have satisfactorily addressed your concerns. If you have any further comments, we will gladly make every effort to address them.

---

> > ### Comment · Reviewer_P9ds · 2024-11-28
> >
> > Thank you for your response. I would like to keep my rating.

---

### Official Review · Reviewer_Q211 · 2024-10-30

**Soundness:** 3
**Presentation:** 3
**Contribution:** 2
**Rating:** 6
**Confidence:** 3

**Summary:**

This paper introduces a manifold-based Wasserstein Distributionally Robust Optimization (WDRO) method aimed at improving the robustness of deep learning models. Furthermore, to tackle the challenges posed by the data manifold structure, the authors propose a game that integrates contrastive learning with Jacobian regularization.

**Strengths:**

S1. The paper is well-organized and presents its ideas clearly.
S2. The proposed methods demonstrate superior performance compared to state-of-the-art algorithms.
S3. The design of the manifold-guided game is a novel concept, leveraging neural networks to encode manifold information, particularly given the absence of a closed-form representation for the data manifold.

**Weaknesses:**

W1. The motivation for this work could be strengthened. In the Introduction, the authors mention that selecting an appropriate threshold for uncertainty is challenging and introduce a new uncertainty set by assuming that data is supported on a manifold. However, constraining the uncertainty set to manifolds also raises the same issue of threshold selection.
W2. The explanation of how the game aids in extracting tangent information is not intuitive enough. Including a figure to illustrate this concept would be beneficial.
W3. As the experimental results show, MWDRO takes more time compared to other algorithms, and it would be desirable to have a further discussion on this part of the reason.

**Questions:**

Q1. In Figure 3, the accuracy of each model is already distinguishable even when the noise size is 0. Could you provide further discussion on this observation?
Q2. Is the use of contrastive learning essential in the design of the manifold-guided game? Are there alternative optimization objectives that could be utilized instead?
Q3. As noted in W3, could you further analyze which phase of training contributes to the increased time required for MWDRO?
Q4. Is it possible to further optimize the design of the algorithm to reduce the time to loss? For example, by utilizing simplified geometric computations or low-rank approximations.

---

> ### Author Response · Authors · 2024-11-23
> **To Reviewer Q211  1**
>
> We thank the reviewer for the thoughtful comments, and address your concerns below.
>
> **W1**, Concerns on the motivation and the threshold selection.
>
> The threshold selection in DRO presents a fundamental trade-off between robustness and accuracy. While our method doesn't completely resolve this dilemma, it significantly alleviates it through manifold constraints, as demonstrated in the enclosed table. Key benefits of our approach for alleviate the over-pessimism feature are summarized as follows:
>
> 1, **We have larger valid threshold range:** the performance of the classical WDRO deteriorates rapidly with even slightly larger thresholds ($\delta$ from 0.01 to 0.02); Our MWDRO maintains performance with larger thresholds due to manifold constraints.
>
> 2, **We have better Robustness-Accuracy trade-off:** Classical WDRO includes irrelevant out-of-manifold samples while our method focuses on semantically meaningful variations. Experimental results in **Appendix F** show that our MWDRO significantly improve the WDRO (or uni-DRO [2]) method across several datasets and tasks under the same setting. We add more results of MWDRO with different thresholds in Table 2,3,4 of **Appendix F.3**.
>
> The table below shows that the clean accuracy of the Uni-DRO method drops more significantly than ours when we adopt a larger threshold $\delta = 0.02$.
>
> | Clean accuracy | Uni-DRO($\delta$ = 0.01) | Uni-DRO(0.02) | ours(0.01) | ours(0.02) |
> | -------------- | ------------------------ | ------------- | ---------- | ---------- |
> | Cifar-100      | 67.67                    | 58.71         | 69.58      | 67.25      |
> | Tiny-ImgNet    | 41.72                    | 31.18         | 43.82      | 40.22      |
>
>
>
> **W2**, Add figures that illustrate this concepts.
>
> Thank you for your suggestions. We have added four types of  visual illustrations to better demonstrate our method. We showcase the method's ability to learn meaningful geometric structures through various perspectives:
>
> 1, The schematic illustration of our methodology (Figure 9);
>
> 2, analysis of manifold dynamics on toy datasets (Figure 10);
>
> 3, comparison of learned tangent spaces across different models (Figure 11);
>
> 4, visualization of perturbation trajectories on the MNIST dataset (Figure 12).
>
>
>
> **W3**, The concern on time complexity.
>
> In **Appendix F.5**, we compare computational running times across various methods using the same number of epochs (200 epochs). As anticipated, our method requires more time than WDRO because it incorporates additional data manifold information while maintaining a similar workflow. The primary additional computation comes from extracting tangent information through SVD of the Jacobian matrix. To address this, we implemented several optimization techniques detailed in **Appendix E**, including the randomized SVD method. This method proves particularly well-suited for our approach because, as demonstrated in Figure 2, our model's Jacobian exhibits only a few dominant singular values. This characteristic enables the randomized SVD method to achieve both high efficiency and accuracy in approximating the decomposition [1].
>
> The experimental results show that our method is 50\% slower than the classical WDRO per epoch. However, our method exhibits faster convergence — It is rational because MWDRO works with a more focused uncertainty set (a subset of WDRO's uncertainty set). This faster convergence allows us to implement early stopping and achieve comparable performance with fewer epochs. Specifically, we monitor the robust performance on the validation set during training and stop training when performance stops improving more than 5 epochs.
>
> Finally we achieve similar performance(our model outperforms others) while using similar training time. We include this new results in **Appendix. F.5** ''Time comparison'' of our paper.

---

> ### Author Response · Authors · 2024-11-23
> **To Reviewer Q211 2**
>
> **Q1**, In Figure 3, the accuracy of each model is already distinguishable even when the noise size is 0.
>
> This is because robust models often sacrifice performance on clean datasets for achieving robustness [3]. This observation aligns with experiments from previous studies [2,3]. Classical WDRO shows the largest drop in performance on clean datasets as shown in Figure 3 and Table 2,3,4. In fact, this aligns with the “overly pessimistic” behavior we described in the introduction: the model becomes overly focused on certain samples within the uncertainty set, which can end up harming its overall performance. Our method better preserves clean accuracy through manifold constraints as we discussed in “Reply to **W1**”.
>
> **Q2**,  Is the use of contrastive learning essential in the design of the manifold-guided game?
>
> Contrastive learning (CL) is essential for our method for several theoretical and practical reasons:
>
> **1,** Theoretically, CL shows strong connection to downstream classification tasks [4,5];
>
> **2,** it naturally aligns with manifold learning objectives as studied in [6,7] and is complementary to Jacobian regularization.
>
> **3,** the augmentation operation of CL enhanced manifold density, making it effective across diverse datasets;
>
> **4,** Additionally, its efficient positive/negative pairing mechanism forces the model to become sensitive to semantic changes in the data, while reduces the number of significant singular vectors. This property improves scalability for large datasets.
>
> A potential alternative for CL might be the auto-encoder. However, the auto-encoder proved less effective in our experiments. Auto-encoder did not perform as well as CL for the downstream classification tasks; Moreover, CL's ability to align positive pairs substantially reduces the number of top singular vectors, as demonstrated in Figure 2, which significantly benefits subsequent random SVD computations (detailed in **Appendix E**).
>
> **Q3**, Which phase of training contributes to the increased time required for MWDRO?  Further optimize the design of the algorithm to reduce the time like low-rank approximation?
>
> The classicial WDRO method is time-consuming, and incorporating the manifold-guided game does not substantially increase the overall processing time. The main additional computation compared with WDRO comes from extracting tangent information through SVD of the Jacobian. For further details, please refer to “Reply to **W3**.”
>
> **Q4**,  Further optimize the design of the algorithm to reduce the time like low-rank approximation? For example, by utilizing simplified geometric computations or low-rank approximations.
>
> Thank you for your suggestions regarding acceleration. We have explored several techniques to reduce the time complexity, including the randomized SVD [1]. This approach leverages low-rank approximation, which is well-suited to our model due to the distinctive "spiked" distribution of its Jacobian matrix's singular values, as illustrated in Figures 2 and 8. This rapid decay in singular values enables randomized SVD to achieve both higher accuracy and improved computational efficiency, as discussed in [1]. We provide detailed discuss in **Appendix E** and present the experimental results in **Appendix F.5**.
>
> ## Reference
>
> [1] Halko et al., Finding structure with randomness:
> Probabilistic algorithms for constructing approximate matrix decompositions. SIAM review, 2011.
>
> [2] Bui et al., A unified Wasserstein distributional robustness framework for adversarial training. NeurIPS'22
>
> [3] Wang et al., Balance, Imbalance, and Rebalance: Understanding Robust Overfitting from a Minimax Game Perspective. NeurIPS'23
>
> [4] Huang et al., Towards the Generalization of Contrastive Self-Supervised Learning. ICLR'23
>
> [5] Haochen et al., Provable Guarantees for Self-Supervised Deep Learning with Spectral Contrastive Loss. NeurIPS'21
>
> [6] Hu et al., Your Contrastive Learning Is Secretly Doing Stochastic Neighbor Embedding. ICLR'23
>
> [7] Tan et al., Contrastive Learning Is Spectral Clustering On Similarity Graph. ICLR'24

---

> > ### Comment · Reviewer_Q211 · 2024-11-26
> >
> > Thank you for your careful consideration of the review comments and the revisions made. Your response effectively addresses my concerns, and I am willing to increase the score.

---

> ### Author Response · Authors · 2024-11-27
> **Thank You for Your Feedback**
>
> We deeply appreciate your time and effort in reviewing our paper and providing insightful and constructive comments. We are glad to hear that our responses address your concerns. If you have any further suggestions or questions, we will make every effort to address them.

---

### Official Review · Reviewer_qtHQ · 2024-11-04

**Soundness:** 2
**Presentation:** 3
**Contribution:** 3
**Rating:** 5
**Confidence:** 5

**Summary:**

This paper proposes performing distributional robustness (DR) over data manifolds. It develops the dual form of DR over data manifolds and    uses the top singular principal vectors of the Jacobian matrix to characterize the tangent space of the data manifolds.

**Strengths:**

- The idea of developing distributional robust on data manifolds is intuitive and interesting.
- The paper is well-written.

**Weaknesses:**

- It is unclear why we can use the linear span of top singular vectors of the Jacobian matrix of g can characterize the tangent space of data manifolds. The current explanations do not really convince me. Moreover, the experiments in Figure 2 only demonstrate that there are some dominant singular vectors and I cannot see how it explains why $T_{appr}(x, \tau_0)$ is a subspace of the tangent space.
- It is also unclear to me why the game in Section 4.2 helps extract tangent information. Evidently, the CL loss encourages the features of data examples in $M_{SI}(x)$ to be close, whereas the Jacobian regularization suppresses the feature variations for perturbations of x across all directions. They seem to share the same purpose and nature.
- It is unclear to me why we can use the $pt^t$ in Alg 2 to approximate the geodesic distance from $q^t$ to $q^0$.
- It is unclear how to do Exp operator to retract to the data manifolds. This seems impossible because we only assume that data lie in manifolds but we do not have anymore information of these manifolds. The Exp operator appears closed-form just for some simple and specific manifolds.
-  The proposed method is very computationally expensive, even more expensive than adversarial training. This is because we need to compute a trajectory for each data example which requires computing Jacobian matrix and also the gradients to data examples. The analysis on the computational complexity is necessary.
- The experiments do not demonstrate the benefit of performing DR on data manifolds. It would be better if the authors keep the same CL loss and Jacobian regularization, while replacing their manifold DR by standard DR on data space for a comparison. Moreover, the authors should provide the visualization of $q^1$,...,$q^t$ on the trajectory to see if we can make them on the data manifolds.

**Questions:**

Please address my questions in Weaknesses.

---

> ### Author Response · Authors · 2024-11-23
> **Response to Reviewer qtHQ 1**
>
> We thank the reviewer for the thoughtful comments, and address your concerns below.
>
> **W1**, Why can we use the linear span of top singular vectors of the Jacobian matrix of $g$ to characterize the tangent space of data manifolds?
>
> The Jacobian matrix $J_\mathbf{g}(x)$, which contains the partial derivatives of the features with respect to the input data, uncovers how the representation changes in response to input variations. Through training with the manifold-guided game, the model maintains strong responses along directions of semantic variation in the tangent while exhibiting weaker responses in other directions. This results in a distinctive “spiked” distribution of the Jacobian matrix’s singular values (as shown in Figure 2 and 8), where the top singular vectors naturally correspond to the semantic variate directions in the tangent.
>
> **The role of the game mechanism:** The contrastive learning component encourages the model to produce discriminative representations for samples with distinct semantics, while Jacobian regularization suppresses feature variations caused by perturbations of $x$ in all directions. Together, these objectives enable the model to effectively capture the intrinsic structure of the data. In the revised version, we include additional visualizations to illustrate this effect more intuitively in **Appendix G**.
>
> **W2**, Why the game in Section 4.2 helps extract tangent information?
>
> The effectiveness of our game mechanism can be understood through the lens of representation learning theory. Following [1], the InfoNCE loss of the contrastive learning can be decomposed into two key effects: **1, Uniformity,** the representations of the negative pairs are pushed further apart. This encourages maximal change in $\mathbf{g}(\cdot)$ along semantically meaningful directions within the manifold's tangent space. **2, Alignment,** This brings representations of semantic-preserving augmentations closer together. While the Alignment effect shares some purpose with Jacobian regularization, the Uniformity effect is crucial for identifying the principal directions of variation on the manifold. Together, they shape the encoder $\mathbf{g}(\cdot)$ to have two essential properties, which are summarized as follows:
>
>  **1, Sensitivity.** The encoder $\mathbf{g}(\boldsymbol{x})$ should change most when the sample $\boldsymbol{x}$ moves along the semantic variant directions within the data manifold;
>
> **2, Insensitivity.** The encoder $\mathbf{g}(\boldsymbol{x})$ should change minimally when moves off the manifold (i.e., $\mathcal{M}^\perp$), or towards the Semantic-Invariant region (i.e., $\mathcal{M}_{\text{SI}}(\mathbf{x})$).
> These properties ensure that the top singular vectors of $J_g(x)$ provide an effective approximation of the semantic variate directions within the tangent as discussed in our response to **W1**.
>
> Further interpretations can be found in **Appendix B**, with visual illustrations provided in **Appendix G**. We believe our approach is novel and hope our work can inspire the further exploration to other optimization problems on manifolds  where closed-form expressions are unavailable.
>
> **W3**, Why can we use the $pt^t$ in Alg 2 to approximate the geodesic distance from $q_t$ to $q_0$？
>
> Computing exact geodesic distances on general manifolds is computationally intensive and often intractable. It is intuitive to approximate the trajectory length using the accumulated discrete step sizes in practice. Our key insight is to leverage the geodesically strong concavity property of the surrogate loss in Eq.(7) to provide theoretical guarantees for this approximation. This strong concavity ensures geometric convergence in optimization, which means the step sizes decrease at a rate following a geometric sequence [2]. This guarantees that the estimated geodesic distance approximates the true geodesic distance within a constant factor, as stated in Lemma 1.

---

> ### Author Response · Authors · 2024-11-23
> **Response to Reviewer qtHQ 2**
>
> **W4**, How to do Exp operator to retract to the data manifolds?
>
> In practice, when the exponential map $Exp_x(v)$ lacks a closed-form expression, it is commonly approximated with a retraction map $R_x(v)$. As suggested in [3], the linear retraction $R_x(v) = x + v$ is widely used, as it is a first-order approximation of $Exp_x(v)$. In this work, we adopt this linear approximation, as noted in line 462-463.
>
> **W5**, The concern on time complexity
>
> In **Appendix F.5**, we compare computational running times across various methods using the same number of epochs (200 epochs). As anticipated, our method requires more time than WDRO because it incorporates additional data manifold information while maintaining a similar workflow. The primary additional computation comes from extracting tangent information through SVD of the Jacobian matrix. To address this, we implemented several optimization techniques detailed in **Appendix E**, including the randomized SVD method. This method proves particularly well-suited for our approach because, as demonstrated in Figure 2, our model's Jacobian exhibits only a few dominant singular values. This characteristic enables the randomized SVD method to achieve both high efficiency and accuracy in approximating the decomposition [4].
>
> The experimental results show that our method is 50\% slower than the classical WDRO per epoch. However, our method exhibits faster convergence — a natural consequence of working with a more focused uncertainty set (a subset of WDRO's uncertainty set). This faster convergence allows us to implement early stopping and achieve comparable performance with fewer epochs. Specifically, we monitor the robust performance on the validation set during training and stop training when performance stops improving more than 5 epochs.
>
> Finally we achieve similar performance(our model outperforms others) while using similar training time (less than 1.2x). We include this new results in **Appendix F.5** ''Time comparison''.
>
>
>
> **W6**, The benefit of performing DR on data manifolds; The visualization of trajectory.
>
> We have expanded our analysis by incorporating additional ablation studies and visual evidence.
>
> The experiments in Figure 7 demonstrate the importance of performing the distributional optimization on data manifold. In Phase 1 of Algorithm 1, we initially train the model for $N_0$ epochs using $L^*(\theta)$ (Eq.(8), which combines the CL and Jacobian regularization) to learn the data manifold structure before implementing the MWDRO algorithm. Our ablation study in Figure 6 varies $N_0$ and reveals three key findings:
>
> 1, both robust and clean accuracy remain low for small values of $N_0$, indicating insufficient manifold learning results in suboptimal performance;
>
> 2, Besides, the performance improves as $N_0$ increases as the model extracts more refined geometric information;
>
> 3, the improvement plateaus once the model adequately captures the manifold structure ($N_0 > 40$). These observations demonstrate that DRO tends to be overly pessimistic without proper manifold constraints, highlighting the crucial role of accurate manifold information for effective robust optimization.
>
> For the visualization aspect, **Appendix G** presents four visual illustrations that demonstrate key aspects of our research. We showcase the method's ability to learn meaningful geometric structures through various perspectives:
>
> 1, The schematic illustration of our methodology (Figure 9);
>
> 2, analysis of manifold dynamics on toy datasets (Figure 10);
>
> 3, comparison of learned tangent spaces across different models (Figure 11);
>
> 4, visualization of perturbation trajectories on the MNIST dataset (Figure 12).
>
> ## Reference
>
> [1] Wang and Isola, Understanding Contrastive Representation Learning through Alignment and Uniformity on the Hypersphere. ICML'20
>
> [2] Boumal, An introduction to optimization on smooth manifolds, 2023.
>
> [3] Bécigneul and Ganea, Riemannian adaptive optimization methods, ICLR'19.
>
> [4] Halko et al., Finding structure with randomness: Probabilistic algorithms for constructing approximate matrix decompositions. SIAM review, 2011.

---

> > ### Author Response · Authors · 2024-11-28
> > **Looking forward to further feedback**
> >
> > We sincerely appreciate your insightful comments, which have greatly helped us enhance the clarity and quality of our manuscript. As noted in our previous response, we have revised the paper in accordance with your suggestions. We hope our revisions have satisfactorily addressed your concerns. If you have any further comments, we will gladly make every effort to address them.

---

> > ### Comment · Reviewer_qtHQ · 2024-12-01
> > **Response to the author rebuttal**
> >
> > Thanks for answering my questions. Indeed, I read your rebuttal and reread the paper. There are some reasons that border me to increase the score for this paper.
> > -  I still do not convince of the connection between the linear span of top singular vectors and tangent vectors of data manifold. I read carefully your answer and the paper again, however, I still feel the arguments are still vague. You explained that *the Jacobian matrix , which contains the partial derivatives of the features with respect to the input data, uncovers how the representation changes in response to input variations*. I agree that there is a connection between the partial derivative of a function and tangent space. However, it should be relevant to the **local defining function** as in Definition 3.10 in [1].
> > -  You further explained *through training with the manifold-guided game, the model maintains strong responses along directions of semantic variation in the tangent while exhibiting weaker responses in other directions. This results in a distinctive “spiked” distribution of the Jacobian matrix’s singular values (as shown in Figure 2 and 8), where the top singular vectors naturally correspond to the semantic variate directions in the tangent*. Why do not you use the partial derivatives in Jacobian matrix directly as the tangent vectors? Why do you use the top singular vectors of this matrix?
> > -  I believe that understanding the manifold structure of data manifolds is crucially important. To me, even if you can effectively demonstrate or prove that the linear span of the top singular vectors of the Jacobian matrix has a strong connection to the tangent space of data manifolds, it is well deserved for a publication.
> > -  Regarding Algorithm 2, in Line 2, it is unclear to me how you compute the Riemannian derivative of the geodesic distance $d_g$ w.r.t. $q^t$. It seems that this is the second term that is relevant to the $u$ vector. But it is unclear to me how it is and why you do not do projection onto the tangent space for the gradient of the geodesic distance $d_g$ w.r.t. $q^t$.
> > -  Finally, you answered that you use $x + v$ (i.e., $q^{t+1} = q^t + \alpha gard^t$ in your algorithm) as the retraction. However, it is a vector on the tangent space not a retraction onto the data manifold.
> >
> >
> >
> > [1] Boumal, N., 2023. An introduction to optimization on smooth manifolds. Cambridge University Press.

---

> ### Author Response · Authors · 2024-12-02
> **Resonse to the reviewer qtHQ 3**
>
> We thank the reviewer for the thoughtful comments, and address your concerns below.
>
> **A1.** Thanks for this question.
> To explain why we don't explicitly construct a local defining function $h(x) = 0$ for extracting the tangent information, we refer to the contractive auto-encoder methodology [2] as an example. In [2], the authors  constructed a topological atlas of charts characterized by the principal singular vectors of the representation mapping's Jacobian. Even though the neural network used in [2] is relatively simple compared with the more complicated networks used nowadays (note [2] is published in 2011), it already demonstrated good results. This line of approach bypasses the challenging task of obtaining local defining functions for manifolds by directly extracting tangent information, a methodology proven effective and efficient in both previous works and ours.
>
> Also, we would like to mention that the concept of using neural networks to extract tangent space information has been validated in several previous articles. For example, Figure 6 in [1] and Figure 1 in [2] clearly demonstrate the extracted tangent vectors. Our experimental results in Appendix G (Visualizations) provide additional evidence supporting this. Furthermore, we validated the effectiveness of this approach in addressing the MWDRO problem of concern.
>
> **A2.** Please note that precisely acquiring the data manifold remains an open challenge, and previous studies suggest that neural network-based approaches could provide promising approximations.
> It is well known that the columns of the Jacobian matrix $J(x)$ represent partial derivatives along each input dimension. In high-dimensional spaces, using these partial derivatives directly would yield an excessive number of potential tangent vectors, with many directions being redundant or capturing noise.
> For instance, in [2], the authors also used the span of the top singular vectors to approximate the data manifold's tangent information.
>
> **A3.** Previous studies [1,2] have shown that it is feasible to leverage the neural network's powerful representational capability to approximately extract the tangent information of this manifold. Building on these insights, we employed contrastive learning techniques to extract the semantic variation subspace within the tangent space, requiring fewer top singular vectors (in Figures 2 and 8). We validated the effectiveness of this approach in addressing the MWDRO problem of concern.
>
> For future work, we hope our currently proposed method could inspire further explorations of other optimization problems on manifolds, where closed-form expressions are unavailable.
>
> **A4.** As you correctly point out, we can project the gradient of the geodesic distance onto the tangent space. Please note that the gradient of the squared geodesic distance  $ \nabla_{q^t} d^2_g(q^t, q)$ can be expressed as $ -2 d_g(q^t, q)  \hat{u}$,  where $\hat{u}$ is the unit-length initial tangent vector of the geodesic from point $q^t$ to point $q$. In our experiments, the $\hat{u}$ vector is approximated by $\mathrm{Proj}^{\tau_0}_{M} \Big( \frac{q - q^t}{\| q^t - q  \|_2} \Big)$.
>
> **A5.** Thanks for this question, and let us explain it in more detail.
> Following the work [3], when $exp_x(v)$ lacks a closed-form expression, it is common practice to employ the map $R_x(v) = x+v$ as a computationally efficient approximation. Since a manifold is locally homeomorphic to Euclidean space in a neighborhood of each point, we can approximate the exponential map $Exp_{q_t}(\alpha \nabla^M f(q))$ using $q^{t+1} = q^t + \alpha\nabla^M f(q)$ with a small step size $\alpha$ (to approximate the gradient flow on the manifold).
>
> We provide several visualizations to demonstrate these effects in our experiments. Moreover, some other empirical validation in previous works can be found [1,2]. For example,  in Figure 6 of [1], the points generated by adding tangent vectors to manifold points successfully capture meaningful transformations (such as rotation and scaling) of the original images. These empirical results suggest that such an approximation maintains the data's local geometric structure, making it suitable for practical applications.
>
> **Reference:**
>
> [1] Patrice Y Simard, Yann A LeCun, John S Denker, and Bernard Victorri. Transformation Invariance in Pattern Recognition – Tangent Distance and Tangent Propagation. NN: Tricks of the Trade, 2nd edn., LNCS 7700, 2002)
>
> [2] Salah Rifai, Yann N Dauphin, Pascal Vincent, Yoshua Bengio, and Xavier Muller. The manifold tangent classifier. Advances in neural information processing systems, 2011.
>
> [3] Bécigneul and Ganea, Riemannian adaptive optimization methods, ICLR'19.

---

### Official Review · Reviewer_v52n · 2024-11-05

**Soundness:** 3
**Presentation:** 3
**Contribution:** 3
**Rating:** 6
**Confidence:** 3

**Summary:**

The authors propose a manifold-based DRO method that incorporates the geometric structure of training data to construct the uncertainty set. By integrating contrastive learning with Jacobian regularization, the method effectively captures the data manifold, providing theoretical guarantees for robustness. Experimental results on benchmark datasets demonstrate improved accuracy and robustness compared to conventional DRO methods.

**Strengths:**

1. The proposed manifold-based DRO method effectively captures the geometric structure of training data, leading to better robustness and accuracy compared to traditional DRO methods. This is achieved by integrating contrastive learning with Jacobian regularization, which helps in maintaining the data manifold structure.
2. The method provides theoretical guarantees for robustness by utilizing a novel approach to approximate geodesic distances on manifolds. This ensures that the model remains reliable even when faced with distributional shifts in the data.
3. Despite the complexity of incorporating manifold constraints, the proposed method is easy to implement and computationally feasible. The authors demonstrate this through experiments on popular benchmark datasets, showing that the method achieves superior performance without significant increases in computational cost.

**Weaknesses:**

I believe the motivation of this paper is reasonable and the method is effective. However, I would like to ask the authors whether the effectiveness of the algorithm depends on specific data? What kind of data characteristics would give MWDRO a greater advantage?

**Questions:**

See Weaknesses above.

---

> ### Author Response · Authors · 2024-11-23
> **Response to Review v52n**
>
> We thank the reviewer for the thoughtful comments, and address your concerns below.
>
> **W1**, Whether the effectiveness of the algorithm depends on specific data?
>
> Our method is well-suited for data with low-dimensional manifold structures, such as natural images. This is because the intrinsic characteristics of such data (e.g., semantic variations) are often concentrated in a few key directions, which can be effectively captured by our manifold-guided game. A potential concern is whether our manifold-guided game remains effective for highly complex or sparse data distributions. We think the augmentation operation of CL might mitigate this concern. On the on hand, the augmentation produces a potentially infinite set of samples on the data manifold. On the other hand, the relationships among the augmented data help uncover the intrinsic geometric structure of the data, please refer to [1,2] for the justification.
>
> In our work, we demonstrate the effectiveness of our approach through extensive experiments across various datasets and tasks. MWDRO achieves strong performance across a range of datasets, including CIFAR-10/100, Tiny-ImageNet, and medical images. This robust performance across different datasets and tasks suggests that our approach effectively capture the underlying manifold structures in many real-world datasets.
>
>
> ## Reference
> [1] Wang et al., Chaos is a Ladder: A New Theoretical Understanding of Contrastive Learning via Augmentation Overlap. ICLR'22
>
> [2] Haochen et al., Provable Guarantees for Self-Supervised Deep Learning with Spectral Contrastive Loss. NeurIPS'21

---

> > ### Author Response · Authors · 2024-11-28
> > **Looking forward to further feedback**
> >
> > We sincerely appreciate your insightful comments, which have greatly helped us enhance the clarity and quality of our manuscript. As noted in our previous response, we have revised the paper in accordance with your suggestions. We hope our revisions have satisfactorily addressed your concerns. If you have any further comments, we will gladly make every effort to address them.

---

### Author Response · Authors · 2024-11-23
**Global rebuttal**

We sincerely appreciate the reviewers' valuable and insightful feedback on our paper. Following the suggestions, we have made extensive revisions, which are summarized below and highlighted in brown within the paper. The key changes are as follows:


1, We add more figures that illustrate the mechanism and effectiveness of our manifold-guided games. (see **Appendix G**).

2, We include more experimental details and additional ablation studies. (**Appendix F**)

3, We discuss more about the computational concerns. (**Appendix F.5**)

4, We conduct a broader review of related literature. (**Appendix A.2**)

Detailed responses to comments are enclosed. Please let us know if you have any additional suggestions. We will be happy to revise our paper accordingly.

Sincerely,

The authors of #8724

---

### Meta-Review · Area_Chair_CggY · 2024-12-17

**Metareview:**

The paper proposes a novel manifold-based Wasserstein-based distributionally robust optimization (WDRO) model. The paper designs a game that integrates contrastive learning and Jacobian regularization to identify the directions of semantic variation in the data manifold. The paper proposes an efficient algorithm for solving the WDRO model. The paper shows experimental results and theoretical guarantee of robustness. The reviewers pointed out the novelty of the proposed method, theory, good writing, and good experiments as the strengths of the paper.

The final scores of reviewers were 6,6,6,5,5. Reviewer qtHQ had a score of 5. The reviewer pointed out many weaknesses that require clarification due to unclearness, a point regarding expensive computational cost, and provided suggestions for better experiments. The authors provided a rebuttal and updated the paper following the suggestions. The reviewer asked some follow-up questions, and the authors provided further detail. The reviewer did not provide a response to this (including the private reviewer-AC discussion phase.)

Reviewer m5cH also rated the paper as a 5. The reviewer pointed out 5 weaknesses, but was well addressed by the authors in the rebuttal. The authors have also updated the paper to incorporate the discussions and additional results.

Reviewer Q211 was satisfied with the rebuttal and raised the score from 5 to 6.

Overall, I would like to recommend acceptance. If accepted, I suggest the authors to further improve the paper based on the final discussions with Reviewer qtHQ.

**Additional Comments On Reviewer Discussion:**

Please see the meta review about the discussions and changes during the rebuttal period.

---

### Decision · Program_Chairs · 2025-01-22

Accept (Poster)